# Spatial and single-nucleus transcriptomic analysis of genetic and sporadic forms of Alzheimer's disease

Emily Miyoshi [1,2,10], Samuel Morabito [2,3,4,10], Caden M. Henningfield[1,2], Sudeshna Das[1,2], Negin Rahimzadeh[2,3,4], Sepideh Kiani Shabestari[1,5], Neethu Michael[1,2], Nora Emerson[1,2], Fairlie Reese[4,6], Zechuan Shi[1,2], Zhenkun Cao[1], Shushrruth Sai Srinivasan[2,3,4,7], Vanessa M. Scarfone[5], Miguel A. Arreola[1,2], Jackie Lu[1], Sierra Wright[2], Justine Silva[1,2], Kelsey Leavy[2], Ira T. Lott[2,8], Eric Doran[8], William H. Yong[2,9], Saba Shahin[1,2], Mari Perez-Rosendahl[2,9], Alzheimer's Biomarkers Consortium–Down Syndrome (ABC–DS)*, Elizabeth Head[2,9], Kim N. Green[1,2] & Vivek Swarup [1,2,4] ✉

The pathogenesis of Alzheimer's disease (AD) depends on environmental and heritable factors, with its molecular etiology still unclear. Here we present a spatial transcriptomic (ST) and single-nucleus transcriptomic survey of late-onset sporadic AD and AD in Down syndrome (DSAD). Studying DSAD provides an opportunity to enhance our understanding of the AD transcriptome, potentially bridging the gap between genetic mouse models and sporadic AD. We identified transcriptomic changes that may underlie cortical layer-preferential pathology accumulation. Spatial co-expression network analyses revealed transient and regionally restricted disease processes, including a glial inflammatory program dysregulated in upper cortical layers and implicated in AD genetic risk and amyloid-associated processes. Cell–cell communication analysis further contextualized this gene program in dysregulated signaling networks. Finally, we generated ST data from an amyloid AD mouse model to identify cross-species amyloid-proximal transcriptomic changes with conformational context.

The fundamental work of pioneers like Santiago Ramón y Cajal revealed that the human brain is spatially organized at macroscopic and microscopic levels, where brain circuitry and function underlie structural organization. Single-nucleus RNA-sequencing (snRNA-seq) has revealed that brain cell populations are heterogeneous at the molecular level[1–6]. In Alzheimer's disease (AD) brains, specific cell populations have been identified as underrepresented or overrepresented relative to the cognitively healthy brain[7–11], revealing an axis of selective vulnerability to resilience in neurodegeneration[11,12] and providing foundational knowledge of the genes, *cis*-regulatory elements and networks altered in AD[13]. The functional consequences of transcriptomic changes in disease-related cell populations remain elusive, and spatial context is critical for solving this puzzle.

Here we examined spatial and single-nucleus transcriptomes of clinical AD samples in early-stage and late-stage pathology and AD in Down syndrome (DSAD). Individuals with Down syndrome (DS) aged >65 years old have an 80% risk of dementia[14]. Despite shared features between sporadic AD (sAD) and DSAD[15,16], there are no single-cell or spatial transcriptomic (ST) studies comparing these populations—presently only one single-cell study of DS brains[17]. Focusing on DSAD as a

**Fig. 1 | ST and single-nucleus transcriptomic analysis of genetic and sporadic forms of AD. a**, We performed ST experiments in the human frontal cortex and the mouse brain using 10x Genomics Visium. Human samples—$n$ = 10 cognitively normal controls, $n$ = 9 early-stage AD, $n$ = 10 late-stage AD and $n$ = 10 DSAD (median 1,316 genes per spot; $n$ = 115,251 ST spots; Supplementary Table 1). Mouse samples—$n$ = 10 WT and $n$ = 10 5xFAD aged 4 months; $n$ = 10 WT and $n$ = 10 5xFAD aged 6 months; $n$ = 10 WT and $n$ = 10 5xFAD aged 8 months; $n$ = 8 WT and $n$ = 12 5xFAD aged 12 months (median 2,438 genes per spatial spot; $n$ = 212,249 ST spots; Supplementary Table 2). **b**, Two representative human ST samples from each of the disease conditions, each spot colored by cortical annotations from BayesSpace[22] clustering analysis. **c**, One representative mouse ST sample from WT and 5xFAD at each time point, each spot colored by brain region annotations derived from BayesSpace clustering analysis. **d**, We performed

snRNA-seq in the frontal cortex and PCC from cognitively normal control donors ($n$ = 27 FCX and $n$ = 27 PCC) and DSAD donors ($n$ = 21 FCX and $n$ = 21 PCC). We also included snRNA-seq data from three previous studies of the cortex in AD[7–9] ($n$ = 27 controls, $n$ = 23 early-stage AD and $n$ = 48 late-stage AD). **e**, UMAP plot depicting a two-dimensional view of the cellular neighborhood graph of 585,042 single-nuclei transcriptome profiles. Each point in this plot represents one cell, colored by their cell-type annotations derived from Leiden clustering[58] analysis. EX, $n$ = 229,041; INH, $n$ = 90,718; MG, $n$ = 20,197; ASC, $n$ = 57,443; OPC, $n$ = 23,053; ODC, $n$ = 153,182; PER, $n$ = 4,659; END, $n$ = 3,637; FBR, $n$ = 2,403 and SMCs, SMC, $n$ = 709. See Table 1 for additional cluster name abbreviations. Illustrations were created with Biorender.com. EX, excitatory neurons; INH, inhibitory neurons; MG, microglia; ASC, astrocytes; ODC, oligodendrocytes; PER, pericytes; END, endothelial cells; FBR, fibroblasts; SMCs, smooth muscle cells.

genetic form of AD, due to the vastly increased risk from triplication of *APP* on chromosome 21 (chr21), provides opportunities for comparative analyses with sAD and to further our understanding of AD genetics. Our analyses uncovered shared and distinct transcriptomic changes in DSAD and sAD and identified relationships between genetic risk and altered transcriptomic signatures.

We also performed ST in 5xFAD mice, an amyloid model of AD, at four time points to facilitate cross-species comparisons. snRNA-seq studies in AD mouse models have found glial disease-associated cell states[18,19]; however, it has been challenging to robustly identify their human counterparts. Although mouse models offer advantages for studying disease, AD clinical trial failures of drugs successful in mice raise questions about the translatability of mouse findings. Translational and druggable targets may be nominated via cross-species integrative approaches, as demonstrated by our previous work[20,21]. We paired our ST experiments with fluorescent amyloid imaging to identify cross-species amyloid-proximal gene expression changes. Together, our multifaceted experimental and analytical approaches illuminate spatially restricted and cell-type-specific transcriptomic changes across AD subtypes and species.

## Results

### ST and cellular transcriptomics of AD

We performed a spatially resolved cross-species gene expression study of AD by generating ST data (10x Genomics Visium) from postmortem human prefrontal cortex (FCX; $n$ = 10 cognitively healthy controls, $n$ = 9 early-stage AD, $n$ = 10 late-stage AD, $n$ = 10 DSAD) and 5xFAD and wild-type (WT) mouse brains ($n$ = 8–12 per group, 4–12 months; Fig. 1a–c and Supplementary Fig. 1). Unbiased clustering analysis[22] identified nine clusters in our human dataset—three white matter (WM) clusters and six gray matter (GM) clusters encompassing the cortical layers—and 15 brain-region-specific clusters in our mouse dataset. We annotated these clusters based on known marker gene expression, tissue localization and unbiased cluster marker gene detection (Supplementary Figs. 2–4, Table 1 and Supplementary Tables 3 and 4).

We also performed snRNA-seq[23] (Parse Biosciences) in cognitively healthy controls ($n$ = 27 FCX and $n$ = 27 PCC) and DSAD ($n$ = 21 FCX and $n$ = 21 PCC, 55 individuals total; Supplementary Table 5). The PCC and FCX may represent early and late AD changes, respectively[24]. We used scANVI[25,26] for reference-based integration of this dataset with three previous AD studies[7–9] (FCX; total $n$ = 27 cognitively healthy controls, $n$ = 23 early-stage AD and $n$ = 48 late-stage AD) for a total of 585,042 nuclei after quality control (Fig. 1d and Supplementary Fig. 1). Clustering analysis identified all major brain cell types, including rare vascular populations[27] (Fig. 1e), and their marker genes provided further context for their transcriptomic identities (Supplementary Fig. 1 and Supplementary Table 6; Methods). Differential abundance analysis[28] revealed widespread shifts in cell state composition, especially among microglia, astrocytes and vascular cells, encompassing both selectively vulnerable and disease-reactive states (Supplementary Fig. 6 and Supplementary Note; Methods).

## Table 1 | Abbreviations for spatial and snRNA-seq cluster names

| Cluster | Abbreviation | Dataset |
|---|---|---|
| Cortical deep layers | Ctx (deep layers) | Mouse ST |
| Cortical upper layers | Ctx (upper layers) | Mouse ST |
| Olfactory cortex | Ctx (olfactory) | Mouse ST |
| Hippocampus pyramidal layer | Hippocampus (pyr.) | Mouse ST |
| Hypothalamus/amygdala | Hypothal./amygdala | Mouse ST |
| White matter (cerebral peduncle) | WM (c. peduncle) | Mouse ST |
| Excitatory neuron | EX | Human snRNA-seq |
| Inhibitory neuron | INH | Human snRNA-seq |
| Astrocyte | ASC | Human snRNA-seq |
| Microglia | MG | Human snRNA-seq |
| Oligodendrocyte | ODC | Human snRNA-seq |
| Oligodendrocyte precursor cell | OPC | Human snRNA-seq |
| Arteriole endothelial cell | Arterial | Human snRNA-seq |
| Capillary endothelial cell | Capillary | Human snRNA-seq |
| Transport pericyte | T-pericyte | Human snRNA-seq |
| Extracellular matrix pericyte | M-pericyte | Human snRNA-seq |
| Smooth muscle cell | SMC | Human snRNA-seq |
| Perivascular fibroblast | P. fibro. | Human snRNA-seq |
| Meningeal fibroblast | M. fibro. | Human snRNA-seq |

### Regional and cell-type-specific gene expression changes

To identify disease-associated gene expression changes, we performed differential expression (DE) analysis in each disease group compared to controls for our human ST and snRNA-seq datasets (Supplementary Figs. 7–12 and Supplementary Tables 7–12). Trisomy 21 suggests overexpression of chr21 genes in our DSAD samples, so we examined differentially expressed genes (DEGs) by chromosome. We found chr21 gene overexpression was dependent on region or cell type (adjusted $P < 0.05$; Fig. 2a,b, Extended Data Fig. 1 and Supplementary Fig. 13). For example, *APP* is upregulated in DSAD samples but interestingly is not significantly different from control samples in spatial cluster L3/L4. Our analysis identified substantial downregulation of genes in cluster L3/L4 across diagnoses, which may reflect preferential pathology accumulation in L3 (refs. 29–31). Layers L3/L4 are central to cognitive processes and known for their dense synaptic connections; these DEGs may reflect the molecular responses to pathological changes impacting cognition. Deconvolution of the spatial DEGs using the snRNA-seq dataset showed that many genes upregulated across all the spatial regions were from glial and vascular cells (Figs. 2c,d and Supplementary Note; Methods).

We also found high correlations ($R \geq 0.5$) of DE effect sizes between diagnoses in most spatial clusters and modest correlations ($R \geq 0.2$) in most snRNA-seq clusters (Fig. 2e,f and Supplementary Figs. 14–16). These trends were stronger in GM clusters compared to WM clusters, consistent with the snRNA-seq data, where the correlations were stronger in neuronal versus glial clusters. Gene ontology (GO) term enrichment of the shared DEGs revealed region-specific enrichment of AD-relevant biological pathways, such as upregulation of genes related to long-term potentiation in L3–L5 and downregulation of those related to amyloid fibril formation in L3/L4 and L3–L5 (Fig. 2g,h and Supplementary Table 13). Similarly, we compared human ST DE effect sizes to 5xFAD versus WT DEGs, revealing the 5xFAD model recapitulates some clinical AD changes (Supplementary Figs. 17–22, Supplementary Tables 14–17 and Supplementary Note).

### System-level analysis of spatial gene expression programs

We performed high-dimensional weighted gene co-expression network analysis (hdWGCNA)[32] in our ST dataset within each cortical layer cluster and WM, yielding 166 gene modules from seven networks (Fig. 3a). Hierarchical clustering of these modules defined 15 cortex-wide 'meta-modules' based on similarity in expression patterns (module eigengenes (MEs)) and their constituent gene sets (Fig. 3a, Extended Data Fig. 2, Supplementary Fig. 23 and Supplementary Table 18; Methods). We interrogated system-level differences between disease groups and controls with differential ME (DME) analysis (Fig. 3a, circular heatmap, and Supplementary Table 19; Methods). Comparing DME effect sizes across diagnoses, we found that many DSAD changes reflect those in sAD, as well as modules uniquely regulated in each group (Fig. 3b,c and Extended Data Fig. 3). Early-stage AD had several modules exclusively downregulated in the WM and L3/L4 networks (Fig. 3c), and pathway enrichment associated these modules with neurotransmission, neurodevelopment and amyloid-β formation (Extended Data Fig. 3). Alternatively, most modules specifically upregulated in late-stage AD originated from the L6b network, while modules specifically upregulated in DSAD largely came from the L1 and L3/L4 networks.

We next sought to compare disease subtypes in the broader 15 meta-modules. These meta-modules were enriched for genes involved in myelination (M1) and chemical synaptic transmission (M3, M4, M7, M10 and M13), as well as previously implicated processes like glutamate signaling (M6), inflammatory response (M11) and amyloid fibril formation (M14; Fig. 3d and Supplementary Table 20). DME analysis of these meta-modules revealed that M6, containing hub genes such as *APP*, *SCN2A* and *CPE*, is upregulated in L1 in all diagnoses (Fig. 3e). While L1 is less densely populated with neurons than other cortical layers, M6 is expressed primarily in neurons, indicating that processes like APP metabolism, macroautophagy and RNA splicing are altered in AD L1 neurons. Alternatively, M11 is expressed in non-neuronal cells and upregulated across cortical upper-layer clusters. M11 contains genes associated with immune response and neuronal death, and its

hub genes include complement pathway genes (*C1QB* and *C3*) and disease-associated astrocyte genes[7,19] (*SERPINA3*, *VIM* and *CD44*). We inspected these modules in the mouse ST dataset, revealing that M1, M6 and M11 expression levels were correlated with age only in 5xFAD mice, therefore representing changes associated with disease progression and amyloid accumulation (Fig. 3f and Supplementary Fig. 24). Module preservation analysis[33] showed that almost all meta-modules were preserved in the mouse dataset (Fig. 3g, Z-summary preservation > 2), although M1, M6 and M11 were moderately to weakly preserved (Z-summary preservation < 10).

### Inflammatory signature correlated with AD genetic risk

We also performed genetic enrichment analyses with single-cell disease relevance scores (scDRS)[34] to investigate cellular and regional enrichment for AD risk genes and to identify links between disease risk signatures with co-expression modules (Extended Data Fig. 4 and Supplementary Figs. 25–29; Methods). We found significant associations with AD genetic risk in clusters L1, L3/L4 and WM in DSAD (Monte Carlo test FDR ≤ 0.05; Fig. 3h). ST spots contain multiple cells, and previous studies show enrichment of AD genetic risk exclusively in myeloid cells[7]. The signal may be obscured by neuronal and oligodendrocytic signatures in sAD samples. 5xFAD mice displayed increasing AD risk scores with age, with significant region-level associations at 12 months for several regions, including deep cortical layers and WM (Fig. 3i). Furthermore, we found significant cluster-level associations for microglia clusters MG1 and MG2 across all four snRNA-seq datasets (Fig. 3j).

We next correlated meta-module expression and AD risk scores in each cluster with a significant scDRS group-level association (Fig. 3k and Extended Data Fig. 5). While these correlations were modest, in the human ST dataset, we found the strongest correlations in the L3/L4 cluster with M13, M10, M7 and M11 in DSAD (Pearson $R \geq 0.2$). Furthermore, we found that M11 was correlated with AD genetic risk in 5xFAD mice, increasing in strength with age. In the snRNA-seq datasets, M11 was highly expressed in microglia and correlated with AD genetic risk across all datasets and disease subtypes, with a stronger correlation in activated microglia cluster MG2 (mean Pearson $R = 0.342$) compared to MG1 (mean Pearson $R = 0.284$). Finally, several meta-module member genes are associated with AD genetic risk through genome-wide association studies (GWAS)[35–37], such as *BIN1* in M1, *APP* and *APOE* in M6 and clusterin (*CLU*) and *ADAMTS1* in M11.

### Transcriptomic sex differences among subtypes of AD

Previous studies have described sex differences in AD clinical manifestations, risk factors and gene expression[8,38–43]. Here we performed DE analysis to investigate sex differences between DSAD and sAD using our ST dataset (Supplementary Tables 21–23; Methods). We found transcriptome-wide differences, with broad upregulation of genes in females compared to males across the spatial clusters and on all chromosomes (Fig. 4a–c and Supplementary Fig. 30). Deconvolution

---

**Fig. 2 | Altered gene expression signatures among subtypes of AD.**
**a**, Heatmap colored by effect size from the DSAD versus control differential gene expression analysis, with genes stratified by chromosome and by spatial region. Statistically significant (FDR < 0.05) genes with an absolute average $\log_2(FC) \geq 0.25$ in at least one region are shown. **b**, Stacked bar chart showing the number of DSAD versus control DEGs in each spatial region stratified by chromosome. **c**, Heatmap showing the gene expression values in the snRNA-seq dataset of spatial DEGs shared between DSAD and late-stage AD. **d**, Deconvolution of spatial DEGs using snRNA-seq cluster marker genes. Bar charts showing the number of DEGs up or down in disease for each spatial cluster are shown on the top and bottom, respectively. Proportional bar charts in the middle show the proportion of spatial DEGs that are also cluster marker genes in each of the snRNA-seq clusters. Spatial DEGs that are not in the snRNA-seq marker genes are shown in gray. **e**, Comparison of DE effect sizes from early-stage AD versus control and DSAD versus control. Genes that were

statistically significant (adjusted P < 0.05) in either comparison were included in this analysis. Genes are colored blue if the direction is consistent, yellow if inconsistent and gray if the absolute effect sizes are smaller than 0.05. Black line represents a linear regression with a 95% confidence interval around the mean shown in gray. Pearson correlation coefficients are shown in the upper left corner of each panel. **f**, Comparison of DE effect sizes from late-stage AD versus control and DSAD versus control, layout as in **e**. **g,h**, Selected pathway enrichment results from DEGs that were upregulated (**g**) or downregulated (**h**) in both late-stage AD and DSAD compared to controls. **i,j**, Selected pathway enrichment results from DEGs that were upregulated (**i**) or downregulated (**j**) in late-stage AD exclusively. **k,l**, Selected pathway enrichment results from DEGs that were upregulated (**k**) or downregulated (**l**) in DSAD exclusively. One-sided Fisher's exact test was used for enrichment analysis. VEGF, vascular endothelial growth factor; NMDA, N-methyl-D-aspartate; NK, natural killer.

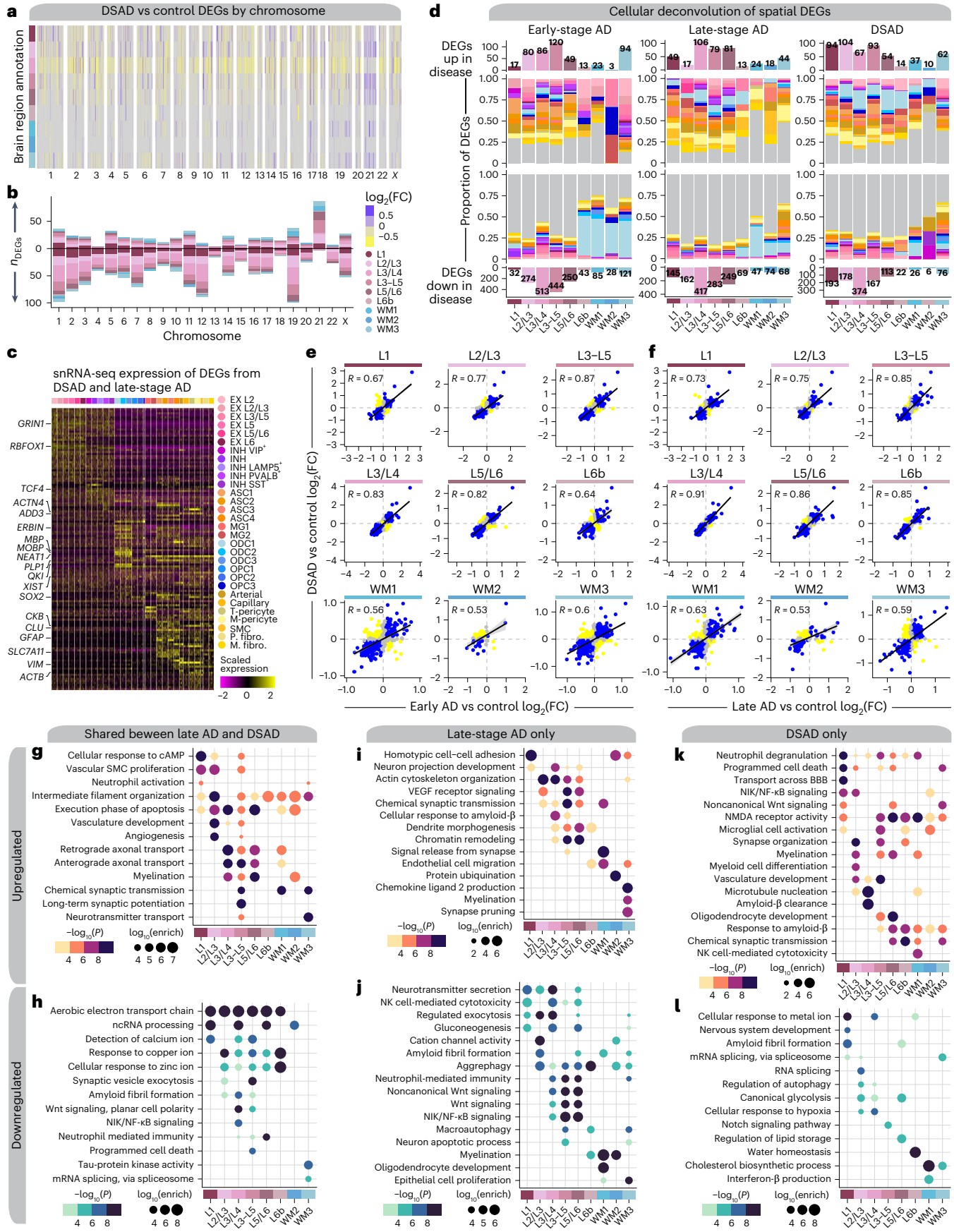

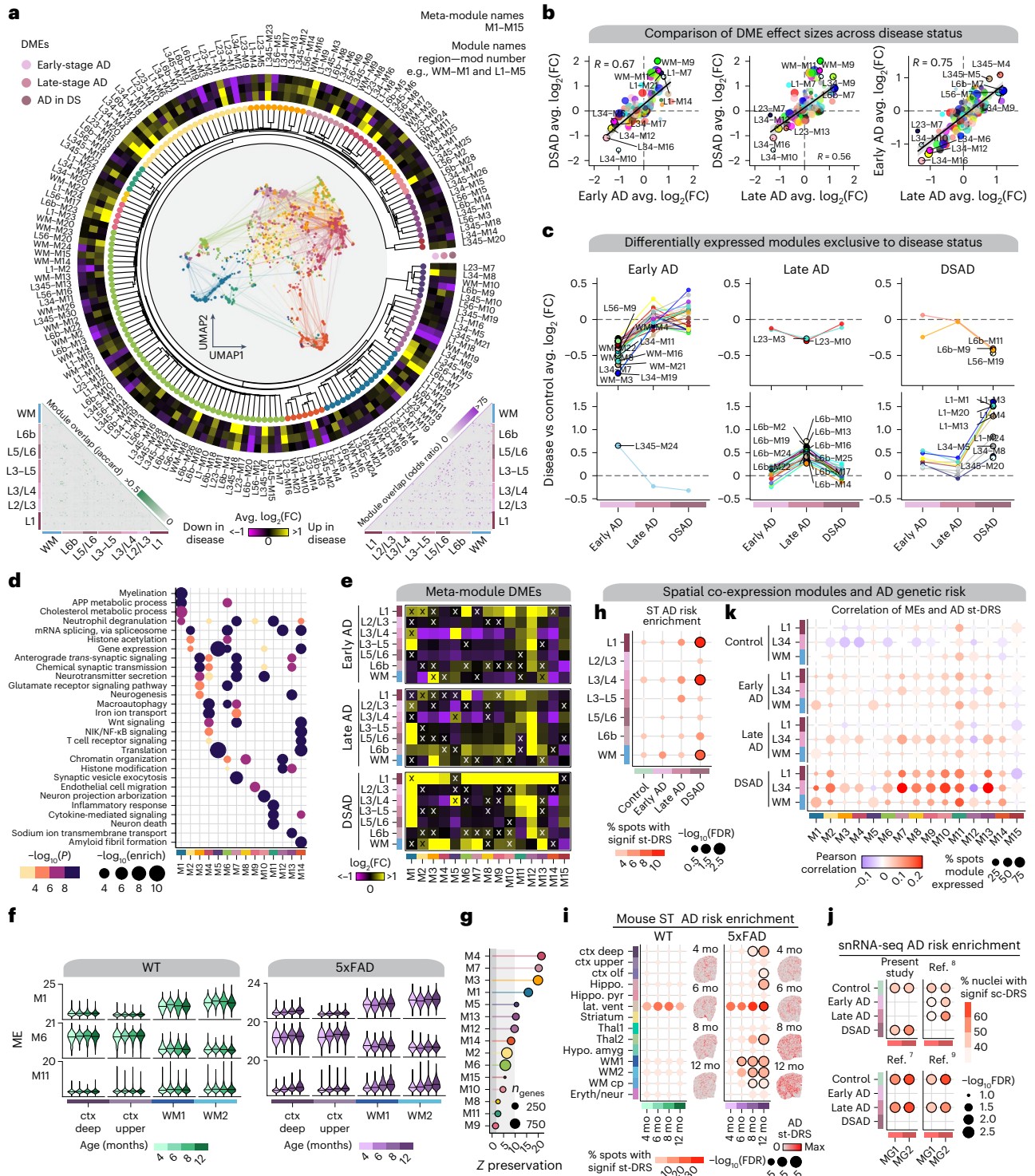

**Fig. 3 | System-level analysis of spatial gene expression programs.**
**a**, Dendrogram shows 166 co-expression modules grouped into 15 meta-modules. Network plot represents the consensus co-expression network; each dot is a gene colored by meta-module assignment. Heatmap shows effect sizes from DME testing. Triangular heatmaps show the distance of gene sets and expression between modules (left, Jaccard; right, odds ratio). **b**, Comparison of effect sizes from DME testing across disease groups. Black lines linear regression with a 95% confidence interval around the mean in gray. **c**, Lineplots showing differentially expressed modules specific to disease groups. Top, downregulated modules. Bottom, upregulated modules. **d**, Selected pathway enrichment results for each meta-module. One-sided Fisher's exact test was used for enrichment analysis. **e**, Heatmaps of meta-module DMEs in each disease group compared to controls. X indicates a lack of statistical significance. **f**, Violin plots showing MEs of selected

meta-modules in mouse ST, split by age, with black line indicating the median ME. **g**, Lollipop plot showing module preservation analysis of the meta-modules projected into the 5xFAD mouse dataset. Z-summary preservation > 10, highly preserved; 10 > Z-summary preservation > 2, moderately preserved and 2 > Z-summary preservation, not preserved. **h**, st-DRS for AD in human ST clusters. Black outlines on the dots denote a significant group-level association (Monte Carlo test false discovery rate (FDR) ≤ 0.05). **i**, st-DRS for AD computed in mouse ST clusters. Feature plots show the st-DRS scores for representative samples in WT and 5xFAD. **j**, Single-cell disease relevance scores for AD in microglia in the snRNA-seq dataset, split by dataset and disease status. **k**, Dot plots show the percentage of ST spots in each group as the size and the correlation of the MEs and the scDRS AD enrichment as the color. mo, month.

analysis showed that many genes upregulated in females were glial and vascular, while more of those upregulated in males were neuronal or oligodendrocytic (Fig. 4d and Supplementary Fig. 31). GO enrichment revealed that genes involved in inflammation, oxidative stress and glucose metabolism are upregulated in females independent of a brain region, whereas male DEGs are related to alternative splicing, chromatin organization and cytoskeletal organization and transport (Fig. 4e,f and Supplementary Table 24). We found that microglial activation was enriched specifically in L6b and WM in females, but amyloid-β-related processes were enriched in both female- and male-specific DEGs. This enrichment was restricted to L3–L5 in males, indicating regional specificity of transcriptomic sex differences related to amyloid processing. Gene set overlap analysis demonstrated that many of the DEGs with the largest effect sizes were shared across multiple regions (Fig. 4g). *C1QB* (M11) is upregulated in DSAD females in all the spatial regions, with the highest effect size in the WM (average $\log_2$(fold change (FC)) = 0.969), and we validated this at the protein level using immunofluorescence (Figs. 4h–j; two-sided $t$ test $P$ value = 0.037). We also performed sex-based DE analysis within the other diagnoses in the human and mouse ST datasets and the snRNA-seq dataset, revealing additional sex-specific signatures of disease (Supplementary Figs. 30, 32–45 and Supplementary Tables 25–31).

To investigate system-level sex-related transcriptomic changes in AD, we performed DME analysis between females and males in each disease group (Figs. 4k,l). In cluster L1, M11 was upregulated in males in early-stage AD but upregulated in females for late-stage AD and DSAD, revealing stronger neuroinflammatory signatures in females with high pathological load. Many of the top region-specific modules in sAD were from the L1 network, yet the direction of effect switched between early and late stages (Fig. 4l). For example, modules L1–M1 and L1–M8 were the top female modules in late-stage AD and among the top male modules in early-stage AD, indicating key temporal differences in disease-related gene expression changes between sexes. In general, more modules were upregulated in late-stage AD females versus more in early-stage AD males. We also compared the DME effect sizes between the sex DMEs and the diagnosis DMEs (Fig. 4m), and found many modules not DE between control and disease showed sex differences.

## Imaging mass cytometry (IMC) reveals protein expression changes

We next used IMC for multiplexed imaging of 23 proteins, including cell-type markers, proteins encoded by genes of interest from our transcriptomic analyses and proteins of interest from an AD proteomic study[44] (Fig. 5a–c and Supplementary Table 1). We analyzed spatial protein expression patterns in FCX samples from our ST cohort (post-QC: $n = 2$ cognitively normal controls, $n = 6$ late-stage AD and $n = 6$ DSAD), and clustering revealed 11 populations (Fig. 5d; Methods). We recovered proteomic profiles for astrocytes (GFAP+ and S100b+), neurons (microtubule-associated protein 2 (Map2+) and NeuN+), microglia

(Iba1+) and populations enriched in extracellular matrix proteins or phospho-tau (Figs. 5e,f).

We compared protein abundances between disease groups in these cell populations (Figs. 5g–l and Supplementary Table 32; Methods). Amyloid-β and phospho-tau significantly increased in AD neurons, as expected, but Map2 decreased, indicative of neurodegeneration (Fig. 5g). DSAD neurons demonstrated elevated phospho-tau burden compared to late-stage AD, and one astrocyte cluster was marked by high phospho-tau, likely engulfing these phospho-tau-bearing neurons and indicating increased tau pathology in DSAD (Fig. 5j). Cystatin C (*CST3*), known to colocalize with amyloid, was significantly changed in microglia and astrocytes from late-stage AD and DSAD (Fig. 5i). In our ST data, *CST3* was upregulated in both groups in spatial clusters L3–L5 and L5/L6. We also found a significant upregulation of *CLU* in both microglia and astrocyte subpopulations with increased CD44 expression in both late-stage AD and DSAD (Figs. 5j,k). *CD44* and *CLU* are hub genes of meta-module M11, which we found upregulated in both 5xFAD and clinical AD samples, demonstrating coregulation at both gene and protein levels.

## Integrated analysis reveals dysregulated cell signaling

We analyzed cell–cell communication (CCC) in disease by predicting spatial coordinates of each snRNA-seq cell with CellTrek[45] (Extended Data Fig. 6, Supplementary Figs. 46 and 47 and Supplementary Note) and inferring CCC using CellChat[46] after stratifying snRNA-seq populations by upper cortical, lower cortical or WM (Fig. 6a; Methods). This revealed changes in the CCC landscape, illuminating dysregulated pathways based on relative information flow between DSAD and controls (Fig. 6b, Extended Data Fig. 7, Supplementary Figs 48–50 and Supplementary Table 33). We focused on the following three signaling pathways: NECTIN, ANGPTL and CD99. NECTIN signaling is downregulated in DSAD (Figs. 6c–f). Nectins are involved in synapse maintenance[47–49], and *NECTIN2* has been implicated in AD genetic risk[50,51]. We report diminished neuronal NECTIN signaling (Figs. 6c,d) and, using immunofluorescence, Nectin2 downregulation in neurons from late-stage AD and DSAD (Fig. 6g,h; two-way $t$ test, $P < 0.05$). In control samples, ANGPTL signaling features astrocyte clusters in lower cortical layers and WM (ASC1 and ASC3, respectively) communicating with neurons, pericytes and oligodendrocyte precursor cells (OPCs) with the ligand *ANGPTL4* (Figs. 6i–l). However, in DSAD, additional astrocytes, such as ASC1 and ASC3 in the cortical upper layers, showed significant interactions mediated by *ANGPTL4*. Increased ANGPTL4 expression has been previously observed in astrocytes from patients with AD with vascular changes[52], and co-immunofluorescence of ANGPTL4 and GFAP confirmed astrocytic ANGPTL4 upregulation in DSAD (two-way $t$ test, $P < 0.05$; Fig. 6m,n). We note a loss of astrocyte-inhibitory neuron ANGPTL communication with the disease, and *ANGPTL4* is a hub gene of meta-module M11. Furthermore, *CD99* is also a hub gene of M11, but CD99 signaling is downregulated in DSAD, confirmed by immunofluorescence (Extended Data Fig. 8).

---

**Fig. 4 | Sex-related transcriptomic differences in subtypes of AD. a**, Effect sizes from DSAD female versus male differential gene expression, genes stratified by chromosome and spatial region. Significant (FDR < 0.05) genes with absolute average $\log_2$(FC) ≥ 0.25 in at least one region are shown. **b**, Stacked bar chart showing the number of DSAD female versus male DEGs by spatial region stratified by chromosome. **c**, Volcano plots showing the effect size and significance level from the DSAD female versus male DE (MAST[59], two-sided test). **d**, Number of DEGs upregulated in females or upregulated in males for each cluster on top and bottom, respectively. Proportional bar charts show the proportion of DEGs that are snRNA-seq marker genes. **e,f**, Selected pathway enrichment analysis from DEGs that were upregulated in females (**e**) or males (**f**). One-sided Fisher's exact test was used for enrichment analysis. **g**, Overlap between sets of DEGs in spatial clusters. **h**, Representative images from the prefrontal cortex (PFC) of age-matched male and female patients with DSAD stained for C1QB (red). Dashed

line visually separates GM and WM. **i**, Bar graph representing mean fluorescence intensity (relative) in ×20 images of C1QB ($n = 5$ brain sections from $n = 3$ female DSAD cases and $n = 5$ brain sections from $n = 3$ male cases) in the WM. $P$ value from the two-sided $t$ test is shown. Error bar shows one s.d. from the mean. **j**, Top, spatial feature plots showing *C1QB* expression in representative female (left) and male (right) DSAD samples. Bottom, samples colored by region annotations. **k**, Heatmap showing the meta-module DSAD female versus male DME results for each cortical layer and WM. X indicates a lack of significance. **l**, Volcano plot showing the effect size and significance level from DSAD female versus male DME analysis (two-sided Wilcoxon rank-sum test). **m**, Comparison of DME effect sizes between sex and diagnosis tests. Black line represents a linear regression with a 95% confidence interval around the mean shown in gray. Pearson correlation coefficients are shown in the upper left corner of each panel. Number of modules significant in either analysis in each quadrant is noted. F, female; M, male.

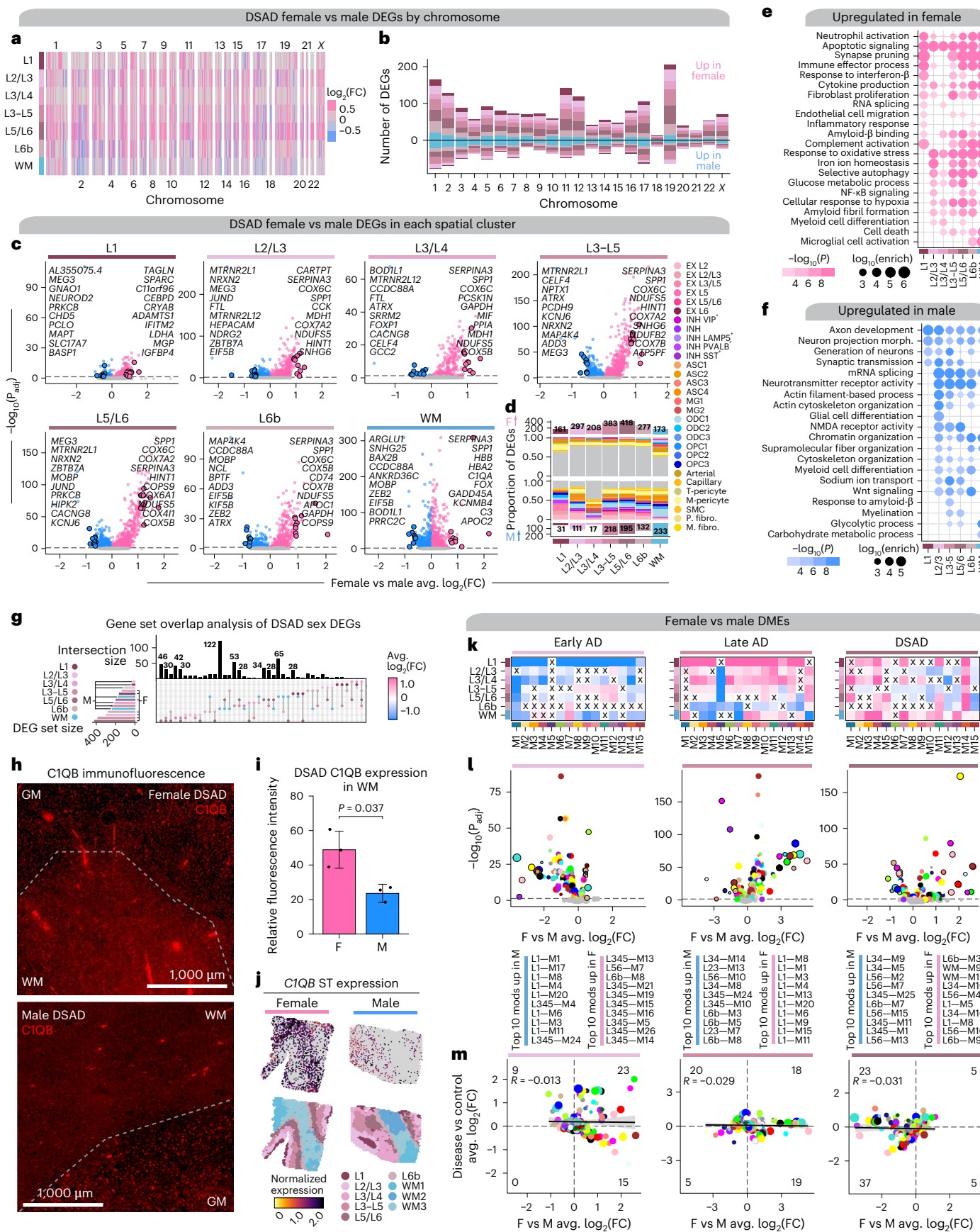

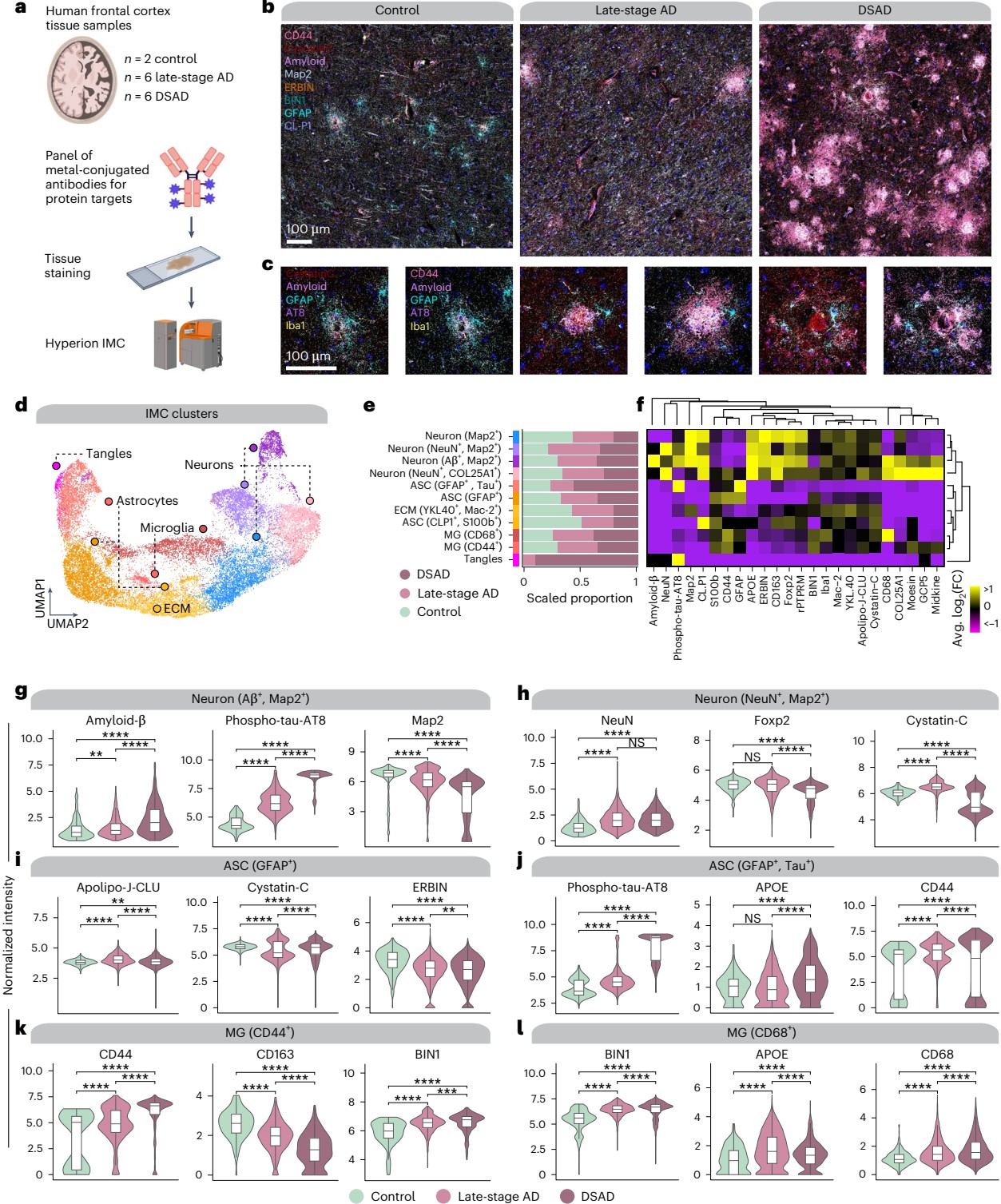

**Fig. 5 | IMC reveals single-cell spatial proteomic changes in AD. a**, IMC was performed in postmortem human cortical tissue (*n* = 2 control, *n* = 6 late-stage AD and *n* = 6 DSAD) using the Hyperion Imaging System (Standard BioTools). Illustrations were created with Biorender.com. **b**, Representative IMC images from control, late-stage AD and DSAD samples with select targets from the panel. **c**, Images as in **b** at higher magnification and focused around amyloid plaques. **d**, UMAP plot showing the unbiased clustering of segmented nuclei from the IMC dataset based on their protein intensity values. Each dot represents a segmented nucleus, colored by cluster assignment. **e**, Stacked bar plots showing the proportion of segmented nuclei assigned to each cluster stratified by disease groups. **f**, Heatmap showing the relative protein intensity of each protein in

each IMC cluster. Dendrograms depict hierarchical clustering results based on these relative intensities. **g**–**l**, Violin plots showing the distribution of protein intensities for selected proteins in IMC clusters Neuron (Aβ+, MAP2+) (**g**), Neuron (NeuN+, MAP2+) (**h**), ASC (GFAP+) (**i**), ASC (GFAP+, Tau+) (**j**), MG (CD44+) (**k**), MG (CD68+) (**l**), stratified by disease groups. For box and whisker plots, box boundaries and lines correspond to the IQR and median, respectively. Whiskers extend to the lowest or highest data points that are no further than 1.5 times the IQR from the box boundaries. Two-sided Wilcoxon rank-sum test results are overlaid on each plot. NS, *P* > 0.05; **\**P* ≤ 0.01; \*\*\**P* ≤ 0.001; \*\*\*\**P* ≤ 0.0001. IQR, interquartile range; NS, not significant; ECM, extracellular matrix.

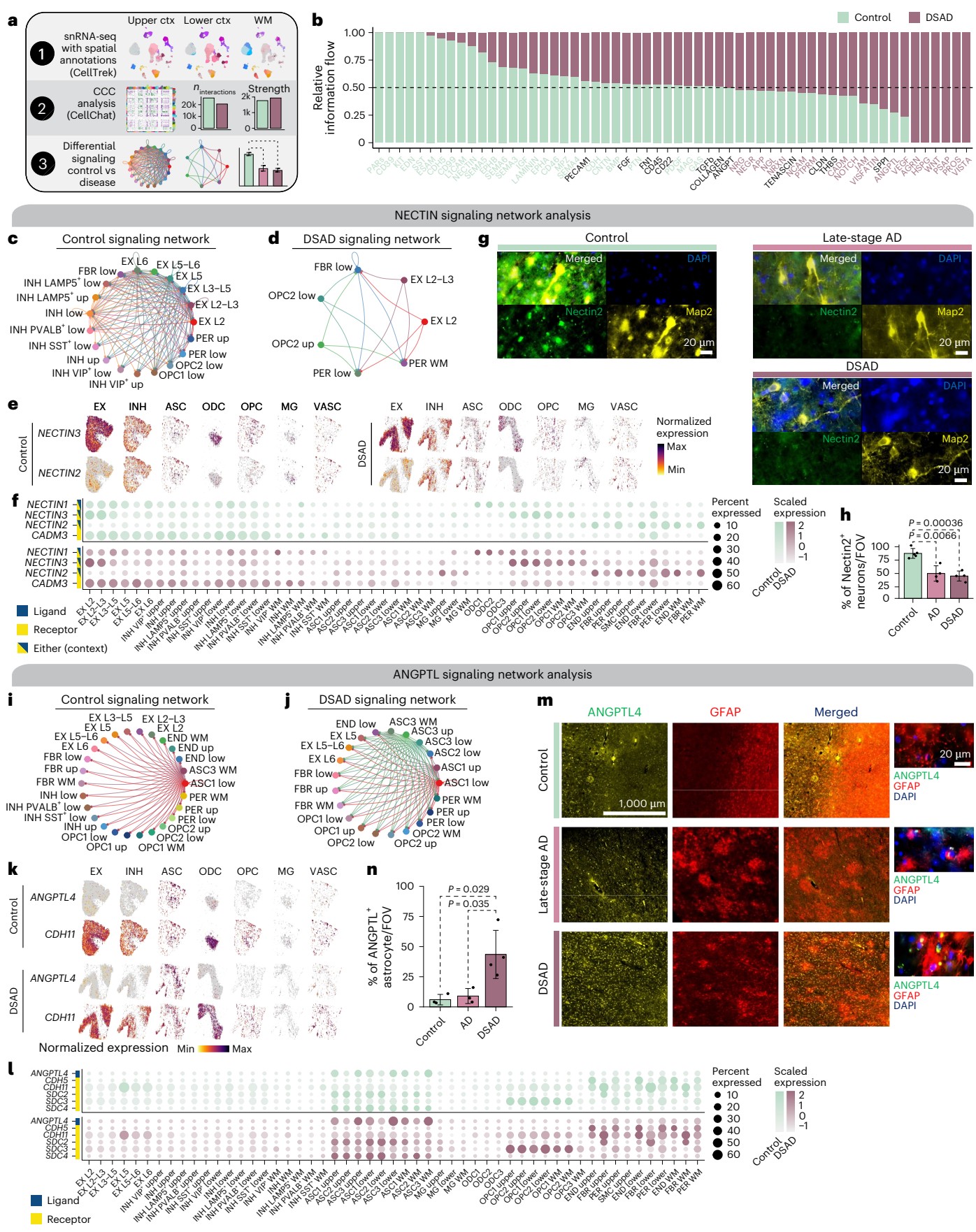

**Fig. 6 | Altered cell–cell communication signaling networks in DSAD.**
**a**, Schematic representation of CCC analysis. **b**, Bar plot showing signaling pathways with significant differences between DSAD and controls, ranked based on their information flow (sum of communication probability among all pairs of cell populations in the network). **c**,**d**, Network plot showing the CCC signaling strength between different cell populations in controls (**c**) and DSAD (**d**) for NECTIN signaling. **e**, Spatial feature plots of the snRNA-seq in predicted spatial coordinates for one control sample (left) and one DSAD sample (right) for select NECTIN signaling genes. **f**, Dot plot showing gene expression of NECTIN signaling genes with significant CCC interactions in control (top) and DSAD (bottom). **g**, Representative double immunofluorescence images for Nectin2 (green), Map2 (yellow) and DAPI (blue) from postmortem human brain tissue (PFC) of control, late-stage AD and DSAD cases. **h**, Bar plot representing results of NECTIN colocalization analysis from ×60 images (n = 5 cognitively healthy control, n = 4 late-stage AD and n = 4 DSAD cases) using the JACoP Plugin

from ImageJ and Manders' correlation coefficient. Data are presented as the average of three different fields of view (FOVs) per sample. P values from two-way t tests are shown. Error bar shows one s.d. from the mean. **i**,**j**, Network plot as in **c** and **d** in control (**i**) and DSAD (**j**) for ANGPTL signaling. **k**, Spatial feature plots as in **e** for one control sample (top) and one DSAD sample (bottom) for the ANGPTL pathway. **l**, Dot plot showing gene expression of ANGPTL signaling genes with significant CCC interactions in control (top) and DSAD (bottom). **m**, Representative double immunofluorescence images at ×10 and ×60 magnification for ANGPTL4 (green), GFAP (red) and DAPI (blue) from postmortem human brain tissue (PFC) of control, late-stage AD and DSAD cases. **n**, Bar plot representing results of ANGPTL colocalization analysis from ×60 images (n = 3 cognitively healthy control, n = 3 late-stage AD and n = 4 DSAD cases) using the JACoP Plugin from ImageJ and Manders' correlation coefficient. Data are presented as the average of three different FOVs per sample. P values from two-way t tests are shown. Error bar shows 1 s.d. from the mean.

## Conformation-specific amyloid gene expression signatures

Deep characterization of amyloid pathology and associated molecular changes is critical to understanding AD pathogenesis. Therefore, we stained human and mouse ST tissues with Amylo-Glo for dense amyloid-β plaques and the Anti-Amyloid Fibrils OC Antibody (OC) for diffuse amyloid fibrils (Fig. 7a,b). In the human dataset, we found that amyloid pathology distribution was consistent with neuropathological plaque staging (Fig. 7c), and we observed increasing amyloid deposition in 5xFAD with age across cortical and subcortical regions (Supplementary Fig. 51). By integrating amyloid imaging and transcriptomic data, we identified amyloid hotspots and 65 plaque-associated and 215 fibril-associated genes proximal to these hotspots in the human dataset (Fig. 7d,f, Supplementary Figs. 52–55 and Supplementary Table 34). Genes associated with plaques and fibrils included diagnosis DEGs and M11 hub genes, *CLU* and *VIM*, and are enriched in processes including intermediate filament assembly, long-term potentiation and blood–brain barrier transport (Fisher's exact test $P = 1.8 \times 10^{-115}$, odds ratio = 1248.92; Fig. 7g,h and Supplementary Table 35). Fibril-specific genes are related to synaptic function and hemopoiesis (Fig. 7i).

In the mouse dataset, we identified amyloid-associated genes within each spatial cluster, revealing 1,829 plaque-associated and 1,759 fibril-associated genes, with the largest overlaps in the hippocampus pyramidal and upper cortical layer clusters (Fig. 7f and Supplementary Table 36). GO enrichment linked these gene sets to inflammatory and neurodegenerative processes (Fig. 7j and Supplementary Table 37), and they overlapped with previously reported gene signatures identified in AD mouse models[18,19,53,54]. Fibrillar-specific genes included *Itgb2*, *Cd53* and *Il33*, suggesting unique immune signatures preceding plaque formation. We found modest yet significant overlaps between human and mouse amyloid-associated genes, particularly among plaque-associated, and microglial genes were more common in mice (Fig. 7k). Shared fibrillar-associated genes in the cortex include *NEFH*, *NEFM*, *ALDOC* and *MAFB*. Notably, meta-module M11's hub genes *VIM* and *CLU* were among the shared amyloid-associated genes, and M11's

expression weakly correlated with amyloid hotspots (max Pearson correlation $R = 0.131$ in L5/L6 early-stage AD, $R = 0.187$ in ctx deep layers in 6 month 5xFAD), implicating it in cross-species amyloid-associated processes (Supplementary Fig. 56). Furthermore, co-expression network analysis in the 5xFAD ST dataset uncovered an amyloid-associated gene module (SM6), which shared similar genes and expression patterns with a previously identified set of 'plaque-induced' genes[53] (Figs. 7l,m, Extended Data Figs. 9 and 10, Supplementary Fig. 57–59 and Supplementary Table 38).

## Discussion

Spatiotemporal pathological progression coupled with cellular dysregulation are focal points of AD. Our ST and single-nucleus transcriptomic analysis of DSAD, sAD and 5xFAD mice reveals insights into key disease processes (Fig. 7n). We identified regional DEGs shared between sAD and DSAD, contextualizing their shared genetic, clinical and biomarker features[16,55,56]. Remarkably, we identified changes in L3/L4 across all diagnoses, coinciding with L3-preferential amyloid deposition in AD; these changes may underlie this regional vulnerability. We identified sex-related differences in DSAD, revealing increased expression of inflammatory genes within females and regulatory genes in males. However, the number of replicates used for these sex comparisons is a limitation of this study. DS samples without AD also would be valuable controls but are extremely rare due to the high prevalence of AD in the aged DS population[57] and not included in this study.

Integrated system-level analyses of ST and snRNA-seq enhanced our understanding of the AD transcriptome among subtypes. CCC analysis identified signaling pathway changes. Dysregulated ANGPTL and CD99 signaling highlight astrocyte modulation of brain vascular integrity in AD and pinpoint downstream targets of astrocyte phenotype changes. Multiscale network analysis[32] identified a diverse array of gene modules, exposing AD spatiotemporal gene expression patterns. We found significant downregulation of L3/L4 and WM modules

**Fig. 7 | Amyloid-associated gene expression signatures. a**,**b**, Representative fluorescent images from DSAD (**a**) and 12-month 5xFAD (**b**) stained with Amylo-Glo and OC to mark dense amyloid plaques and diffuse amyloid fibrils, respectively. ST data colored by cluster, amyloid quantification and hotspot analysis are below the images. **c**, Box and whisker plots showing the distribution of amyloid quantifications in the human ST dataset, stratifying samples by neuropathological plaque staging. Box boundaries and lines correspond to the IQR and median, respectively. Whiskers extend to the lowest or highest data points that are no further than 1.5 times the IQR from the box boundaries. Number of samples per stage—none, n = 8; stage A, n = 3; stage B, n = 7 and stage C, n = 14. Two-sided Wilcoxon test was used for pairwise comparisons. **d**,**e**, Amyloid hotspot results (Getis-Ord Gi*) for human (**d**) and mouse (**e**). **f**, Number of amyloid-associated genes from Amylo-Glo, shared and OC for mouse clusters. Euler diagram shows the overlap of Amylo-Glo- and OC-associated genes in the

human dataset. **g**, Number of Amylo-Glo- and OC-associated genes that overlap with disease DEGs. **h**,**i**, Selected pathway enrichment results from amyloid-associated genes that were shared between Amylo-Glo and OC (**h**) and OC-specific (**i**) in the human ST dataset. **j**, Selected pathway enrichment results from amyloid-associated genes shared between Amylo-Glo and OC in the mouse ST dataset. One-sided Fisher's exact test was used for enrichment tests. **k**, Heatmap showing gene set overlap results of mouse and human amyloid-associated genes, as well as with other gene sets (DAA[19], DAM[18], DOL[54] and PIGs[53]). NS, P > 0.05; *P ≤ 0.05; **P ≤ 0.01; ***P ≤ 0.001; ****P ≤ 0.0001. **l**, Expression of SM6 and PIGs modules in representative mouse ST samples. **m**, Euler plot showing overlap of the SM6 and PIGs[53]. **n**, Overview of the experiments, data analysis and selected conclusions of this entire study. Illustrations were created with Biorender.com. DAA, disease-associated astrocytes; DAM, disease-associated microglia; DOL, disease-associated oligodendrocytes; BBB, blood–brain barrier.

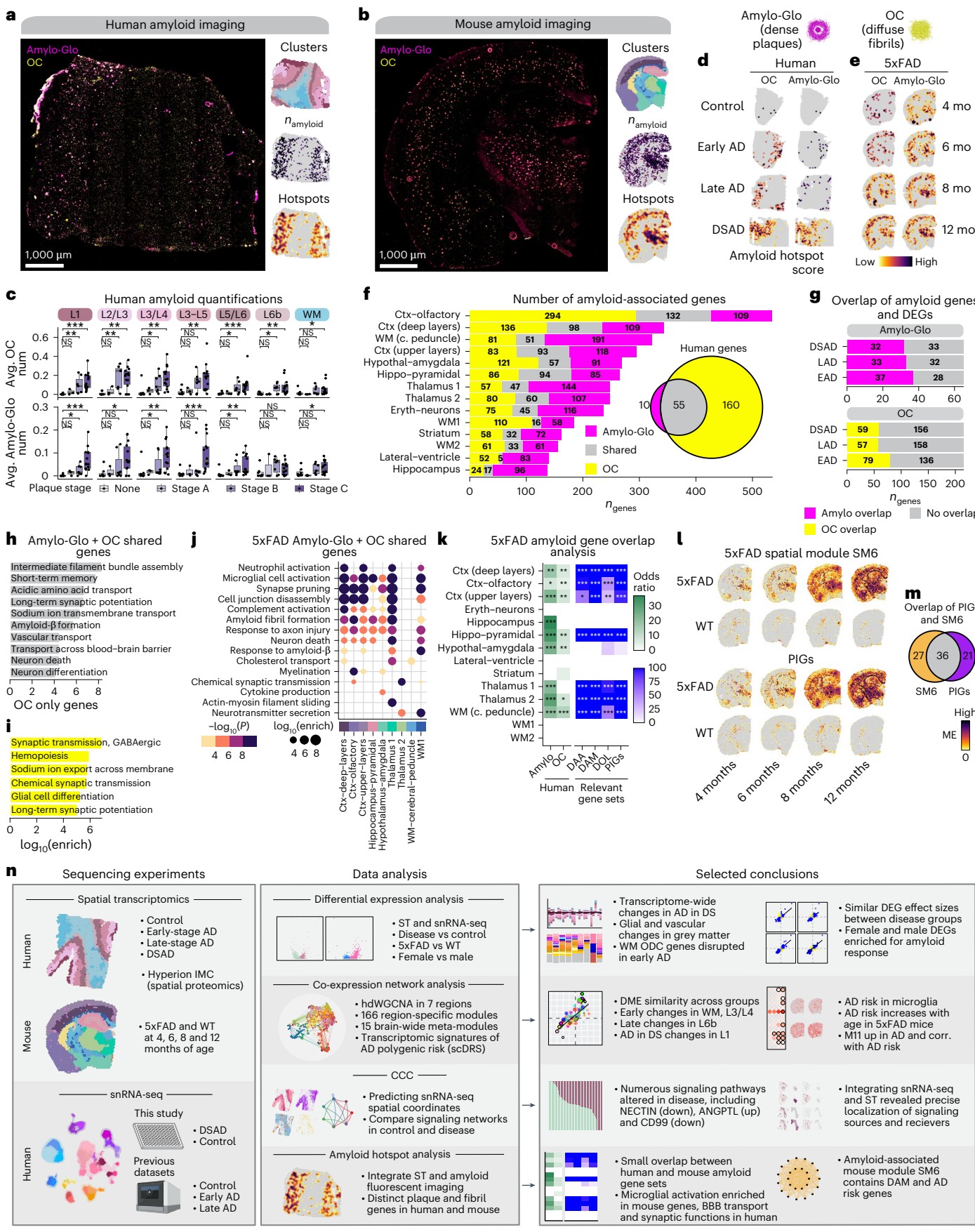

specifically in early-stage AD, indicating dynamic shifts in gene expression that may have a pivotal role in AD progression. Glial meta-module M11 is upregulated in the cortical upper layers and contains DEGs shared between sAD and DSAD. We also found evidence of an association between AD genetic risk and M11 by comparing risk scores with M11 across data modalities and species.

We integrated amyloid imaging and ST to identify transcriptomic signatures proximal to dense amyloid plaques and diffuse fibrils, yielding cross-species amyloid-associated gene sets. We acknowledge that the present resolution of ST (55 μm) may have limited our findings—sample differences between the human and mouse datasets likely also contributed to the differences in these gene sets. However, we identified M11 hub genes in the amyloid-associated genes, observing M11 expression at regions with amyloid deposition in mice and humans, altogether emphasizing the critical role of M11's associated biological processes and genes in AD pathophysiology.

This study offers an exploration into the spatiotemporal dynamics of AD gene expression using ST and snRNA-seq. Our analysis highlights specific cell types in AD, describes disease-associated changes in human cortical niches and mouse brain regions, identifies key networks coordinating spatiotemporal and cellular changes and links these transcriptomic signatures to AD genetic risk and amyloid pathology accumulation. Together, this work adds new dimensions to our understanding of AD, emphasizing the dynamic shifts in gene expression and the critical involvement of both neuronal and glial components in disease progression, and highlights the importance of studying AD subtypes like DSAD.

## Online content

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

[1]Department of Neurobiology and Behavior, University of California, Irvine, Irvine, CA, USA. [2]Institute for Memory Impairments and Neurological Disorders (MIND), University of California, Irvine, Irvine, CA, USA. [3]Mathematical, Computational, and Systems Biology (MCSB) Program, University of California, Irvine, Irvine, CA, USA. [4]Center for Complex Biological Systems (CCBS), University of California, Irvine, Irvine, CA, USA. [5]Sue and Bill Gross Stem Cell Research Center, University of California, Irvine, Irvine, CA, USA. [6]Department of Developmental and Cell Biology, University of California, Irvine, Irvine, CA, USA. [7]Department of Computer Science, University of California, Irvine, Irvine, CA, USA. [8]Department of Pediatrics, University of California, Irvine, School of Medicine, Orange, CA, USA. [9]Department of Pathology and Laboratory Medicine, University of California, Irvine, Irvine, CA, USA. [10]These authors contributed equally: Emily Miyoshi, Samuel Morabito. *A list of authors and their affiliations appears at the end of the paper. ✉e-mail: vswarup@uci.edu

## Alzheimer's Biomarkers Consortium–Down Syndrome (ABC–DS)

### Eric Doran[8], Elizabeth Head[2,9] & Ira T. Lott[2,8]

A full list of members and their affiliations appears in the Supplementary Information.

## Methods

### Postmortem human brain tissue

Human brain tissue from the prefrontal cortex and posterior cingulate cortex (PCC) was obtained from the University of California, Irvine's (UCI) Alzheimer's Disease Research Center and the National Institutes of Health NeuroBioBank under UCI's Institutional Review Board (IRB). Additional postmortem human brain tissue originating from other studies[8,9] was obtained under the IRB of Rush University Medical Center. Informed consent was obtained from all human participants. Samples were assigned to groups based on both neurofibrillary tangle and plaque staging, in addition to clinical diagnoses. Samples were also selected based on several covariates, including age, sex, race, postmortem interval (PMI), RNA integrity number (RIN) and disease comorbidity. RIN values were obtained by isolating total RNA with the Zymo Direct-zol RNA Isolation Kit and running on the Agilent TapeStation 4200. Sample information is given in Supplementary Table 1.

### Mouse brain tissue

All mouse work was approved by the Institutional Animal Care and Use Committee at UCI. 5xFAD hemizygous (C57BL/6J; Jackson Laboratory, 034848) and WT (Jackson Laboratory, 000664) littermates were bred and housed until killed at 4, 6, 8 and 12 months. Sample information is given in Supplementary Table 2. For genotyping, we used the following primers (for *PSEN1*): 5′-AAT AGA GAA CGG CAG GAG CA-3′ (forward) and 5′-GCC ATG AGG GCA CTA ATC AT-3′ (reverse). Mice were killed by carbon dioxide inhalation. After PBS transcardiac perfusion, one brain hemisphere was flash-frozen in isopentane chilled with dry ice for RNA analyses, while the other hemisphere was fixed in 4% paraformaldehyde for immunohistochemistry.

### snRNA-seq

In total, 50 mg of fresh-frozen postmortem human brain tissue was homogenized in Nuclei EZ Lysis buffer (NUC101-1KT; Sigma-Aldrich) and incubated for 5 min before being passed through a 70 μm filter. Samples were then centrifuged at 500$g$ for 5 min at 4 °C and resuspended in additional lysis buffer for 5 min. After another centrifugation at 500$g$ for 5 min at 4 °C, samples were incubated in nuclei wash and resuspension buffer (NWR; 1× PBS, 1% BSA and 0.2 U μl$^{-1}$ RNase inhibitor) for 5 min. To remove myelin contaminants, we prepared sucrose gradients with nuclei PURE sucrose buffer and nuclei PURE 2M sucrose cushion solution from the Nuclei PURE Nuclei Isolation Kit (NUC-201; Sigma-Aldrich), and samples were carefully overlaid and centrifuged at 13,000$g$ for 45 min at 4 °C. Samples were then washed in NWR before processing with the Nuclei Fixation Kit (Parse Biosciences, SB1003). After nuclei fixation and permeabilization, samples were cryopreserved with DMSO until the day of library preparation. We generated single-nucleus libraries with the Whole Transcriptome Kit (WTK, Parse Biosciences, SB2001). cDNA library quantification and quality were assessed with a Qubit dsDNA HS Assay Kit (Invitrogen, Q32851) and D5000 HS Kit (Agilent, 5067-5592 and 5067-5593) or D1000 HS Kit (Agilent, 5067-5584 and 5067-5585) for the Agilent TapeStation 4200. Libraries were sequenced using the Illumina NovaSeq 6000 S4 platform using 100 bp paired-end sequencing for a sequencing depth of 50,000 read pairs per cell.

### ST

Fresh-frozen tissue samples were sectioned on an HM525NX cryostat (Thermo Fisher Scientific) at −15 °C for 10-μm thick sections that were immediately mounted onto 10x Genomics Visium slides. Slides were individually stored in slide mailers (sealed airtight in a plastic bag) at −80 °C until staining. We followed 10x Genomics Methanol Fixation, Immunofluorescence Staining and Imaging for Visium Spatial Protocols (Rev C), except after tissue sections were fixed in methanol and blocked, the sections were incubated with Amylo-Glo (Biosensis, TR-300; 1:100) for 20 min. Sections were then incubated with the primary antibody OC

(1:500 for mouse and 1:200 for human; polyclonal, Millipore, AB2286) and respective secondary antibody (1:400; goat anti-rabbit secondary antibody, Alexa Fluor 488 or Alexa Fluor 647, both Life Tech). Immediately after immunostaining, capture areas were imaged on a widefield Nikon Ti2-E microscope at ×20 magnification. ST libraries were then generated from the tissue sections according to the 10x Genomics Visium User Guide (Rev E). Library quantification, quality check and sequencing were performed as previously described, but sequencing depth was based on an estimated 60% tissue area coverage per sample for 50,000 read pairs per covered spot. The individual ST spots may contain one to ten cells per spot, as reported by 10x Genomics based on their Visium analysis of 10-μm thick mouse brain sections.

### IMC

Primary antibodies were formulated carrier-free except for YKL-40, which contained glycerol and was purified before metal conjugation with Amicon 10K Buffer Exchange Columns (EMD Millipore, UFC501096). All antibody concentrations were obtained using a Nanodrop 2000c Spectrophotometer and formulated with a final stock concentration of 0.5 mg ml$^{-1}$. All antibodies were conjugated using Standard BioTool's (SBT, formerly Fluidigm) Maxpar X8 metal conjugation protocol with Maxpar metal labeling kits (SBT, 201300).

Fixed and cryoprotected tissue was sectioned on an HM525NX cryostat (Thermo Fisher Scientific) at −15 °C for 14-μm thick sections onto Fisher Superfrost Plus slides. Slides were stored at −80 °C until staining and sealed airtight in a plastic bag. We followed the fresh-frozen staining protocol from SBT; however, because the tissue was previously fixed, we skipped the fixation step. Slides were transferred on dry ice to incubate at 37 °C for 5 min on a PCR machine, similar to the 10x Genomics Visium protocol. Sections were washed in PBS three times for 5 min before drawing a hydrophobic barrier. After the hydrophobic barrier dried, we incubated the sections with 3% BSA in PBS with 0.2% Triton X-100 for 45 min at room temperature. We then incubated the sections with the primary antibody cocktail diluted in 0.5% BSA/PBS with 0.2% Triton X-100 overnight at 4 °C. The antibodies and dilutions used in the primary antibody cocktail are given in Supplementary Table 29. Sections were then washed in PBS with 0.2% Triton X-100 twice for 8 min before incubating with the iridium intercalator (SBT, 201192A; 1:100 in PBS) for 30 min at room temperature. We then washed the sections in water twice for 5 min before allowing them to air dry before ablation. Hyperion Imaging System (SBT) was tuned before ablation using Hyperion Tuning Slide (SBT, 201088) for optimal instrument performance. Ablations were performed with ablation energy of four with a reference energy of zero in 1,000 × 1,000 μm regions of interest, except for one due to unexpected consumption of Argon gas that resulted in a 1,000 × 922 acquisition.

### Immunofluorescence

PFA-fixed human postmortem brain tissues were sectioned at 30 μm using a cryotome (Leica, SM2010R). Sections were then rehydrated (1× PBS) and permeabilized using sodium citrate buffer (heated at 95 °C for 10 min). After blocking with 3% BSA solution or serum (respective to the antibodies), sections were incubated with primary antibodies at 4 °C overnight (ANGPTL4 antibody (1:500; Thermo Fisher Scientific, 710186), GFAP polyclonal antibody (1:500; Thermo Fisher Scientific, PA3-16727), CD99 antibody (1:250; Thermo Fisher Scientific, MA5-12287), CD99L2 antibody (1:500; Thermo Fisher Scientific, PA5-58539), Nectin2 antibody (1:250; Thermo Fisher Scientific, PA582470), MAP2 antibody (1:250; Thermo Fisher Scientific, PA1-10005) and C1QB polyclonal antibody (1:250; Thermo Fisher Scientific, PA5-42554)). Secondary antibodies were selected according to the manufacturer's instructions and incubated for 2 h. Slides were imaged using Zeiss Axio Scan Z1 Slide scanner (for ×10 images) and Nikon ECLIPSE Ti2 inverted microscope (for ×20/×40/×60 images). Images from three randomly selected areas of each slice were used for analysis. For C1QB staining,

we tried to dissect WM-enriched regions from postmortem fixed brain samples and verified the presence of both WM and GM at a ratio of approximately 3:1 (for most male samples) and 2:1 (for most female samples) through visual inspection and subsequent confirmation with MOBP staining, ensuring accurate tissue characterization. To control for potential variations in background fluorescence between samples, we used ImageJ to quantify the average mean fluorescence intensity of the background. We then applied background subtraction by deducting this value from the total mean fluorescence intensity, thereby minimizing the impact of uneven background signals on our fluorescence measurements.

### Preprocessing gene expression data

For the snRNA-seq dataset, we aligned sequencing reads to the reference transcriptome (GRCH38) and quantified gene expression using splitpipe (Parse Biosciences) in each of the five snRNA-seq experiments. We quantified and corrected the ambient RNA signal present in our samples using Cellbender[60] remove-background (v0.2.0). Heterotypic barcode collisions were inferred in each snRNA-seq experiment using Scrublet[61] (v0.2.3) with default settings. We merged the individual snRNA-seq experiments into a single AnnData (v0.8.0) object, totaling 611,999 barcodes and 29,889 genes before additional quality control filtering. For each snRNA-seq experiment, we removed barcodes in the 95th percentile for the number of unique molecular identifiers (UMIs) detected, doublet score from Scrublet and percentage of mitochondrial reads. We also applied dataset-wide cutoffs to remove barcodes with less than or equal to 250 UMIs, greater than or equal to 50,000 total UMIs and greater than or equal to 10% mitochondrial reads. For one of the snRNA-seq experiments, we applied a more stringent filter to remove cells with less than or equal to 500 UMIs and greater than or equal to 5% mitochondrial reads. We retained 431,534 barcodes for downstream analysis.

The 10x Genomics Loupe Browser image alignment tool was used to select Visium ST spots that intersected the tissue based on the fluorescent images. Sequencing reads from the human and mouse Visium experiments were processed using the 10x Genomics Spaceranger (v1.2.1) pipeline, with GRch38 and MM10 as the respective reference transcriptomes. Spaceranger count was used to align sequencing reads to the reference, quantify gene expression and perform a preliminary clustering analysis for each sample. Unlike the snRNA-seq dataset, we did not filter out additional spots based on sequencing QC metrics; however, our mouse ST clustering analysis did reveal a group of low-quality spots that we excluded from many downstream analyses like DE. The UMI counts matrices and fluorescent images for the human and mouse samples were combined into merged Seurat[62] objects for the respective species.

### Initial snRNA-seq data analysis

Following QC filtering, we processed the snRNA-seq dataset using SCANPY[63] and scVI[64]. The UMI counts matrix was first normalized using the functions `sc.pp.normalize_total` and `sc.ppl.log1p`. We set up the AnnData object to train the scVI model using snRNA-seq as the batch key and the following additional continuous and categorical covariates—sample ID, diagnosis, brain region, age at death, percentage of mitochondrial counts, number of UMI, PMI and RIN. We set up the scVI model with two hidden layers, 128 nodes per layer, a 30-dimensional latent embedding after the encoder phase and a dropout rate of 0.1. We trained the model over 50 epochs and noted a flattened loss curve by the end of the training procedure. The latent embedding learned from the scVI model accounts for the batch effects and additional covariates specified in the model setup step, and we used this embedding for Leiden clustering and uniform manifold approximation and projection (UMAP)[65] dimensionality reduction in SCANPY. With a resolution parameter of 1.5, we identified 43 clusters. We inspected gene expression patterns in these clusters for a panel of canonical central nervous

system cell-type marker genes to assign major cell-type labels to each cluster. We also checked the distribution of QC metrics in each cluster to identify outlier clusters. Six clusters (7, 29, 33, 35, 50 and 51) were removed from the downstream analysis as QC outliers or due to the presence of potential doublets. We recomputed the UMAP and Leiden clustering (resolution = 1.2) after filtering these clusters, yielding 29 clusters. Glutamatergic neuron clusters were annotated based on the expression of known cortical layer marker genes, and GABAergic neuron clusters were annotated based on the expression of known markers (*VIP*, *SST*, *PVALB* and *LAMPS*). At this stage, non-neuronal cell clusters were simply labeled by their major cell types (astrocytes, microglia, oligodendrocytes, oligodendrocyte progenitors and vascular cells). To identify subpopulations in non-neuronal cells, we performed subclustering analysis in each of the major non-neuronal cell populations (microglia, astrocytes, oligodendrocytes and vascular cells). Each group was isolated in its own AnnData object, and Leiden clustering was performed (see GitHub repository for subclustering parameters).

### Reprocessing publicly available single-nucleus gene expression datasets

We obtained sequencing data from three published snRNA-seq studies[7–9] of AD. Sequencing data from refs. 8,9 datasets were downloaded from Synapse (syn18485175 and syn21670836), and the data from ref. 7 generated by our own group was not redownloaded. We used a uniform pipeline to process each of these datasets, with slightly varying parameters that are noted in our GitHub repository. This pipeline and the resulting AnnData objects are identical to those used in another study from our group[32], and we reiterate the main analysis steps here. Sequencing reads were pseudoaligned to the reference transcriptome (GRch38), and gene expression was quantified using the count function from kallisto bustools[66]. The ambient RNA signal was corrected in UMI counts matrices for each sample using Cellbender[60] remove-background, and we used Scrublet[61] to identify barcodes attributed to more than one cell. Individual samples were then merged into one AnnData object for each of the three studies. Analogous to the snRNA-seq data we generated in this study, we performed percentile filtering based on the following QC metrics: doublet score, number of UMI per cell and percentage of mitochondrial reads per cell. The downstream processing was performed using SCANPY[63]. Gene expression was normalized using the functions `sc.pp.normalize_total` and `sc.pp.log1p`, resulting in a ln(counts per million (CPM)) transformation of the input UMI counts data. Highly variable features were identified using `sc.pp.highly_variable_genes`, which were then scaled to unit variance and centered at zero using sc.pp.scale. Linear dimensionality reduction was performed on the scaled expression matrix using principal component analysis (PCA) with the function sc.tl.pca. Harmony[67] was used to batch-correct the PCA representation with the function `sc.external.pp.harmony_integrate`. A cell neighborhood graph was computed based on the harmony representation using sc.pp.neighbors, followed by Leiden[58] clustering and nonlinear UMAP dimensionality reduction with sc.tl.leiden and sc.tl.umap, respectively. Canonical central nervous system cell-type marker genes were used to assign coarse-grain identities to each cluster and to identify additional doublet clusters that passed our previous filtering steps. We inspected the distribution of the QC metrics in each cluster and removed outlier clusters. After filtering additional low-quality clusters, we ran UMAP and Leiden clustering again to result in the final processed AnnData object for each dataset.

### Differential cell state abundance testing in the snRNA-seq datasets

We sought to test for differential cell state abundance across conditions within our snRNA-seq dataset using the R package miloR[28] (v1.8.1). MiloR offers a statistical framework for testing for the enrichment or depletion of certain cell states across two conditions of interest

(control versus disease status) using partially overlapping cellular neighborhoods on the *k*-nearest neighbors (KNN) graph. For our differential abundance analysis, we first constructed the KNN graph and identified cellular neighborhoods using the scANVI[25] representation of our snRNA-seq dataset. After constructing this cell neighborhood cell counts matrix, a negative-binomial generalized linear model (GLM) is fit to obtain *P* values and FC differences for the differences in neighborhood abundances across the conditions of interest. For our analysis, we applied this differential abundance procedure separately for the four snRNA-seq datasets used in this study.

## ST clustering analysis

In the human and mouse ST datasets, we grouped spots into biologically relevant clusters by accounting for transcriptome measurements and spatial coordinates. The BayesSpace[22] clustering algorithm uses a low-dimensional representation of the transcriptome with a spatial before encouraging the assignment of neighboring spots in the same cluster. Critically, BayesSpace produces a single unified clustering across many different ST experiments rather than separate clustering and annotation for each ST slide. Seurat objects were converted to the SingleCellExperiment format using the function as.SingleCellExperiment. Absolute and relative spatial coordinates were stored in the metadata compartment of the SingleCellExperiment objects to inform the BayesSpace model of the spatial information, ensuring to offset each sample such that there was no overlap. Each dataset was log-normalized, and linear dimensionality reduction was performed with PCA using the function spatialPreprocess from the BayesSpace R package. Harmony[67] batch correction was applied on the basis of individual samples using the RunHarmony function. For the human dataset, we ran BayesSpace clustering using the spatialCluster function in the BayesSpace R package, varying the *q* parameter (number of resulting clusters) from five through ten. We inspected the output of each clustering and found that $q = 9$ produced results that were most consistent with the underlying anatomy of the cortex, allowing us to annotate clusters based on cortical layers and WM. Similarly, we ran BayesSpace clustering on the mouse dataset varying the *q* parameter between 10 and 20, and we selected $q = 15$ for downstream analysis. To inspect the tissue composition variability among the human ST samples, we calculated the normalized difference between the number of GM and WM spots in each sample.

## Reference-based integration of snRNA-seq datasets

We performed reference-based integration of the snRNA-seq dataset from the present study with the three published AD snRNA-seq datasets. Using our new snRNA-seq dataset as the reference, we projected the three published datasets into the reference latent space using scANVI[25], and we performed transfer learning to predict cell identities using scArches[68]. While scANVI shares similarities with the scVI model that we previously used to process our snRNA-seq data, it is a semi-supervised model that leverages cell annotations in the reference dataset to inform the latent representation of the query dataset. We trained the scANVI model separately for each of the query datasets using the class scvi.model.SCANVI, training for 100 epochs in each case. For each query dataset, this process resulted in a low-dimensional representation of the transcriptome in the latent space originally learned from the reference snRNA-seq dataset with the `model.get_latent_representation` function and predicted cell annotation labels from the model.predict function. We merged the reference dataset with the three query datasets, and we ran UMAP on the scANVI latent representation to visually represent the unified dataset in two dimensions.

## Cluster marker gene analysis

We performed cluster marker gene analysis for each snRNA-seq cluster using the snRNA-seq dataset generated in this study. For this analysis, we used a 'one-versus-rest' strategy to systematically perform differential gene expression analysis, comparing each cluster to all the other clusters. We used MAST[59] as our DE model for this test, accounting for the sequencing batch and the number of UMI per nuclei as covariates in our model. We used a similar strategy to perform cluster marker gene analysis in our human and mouse Visium ST datasets where we used the biological sample ID and the number of UMI per spot as model covariates for MAST.

## Differential gene expression analysis

We systematically performed differential gene expression analysis in each of our datasets to compare the disease conditions with controls. For all our differential gene expression tests, we use MAST[59] as the underlying model, which has shown favorable results in recent benchmarks for DE analysis in datasets with multiple sequencing batches. For the snRNA-seq dataset generated in this study, we compared gene expression between the DSAD and cognitively normal control groups for each cluster and major cell type, and we performed this analysis separately for the two brain regions profiled in this study (frontal cortex and PCC). We used the sequencing batch, number of UMI per nuclei, sample PMI and sample RIN as model covariates for these tests. We used a similar strategy for the DE analysis of the previously published snRNA-seq datasets[7–9] to compare late-stage AD samples with cognitively normal controls, accounting for the study of origin and number of UMI per nuclei as model covariates. We compared gene expression between cognitively normal controls and the three experimental groups (early-stage AD, late-stage AD and DSAD) separately in our human ST dataset. For these comparisons, we used RIN, PMI, number of UMI per spot, date of sequencing library preparation and sequencing batch as model covariates. In the mouse ST dataset, we performed differential gene expression analysis to compare gene expression between 5xFAD and WT mice in each ST cluster, and we stratified this analysis by each age group (4, 6, 8 and 12 months). For the mouse analysis, we used the number of UMI per spot, sequencing batch, date of killing and date of sequencing library preparation as model covariates. We visualized the results of the DE tests as volcano plots to show the statistical significance and the effect sizes for each gene. For the human datasets, we also visualized the DE results using a heatmap stratified by chromosome to inspect the contribution of each chromosome to the overall set of DEGs. In the human ST datasets, we 'deconvolved' the spatial DEGs by their cell populations by comparing each set of DEGs to the snRNA-seq marker genes. For each disease versus control spatial DEG, we checked if it was also a marker gene for one of the cell populations. In the case where a DEG was a marker gene in more than one population, we broke ties based on the highest effect size from the marker gene test. In this analysis, we also noted which genes were not markers of any snRNA-seq cell population. We then inspected the proportion of the set of DEGs that were attributed to each of the cell populations.

Within the human snRNA-seq and ST datasets, we compared the results of the DE analyses across the different conditions within each snRNA-seq and ST group. For these comparisons, we computed Pearson correlations and performed linear regression on the DE effect sizes for the set of genes that were significantly different between condition and control for either of the analyses. We visualized these results using scatter plots and indicated genes that were consistent or inconsistent in terms of their effect sizes across the different comparisons. We used the EnrichR[69] R package (v3.1) to identify biological pathways and processes that are enriched in our sets of DEGs and in sets of DEGs that were shared across different DE tests.

## Hierarchical co-expression network analysis in human ST

We performed hierarchical co-expression network analysis across different cortical layers and WM in our human ST dataset using the R package hdWGCNA[32,70] (v0.2.19). Before network analysis, we computed pseudo-bulk gene expression profiles for each of our 39 ST samples

in for the cortical layer clusters and WM (L1, L2–L3, L3–L4, L3–L4–L5, L5–L6, L6b, WM), and we calculated $\log_2$(CPM mapped reads) normalized expression values. Genes were retained for network analysis if they were expressed above 0 UMI counts in at least 5% of cells in any of the spatial annotations, thereby retaining 10,199 genes. We selected soft-power threshold parameters for each spatial region by computing the scale-free topology model fit for different soft-power values and using the smallest parameter that yielded a model fit greater than 0.8. Co-expression networks were then computed separately for the seven spatial regions using the hdWGCNA function ConstructNetwork, using a minimum module size of 50, a merge cut height of 0.1 for dynamic tree cutting[71], the soft-power thresholds as previously described and all other parameters set to the default values. This process yielded 166 gene co-expression modules across the seven spatial regions, and we assigned a unique name to each module based on a combination of their spatial region of origin and a numeric identifier. For example, module 'L1–M7' is the seventh module originating from spatial region L1. We calculated gene expression summary values, called MEs, for each of these 166 modules in each of the ST spots using the hdWGCNA function ModuleEigengenes, applying a Harmony95 correction to the MEs based on biological sample of origin. We then computed eigengene-based connectivity (kME) for each gene using the hdWGCNA function ModuleConnectivity.

We next sought to perform a hierarchical analysis of these co-expression networks using a strategy similar to a previous study[72]. First, we calculated similarity metrics between pairs of co-expression modules that arose from different spatial regions by computing gene overlap statistics such as Jaccard similarity ($J$) using the R package GeneOverlap (v1.34.0). Next, we calculated pairwise Pearson correlations between each of the 166 co-expression modules within each of the seven brain regions, and we kept the component-wise maximum of these correlations ($E$). Using these two measures of similarity, we computed a module-module dissimilarity matrix $D = 1 − (E + (3J)/4)$. We then computed the Euclidean distance between modules in this matrix and performed hierarchical clustering with the R function hclust. We refer to the 15 groups of modules that arose from this hierarchical clustering analysis as 'meta-modules'. Similar to our previous co-expression network analysis, we computed MEs and kMEs for these meta-modules to summarize gene expression in the ST spots and to quantify the eigengene-based connectivity of each gene. For cases where a single gene was assigned to more than one meta-module, we re-assigned the gene to only the meta-module where the gene had the highest eigengene-based connectivity. Additionally, we computed a consensus topological overlap matrix (TOM) based on the seven TOMs from the co-expression analysis in each spatial region, and we visualized this consensus TOM as a UMAP plot using the hdWGCNA function, RunModuleUMAP. We identified enriched biological processes in each of the meta-modules using the EnrichR[69] R package (v3.1) to overlap these gene sets with those associated via GO.

We performed DME analysis to compare the differences in the expression of each co-expression module between disease groups (early AD, late AD and DSAD) with cognitively normal controls. This analysis was done using the hdWGCNA function FindDMEs, using a two-sided Wilcoxon rank-sum test for the comparisons. Like differential gene expression analysis, DME analysis results in measures of statistical significance and effect sizes for each co-expression module across each comparison. For the 166 region-specific co-expression modules, we performed DME analysis only within the region that the modules were derived from. Alternatively, for the meta-modules, we performed DME analysis within each of the spatial regions. After performing these tests, we sought to compare the results of these tests across the different disease conditions to identify similarities and differences among the co-expression patterns. For the 166 co-expression modules, we computed Pearson correlations and linear regressions comparing the effect sizes of the DME results and visualized the results as scatter plots.

We next tested the overlap between sets of co-expression modules that were significantly differentially expressed between the groups using the UpSetR[73] package.

To inspect the activity of these spatially derived co-expression networks in other related contexts, we projected the 166 co-expression modules and the 15 meta-modules into our snRNA-seq and mouse ST datasets using the hdWGCNA function ProjectModules. This function computes MEs in a query dataset for sets of genes from co-expression modules that were found in a different reference dataset. In this case, the reference dataset is the human ST dataset, and the query datasets are the human snRNA-seq and the mouse ST datasets. We coarsely inspected the distributions of the meta-modules in specific cell types within the human snRNA-seq dataset with UMAP visualizations where each cell is colored by the projected ME value. In the mouse ST dataset, we inspected these trends using violin plots stratified by age group and genotype. Furthermore, in the mouse dataset, we performed module preservation analysis[33] to assess the reproducibility of these modules across species using the hdWGCNA function ModulePreservation.

### Co-expression network analysis in mouse ST
We used hdWGCNA[32] to perform a brain-wide gene co-expression network analysis in our mouse ST dataset, similar to the analysis from the mouse brain ST dataset from ref. [53], which was used to identify the 'plaque-induced gene' (PIG) network. First, because co-expression network analysis is sensitive to noise in sparse datasets, we computed 'meta-spots' by merging the transcriptomes of adjacent ST spots in a grid pattern for each of the 80 ST samples. Genes were retained for network analysis if they were expressed above 0 UMI in at least 5% of spots originating from any of the different brain regions, yielding a total of 12,579 genes. After performing a soft-power threshold search similar to our human co-expression network analysis, we computed the co-expression network TOM and identified gene modules using the hdWGCNA function ConstructNetwork with default parameters. In total, this process yielded ten brain-wide spatial co-expression modules. We computed MEs for these modules using the hdWGCNA function ModuleEigengenes, applying a Harmony[67] correction to the MEs based on the sequencing batch. Next, we computed eigengene-based connectivity (kME) for each gene using the hdWGCNA function ModuleConnectivity. We visualized the co-expression network in two dimensions by computing a UMAP representation of the TOM with the hdWGCNA function RunModuleUMAP. We used the EnrichR[69] R package (v3.1) to identify biological processes from GO that were enriched in these co-expression modules.

We sought to compare these co-expression modules with other relevant gene sets. For this analysis, we used the same gene sets described in the 'Quantifying gene expression signatures of disease-relevant gene sets' section. We performed pairwise Pearson correlations of the MEs with the UCell[74] scores of these gene signatures to assess the similarity of gene expression patterns using the hdWGCNA function ModuleTraitCorrelation. We next computed gene overlap statistics between the sets of genes in each co-expression module with these other gene sets using the R package GeneOverlap. To further our comparison with the spatial co-expression modules from ref. [53], such as the PIG module, we used the hdWGCNA function ProjectModules to quantify gene expression patterns of ref. [53] modules in our mouse ST dataset. We then computed pairwise Pearson correlations between the MEs for the co-expression modules derived from our mouse ST dataset to those derived from the dataset of ref. [53].

### Polygenic disease enrichment analysis in ST and snRNA-seq
We performed polygenic disease enrichment analysis in our ST and snRNA-seq datasets using the Python package scDRS[34] (v1.0.0). Briefly, scDRS computes cell-level disease enrichment scores using transcriptomic measurements and putative disease gene sets from GWAS. These transcriptomic datasets are inspected for deviations from expected

expression levels based on 1,000 'control' gene sets with matching mean and variance to the disease gene sets. After normalizing the disease enrichment scores, scDRS computes cell-level $P$ values using the empirical distribution of the pooled normalized control scores. In addition to cell-level enrichments, scDRS computes group-level enrichment $P$ values (cluster, cell type, region, etc.) using a unified Monte Carlo test based on the test statistics for the disease scores in a given group and the distribution of test statistics for the control scores in the same group. The scDRS Python package includes putative disease gene sets curated by the authors of scDRS, and we used these provided gene sets for our analysis (74 gene sets for humans and 22 for mice). We applied scDRS to the following datasets: human ST, mouse ST, snRNA-seq from the present study, snRNA-seq from ref. 8, snRNA-seq from ref. 9 and snRNA-seq from ref. 7. scDRS was performed in each dataset separately, and we report the results separately because batch effects from distinct datasets could potentially affect the enrichment results, which the scDRS algorithm does not explicitly account for. We computed group-level associations for cell clusters in the snRNA-seq data and the region clusters for the spatial data separately for each disease group in humans and separately for each genotype and age group in the mouse dataset. We next assessed a potential relationship between the scDRS score for AD with the spatial co-expression modules by computing Pearson correlations of MEs with the scDRS scores. These correlations were performed separately for different clusters and disease groups. This analysis was performed for the 166 spatial co-expression modules in the human ST dataset; the 15 co-expression meta-modules in the human ST dataset, the mouse ST dataset and the human snRNA-seq dataset; and the 10 mouse spatial co-expression modules.

## Sex differences within disease groups

We performed additional differential gene expression tests to identify sex differences within the different disease groups present in our dataset, using a similar strategy as the other DE tests with MAST[59] as the underlying model. For the snRNA-seq datasets, we compared expression between nuclei from female versus male samples in each cell type and each cell cluster, and we used sequencing batch, the number of UMI per nuclei and the PMI as the model covariates. Because the ST dataset has fewer samples than the snRNA-seq dataset, the results were more likely to be skewed by a dataset imbalance between the female and male samples. For instance, within the DSAD cohort, there were a greater number of female samples (seven) as compared to male samples (three). For the human ST dataset, before running the DE analysis, we first downsampled the dataset (stratified by biological sample) such that the number of spots from the female samples matched that from the male samples. We then performed DE analysis with MAST using the unique sample identifier, the number of UMI per spot and the PMI as the model covariates. For the mouse ST dataset, we performed DE tests between females and males in the 5xFAD mice within the same age group (4, 6, 8 and 12 months) using the unique sample identifier and the number of UMI per spot as model covariates. We inspected the overlap between sets of DEGs between the different spatial regions using the R package UpSetR[73] (v1.4.0). We used the EnrichR R package to identify biological processes enriched in DEGs in each spatial region. To complement the DE analysis, we also used the hdWGCNA R package to perform DME analysis to compare the expression of our co-expression modules between female and male samples in each spatial region.

## Spatial proteomics data analysis

SBT MCD files were imported into Visiopharm Software (v2022.03). Image classes were created for training and included background, nuclei and nuclei border. Nuclei detect AI App (v2023.01.2.13695), a pretrained deep learning app developed by Visiopharm, was used to detect nuclei with Ir191 and Ir193 nuclear channels. Single-cell data was exported to a.tsv file for further analysis.

We performed an unbiased clustering analysis of our IMC spatial proteomic dataset using the R package Seurat[62] (v4.3.0). We first created a Seurat object using the protein intensity by nuclei segments matrix as the input, and then we log-normalized this matrix using the Seurat function `NormalizeData`. We next performed dimensionality reduction by scaling and centering the data with the `ScaleData` function and performing PCA with the `RunPCA` function. To correct the sample-specific differences in our protein intensity data, we ran Harmony[67] to correct the PCA matrix before running Louvain clustering and UMAP. We then performed a one-versus-rest marker test (two-sided Wilcoxon rank-sum test) with the Seurat function `FindAllMarkers` to identify proteins that were significantly expressed in each cluster to annotate them with cell-type labels. Following our cell-type annotation, we performed additional Wilcoxon rank-sum tests to compare the different experimental groups (cognitively normal controls, late-stage AD and DSAD) within each cluster for each protein in our panel.

## Spatial mapping of snRNA-seq data

We mapped our snRNA-seq dataset into spatial coordinates using the R package CellTrek[45] (v0.0.94). Briefly, the CellTrek pipeline enables spatial mapping of single-cell transcriptomes by creating an integrated co-embedding of ST and single-cell data, followed by a multivariate random forest model to predict the biological coordinates from the shared feature space. In our testing, we found that this algorithm was limited in that it could not scale to large datasets comprising hundreds of thousands of single cells. Additionally, this algorithm only maps data to a single ST slide at a time. We also found that the CellTrek algorithm only provided predicted coordinates for a subset of the input single-cell transcriptomes. For these reasons, we mapped our snRNA-seq frontal cortex data to the human ST dataset in a pairwise fashion for each snRNA-seq sample and each ST sample. For a given pair of ST and snRNA-seq samples, we constructed an integrated co-embedding using the CellTrek function `traint` with default parameters. We then iteratively mapped the single-cell transcriptomes into the ST coordinates using the CellTrek function over three iterations. The second iteration only included cells that were not mapped in the first iteration, and the third iteration only included cells that were not mapped in the first or second iterations. We then computed the Euclidean distance between each mapped cell and each of the ST spots, and we labeled each cell with a spatial annotation based on the most frequently observed annotation among the labels of the ten closest spots. After running the pairwise CellTrek mappings, we compiled the results into a single table. In sum, this process yielded multiple spatial coordinates and multiple annotations for each cell across the 39 human ST samples in this study. Given that these tissue samples varied in their GM and WM content, the CellTrek mappings and inferred spatial annotations are generally not consistent across the ST samples. To come up with a consensus regional annotation across the different spatial mappings, we excluded the mappings from ST samples that were excessively high in WM or GM content. We computed a metric summarizing the GM to WM ratio in each ST sample by counting the number of GM spots and WM spots, taking the difference and dividing by the total number of spots. Positive values indicate higher GM content, while negative values indicate higher WM content. We excluded samples with greater than 0.9 and less than −0.3, thereby retaining mappings from 34 of the ST samples. For each cell, we counted the number of times it was mapped to each spatial region and labeled the cell based on the most frequently mapped region across the different samples. We further simplified these spatial annotations by upper cortical, lower cortical or WM regions. We performed differential abundance testing with miloR[28] to compare the abundance of nuclei in each cell cluster stratified by these spatial annotations between the control and DSAD groups.

## Cell–cell signaling analysis

We performed cell–cell signaling analysis in our snRNA-seq frontal cortex dataset with CellChat[46] (v1.1.3), using the predicted spatial

annotations in addition to cell-type labels. The human CellChatDB ligand–receptor interaction database was used for this analysis. To facilitate downstream comparisons of the signaling networks in DSAD versus control samples, we ran the CellChat workflow separately based on disease status. The CellChat object was created using the normalized gene expression matrix and the cell-type annotations with the predicted spatial regions from CellTrek, removing any cell groups with fewer than 30 cells. We then ran the recommended CellChat workflow using the following functions: `identifyOverExpressedGenes`, `identifyOverExpressedInteractions`, `projectData`, `computeCommunProb`, `filterCommunication`, `subsetCommunication`, `computeCommunProbPathway`, `aggregateNet` and `netAnalysis_computeCentrality`. The DSAD and control CellChat objects were merged into one object using the mergeCellChat function. We compared the signaling networks across conditions both functionally and structurally using the computeNetSimilarityPairwise function. Furthermore, we used the rankNet function to compute the relative information flow changes between DSAD and control across all signaling pathways. We identified differentially expressed ligands and receptors as well as their signaling pathways using the identifyOverExpressedGenes function, visualizing selected results with the `netVisual_bubble` function.

We next compared the results from CellChat to another cell–cell signaling analysis pipeline, LIANA[75] (R package (v0.1.13)). LIANA differs from CellChat in that it is a unified analysis package for running a number of different CCC inference methods (including CellChat) with a number of different ligand/receptor interaction databases. Thus, the results from LIANA are aggregated from different analysis approaches. Similar to our CellChat analysis, we ran LIANA with the default parameters for snRNA-seq profiles from our control and DSAD groups. To compare the results between CellChat and LIANA, we computed Pearson correlations between the number of predicted cell–cell interactions between each cell group as the signal sender versus the other groups as the signal receivers.

### Quantifying gene expression signatures of disease-relevant gene sets

We used the UCell[74] R package (v2.2.0) to quantify gene expression signatures of several relevant gene sets with the function `AddModuleScore_UCell`. The following gene sets were used for this analysis: homeostatic microglia[18], disease-associated microglia[18], disease-associated astrocytes[19], disease-associated oligodendrocytes[54] and PIGs[53]. The full list of genes within each gene set used for this analysis can be found on our GitHub repository.

### Integration of amyloid imaging data and ST data

Because we stained the brain sections used for ST with Amylo-Glo and OC, we developed a custom data analysis pipeline to identify gene expression changes associated with amyloid-β plaque depositions in our human and mouse ST datasets. For this analysis, our data processing pipeline was uniform among the human and mouse datasets. We used custom automated imaging analysis protocols (General Analysis protocols on NIS-Elements) to obtain Amylo-glo⁺ and OC⁺ binaries thresholding by intensity and size, as well as accounting for autofluorescence/nonspecific staining by negative thresholding based on an empty channel. We exported the following values for each binary: area ($\mu m^2$), diameter, center $X$ and $Y$ coordinates. Only 5xFAD samples were used for the mouse samples, as there is no amyloid pathology in WT mice. Samples with high backgrounds were excluded. The image analysis of the Amylo-Glo and OC fluorescent images gives us the coordinates and sizes of stained amyloid bodies that can then be directly compared to the Visium ST data. We separately counted the number of amyloid aggregates stained with Amylo-Glo or OC that overlapped each of the ST spots. We then calculated the number of Amylo-Glo or OC⁺ binaries per spot by testing for an intersection between a spot and a binary. The

radius of a spatial spot was calculated according to values provided by 10x Genomics, where a spot is 55 μm with a 100 μm distance between spot centers, and expanded the radius of a spot to account for the gap between spots. To account for the size of each amyloid aggregate, we also computed the sum of the areas of all amyloid aggregates overlapping each ST spot. We next used the R package Voyager[76] (v1.0.10) to perform hotspot analysis by computing the Getis-Ord Gi*[77,78] statistics for the Amylo-Glo and OC area scores in each ST sample.

### Identifying amyloid-associated gene expression signatures in human and mouse

We used the Getis-Ord Gi*[77,78] hotspot statistics for amyloid aggregates stained with Amylo-Glo and OC to identify gene expression signatures associated with amyloid aggregation in the human and mouse ST datasets. We used GLMs to identify genes that were significantly altered in expression with respect to the amyloid hotspot statistics. This analysis was done using the `fit_models` function from the R package monocle3[79] (v1.3.1). We used the biological sample of origin and the number of UMI per spot as model covariates, and statistical significance is evaluated using a two-sided Wald test. Furthermore, this analysis was performed separately for the Amylo-Glo and OC hotspot statistics because these two stains identify different forms of amyloid aggregates. Because our mouse brain dataset profiled an entire brain hemisphere and the clusters broadly corresponded to different major brain regions, we also performed this analysis separately for each of the clusters in the mouse ST dataset. Alternatively, the human dataset contains ST profiles only in the frontal cortex GM and WM. Amyloid aggregation tends to primarily occur in the GM, as seen in our hotspot analysis, and for this reason, we chose to exclude the human WM ST spots from the analysis. After running the GLM, we computed Pearson correlations between the amyloid hotspot scores and gene expression for significant results. We consider genes to be amyloid-associated if there is a significant result from the GLM (FDR < 0.05) and a positive correlation between gene expression and the amyloid score.

For each of these sets of amyloid-associated genes, we performed biological pathway enrichment analysis using the R package EnrichR[69] (v3.1) with gene sets from the GO database. In both the human and mouse datasets, we computed overlap statistics between the sets of amyloid-associated genes from Amylo-Glo and OC using the R package GeneOverlap (v1.34.0). Furthermore, we computed overlap statistics between each set of amyloid-associate genes and other disease-relevant gene sets, which were previously described in the Quantifying gene expression signatures of disease-relevant gene sets. We also performed a gene overlap analysis to compare the set of amyloid-associated genes between the human and mouse datasets.

### Statistics and reproducibility

No statistical method was used to predetermine the sample size for the experiments. The investigators were not blinded to allocation during experiments and outcome assessment. Samples were only excluded from analyses if sample loss occurred or they did not meet QC criteria. snRNA-seq samples were randomized for nuclei isolation and library preparation. Single-nucleus isolations were performed in randomized groups of 12 samples. Library preparations were performed as four batches of 24 samples, with an additional batch to increase the number of nuclei per sample for 16 samples. ST samples were assigned to slides in a supervised manner due to the restriction of four samples per slide and to ensure a control sample was included in each for imaging analysis. Human and mouse ST samples were distributed to avoid perfectly confounding variables as much as possible based on the provided sample metadata variables. All sequencing data were analyzed in total and with appropriate batch correction methods. IMC samples were allocated for $n = 2$/group per slide (three slides total) and selected to balance by sex and age where possible.

## Reporting summary

Further information on research design is available in the Nature Portfolio Reporting Summary linked to this article.

## Data availability

All raw and processed ST and single-nucleus RNA-sequencing data have been deposited into the National Center for Biotechnology Information Gene Expression Omnibus database under accession GSE233208. Our datasets are also publicly available to browse interactively on the Cellx-Gene data portal at the following link: https://cellxgene.cziscience. com/collections/7c1fbbae-5f69-4e3e-950d-d819466aecb2. Additional snRNA-seq datasets from published studies were obtained from Synapse with the following accessions: syn18485175 (ref. 8), syn21670836 (ref. 9) and syn22079621 (ref. 7). Source imaging data as well as Hyperion source data are available with figshare using this link.

## Code availability

The data analysis code used for this study is available on GitHub[80] at https://github.com/swaruplabUCI/DSAD_Spatial_Miyoshi_Morabito_2024.

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

## Acknowledgements

Funding for this work was provided by the National Institutes on Aging, Neurological Disorders and Stroke, and Drug Abuse (grants 4R01AG071683-02, P01NS084974-06A1, 1U01DA053826, U54 AG054349-06 (MODEL-AD), 5R01NS135556-02 and 3U19AG068054-02S); Adelson Medical Research Foundation funds (to V.S.), T32AG000096-38 (to E.M.) and the National Institute on Aging predoctoral fellowship (1F31AG076308-01 to S.M.). The funders had no role in study design, data collection and analysis, decision to publish or preparation of the manuscript. We thank the UCI Genomics Research and Technology Hub for providing their facilities and sequencing our snRNA-seq and ST libraries. This work used the infrastructure for high-performance and high-throughput computing, research data storage and analysis, and scientific software tool integration built, operated and updated by the Research Cyberinfrastructure Center at the UCI. We also thank R. Stanciauskas and R.D. Casas for their technical

assistance. The ABC–DS is funded by the National Institute on Aging and the National Institute for Child Health and Human Development (U01 AG051406, U01 AG051412 and U19 AG068054). The work contained in this publication was also supported through the following National Institutes of Health Programs: The Alzheimer's Disease Research Centers Program (P50 AG008702, P30 AG062421, P50 AG16537, P50 AG005133, P50 AG005681, P30 AG062715 and P30 AG066519), the Eunice Kennedy Shriver Intellectual and Developmental Disabilities Research Centers Program (U54 HD090256, U54 HD087011 and P50 HD105353), the National Center for Advancing Translational Sciences (UL1 TR001873, UL1 TR002373, UL1 TR001414, UL1 TR001857 and UL1 TR002345), the National Centralized Repository for Alzheimer Disease and Related Dementias (U24 AG21886) and DS-Connect (The Down Syndrome Registry) supported by the Eunice Kennedy Shriver National Institute of Child Health and Human Development. We are grateful to the ABC–DS study participants, their families and care providers, and the ABC–DS research and support staff for their contributions to this study. This manuscript has been reviewed by ABC–DS investigators for scientific content and consistency of data interpretation with previous ABC–DS study publications.

## Author contributions

E.M., S.M. and V.S. conceptualized this study. E.M., S.M. and V.S. wrote the manuscript with assistance and approval from all authors. E.M. generated the ST data with assistance from C.M.H. and S.K.S. E.M. and V.M.S. generated the spatial proteomic data with assistance from J.L. E.M. and S.K.S. generated the snRNA-seq libraries with assistance from C.M.H. and M.A.A. S.D. performed dissections and RNA isolation. S.M. performed bioinformatics and network analysis on human and mouse datasets with assistance from E.M., N.R., F.R., Z.S., S.S. and Z.C. V.M.S. also helped with spatial proteomic data analyses. E.M., C.M.H., N.M., M.A.A., S.K.S. and V.S. processed mouse samples. N.M., S.S., C.M.H. and K.N.G. generated the mouse colony. N.M., S.S., C.M.H. and S.D. performed genotyping. S.D. and N.E. performed immunofluorescence experiments. S.W, I.T.L., J.S, E.D., W.H.Y., K.L, M.P.-R. and E.H. provided human brain samples from UCI ADRC.

## Competing interests

The authors declare no competing interests.

## Additional information

**Extended data** is available for this paper at https://doi.org/10.1038/s41588-024-01961-x.

**Correspondence and requests for materials** should be addressed to Vivek Swarup.

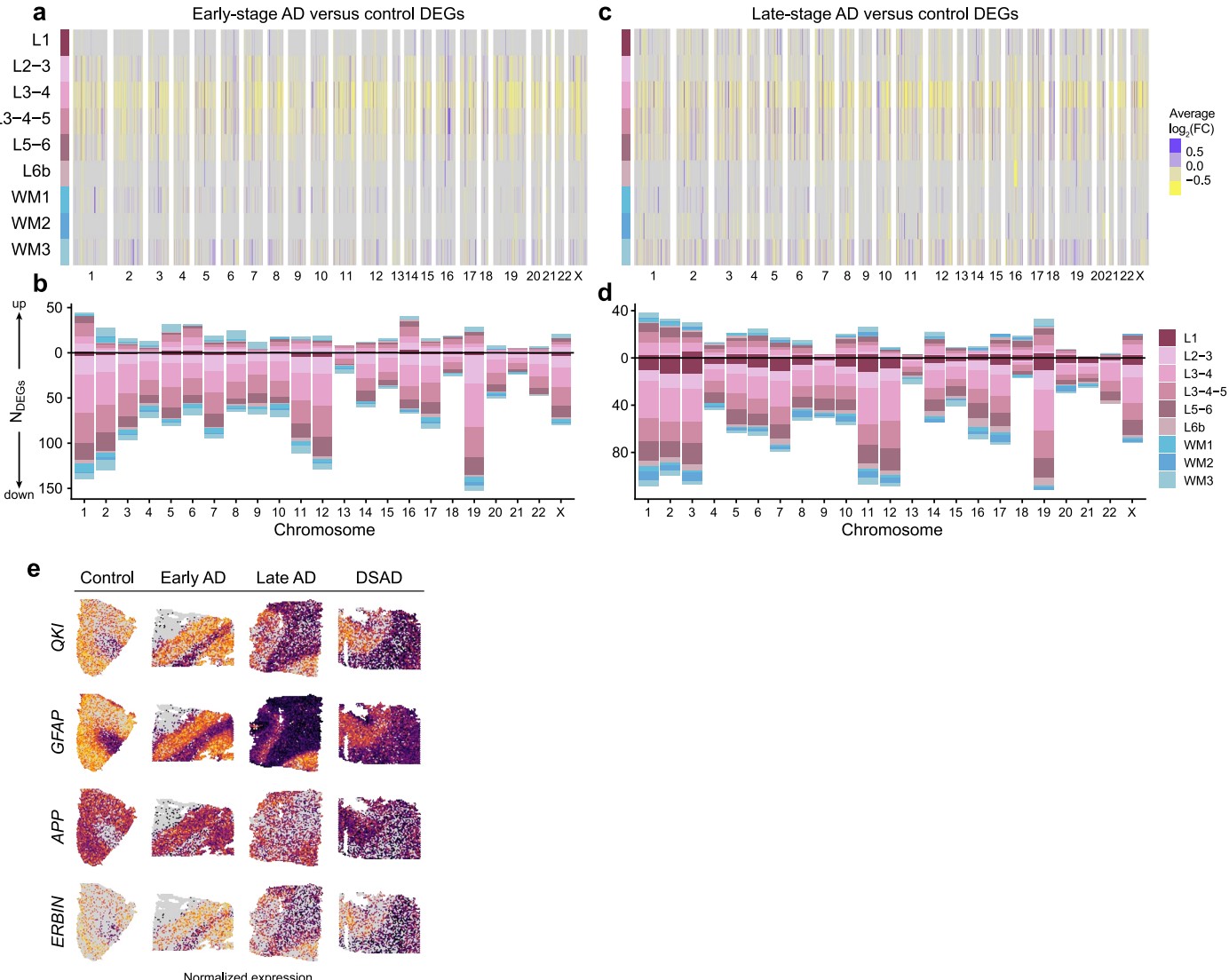

**Extended Data Fig. 1 | Spatial transcriptomic DEGs examined by chromosome. a**, Heatmap colored by effect size from the spatial transcriptomic early-stage AD versus control differential gene expression analysis, with genes stratified by chromosome and by spatial region. Statistically significant (FDR < 0.05) genes with an absolute average log₂(fold change) ≥ 0.25 in at least one region are shown. **b**, Stacked bar chart showing the number of spatial transcriptomic early-stage AD control DEGs in each spatial cluster stratified by chromosome. **c**, Heatmap colored by effect size from the spatial transcriptomic late-stage AD versus control differential gene expression analysis, with genes stratified by chromosome and by spatial region. Statistically significant (FDR < 0.05) genes with an absolute average log₂(fold change) ≥ 0.25 in at least one region are shown. **d**, Stacked bar chart showing the number of spatial transcriptomic late-stage AD versus control DEGs in each spatial cluster stratified by chromosome. **e**, Spatial feature plots of four selected DEGs in one representative sample from each disease group.

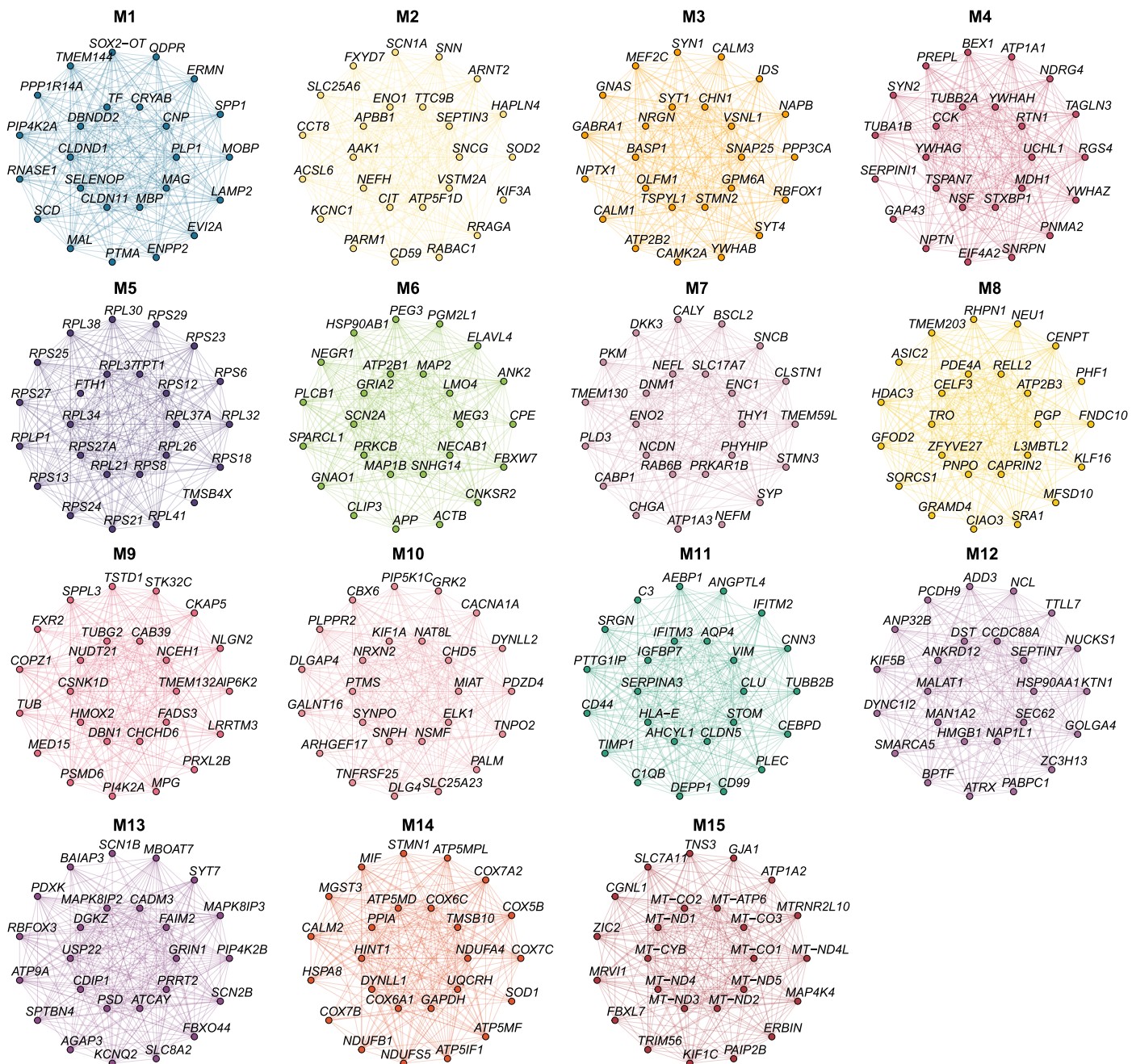

**Extended Data Fig. 2 | Module hub gene networks for the human ST co-expression meta-modules.** Hub gene networks for each of the 15 human spatial co-expression meta-modules. The top 25 hub genes ranked by kME are visualized. Nodes represent genes, and edges represent co-expression links.

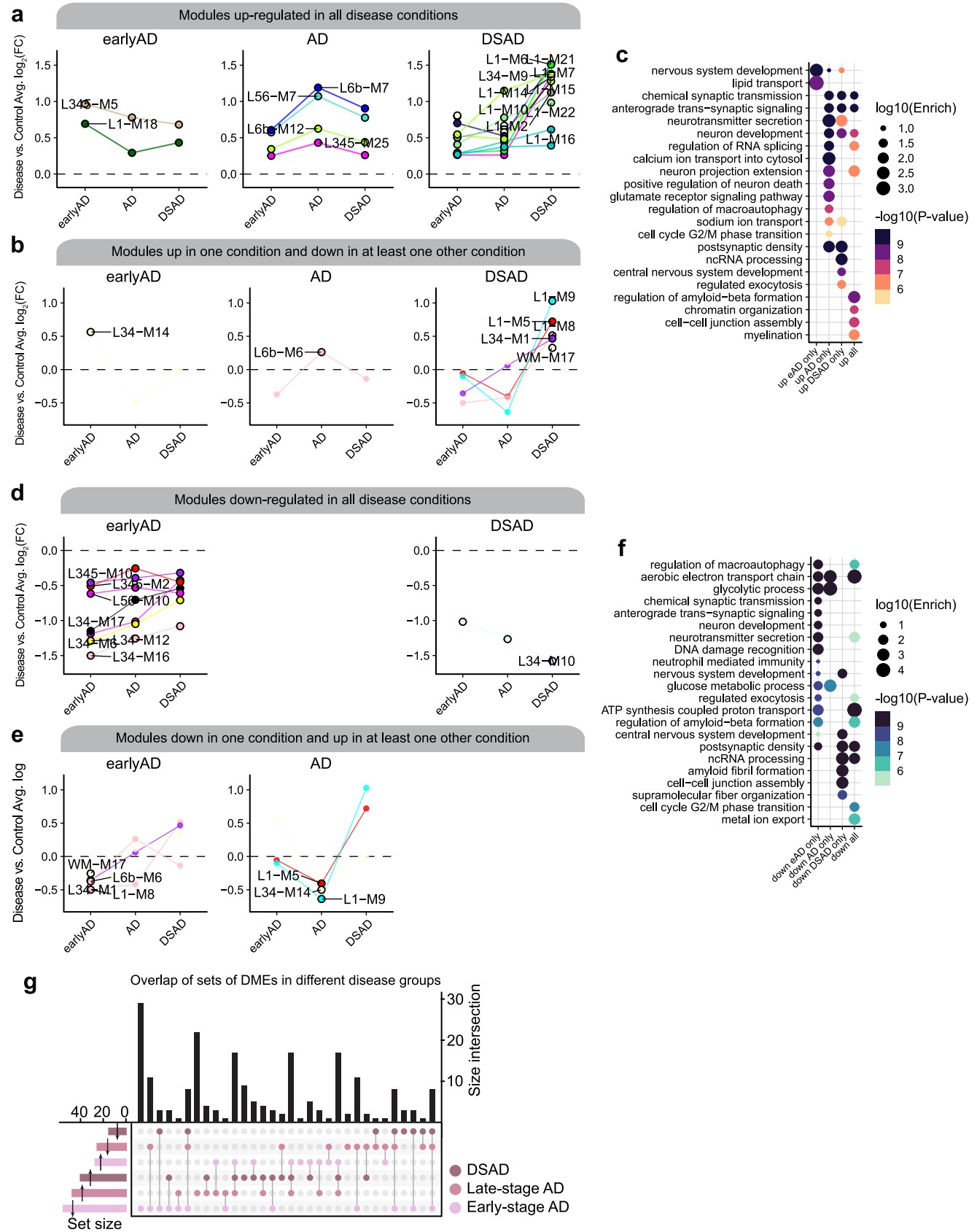

**Extended Data Fig. 3 | See next page for caption.**

**Extended Data Fig. 3 | Differential module eigengenes (DMEs) among disease conditions. a**, DMEs that are upregulated in all disease conditions compared to control. Modules are grouped into different plots based on which disease condition had the highest effect size. Higher fold change values correspond to modules that are upregulated in disease. **b**, DMEs that are upregulated in one condition and downregulated in at least one other condition. **c**, Selected pathway enrichment results for the top 25 hub genes for each module for the set of modules upregulated in different disease conditions. **d**, DMEs that are downregulated in all disease conditions compared to control, similar to **a**. **e**, DMEs that are downregulated in one condition and upregulated in at least one other condition. **f**, Selected pathway enrichment results for the top 25 hub genes for each module for the set of modules downregulated in different disease conditions. **g**, Upset plot showing the overlap between sets of differentially expressed modules in each disease group. One-sided Fisher's exact test was used for enrichment analysis.

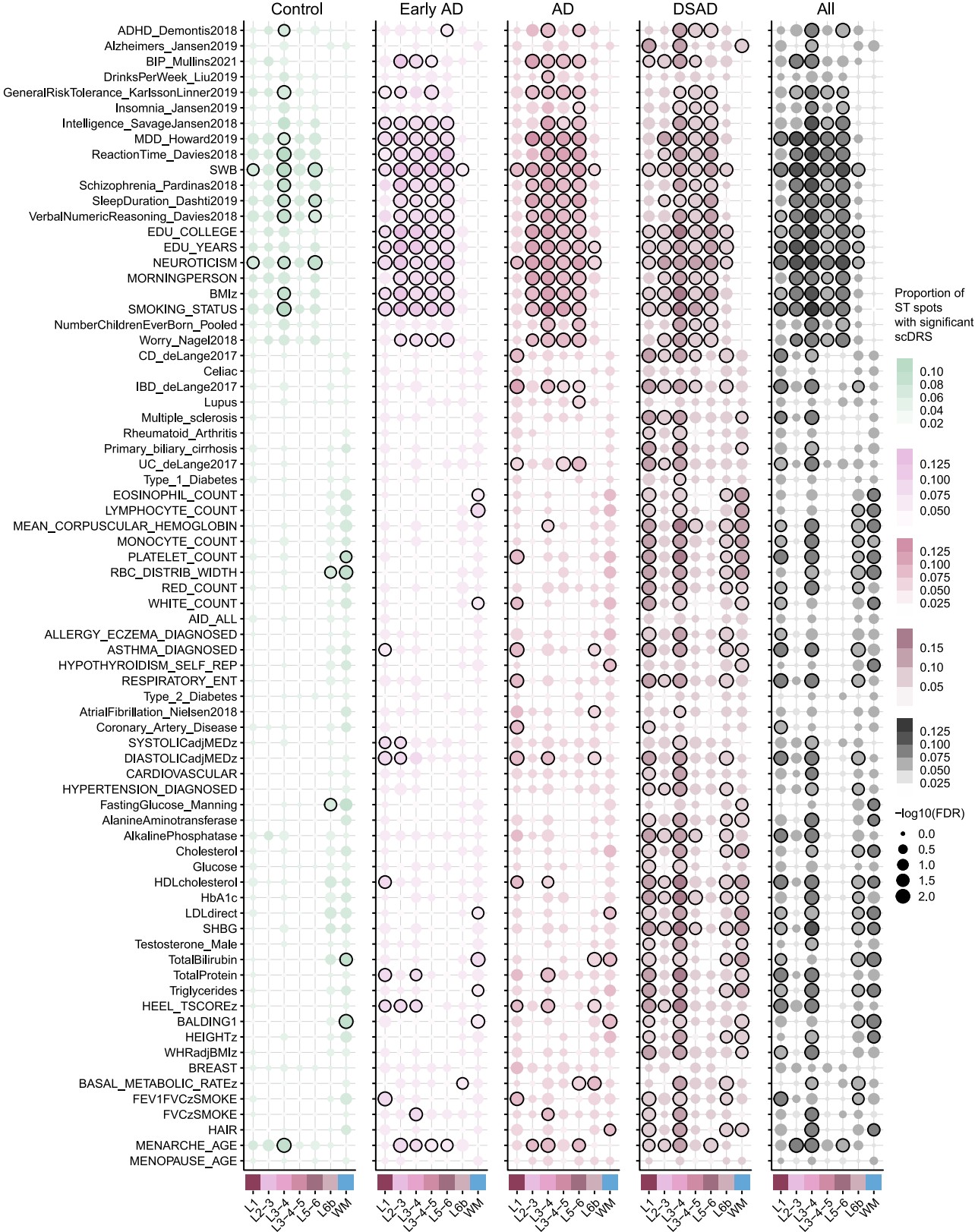

**Extended Data Fig. 4 | Genetic enrichment analysis in human spatial transcriptomics.** Dot plots showing the results of genetic enrichment analysis performed in the human Visium ST dataset using scDRS. The scDRS Python package was run on the human ST dataset to compute spatial transcriptomic disease relevance scores (st-DRS) across a corpus of 74 traits provided by the scDRS package, resulting in spot-level disease enrichment scores and significance levels. Gene-trait association information was derived from the scDRS package, which was compiled from several genetic studies[37,81–90] A Monte Carlo (MC) test was used to test for group-level significance between each trait and the ST clusters, separately for each disease group and the entire dataset together (all, right side). Black outlines on the dots denote a significant group-level association (FDR ≤ 0.05).

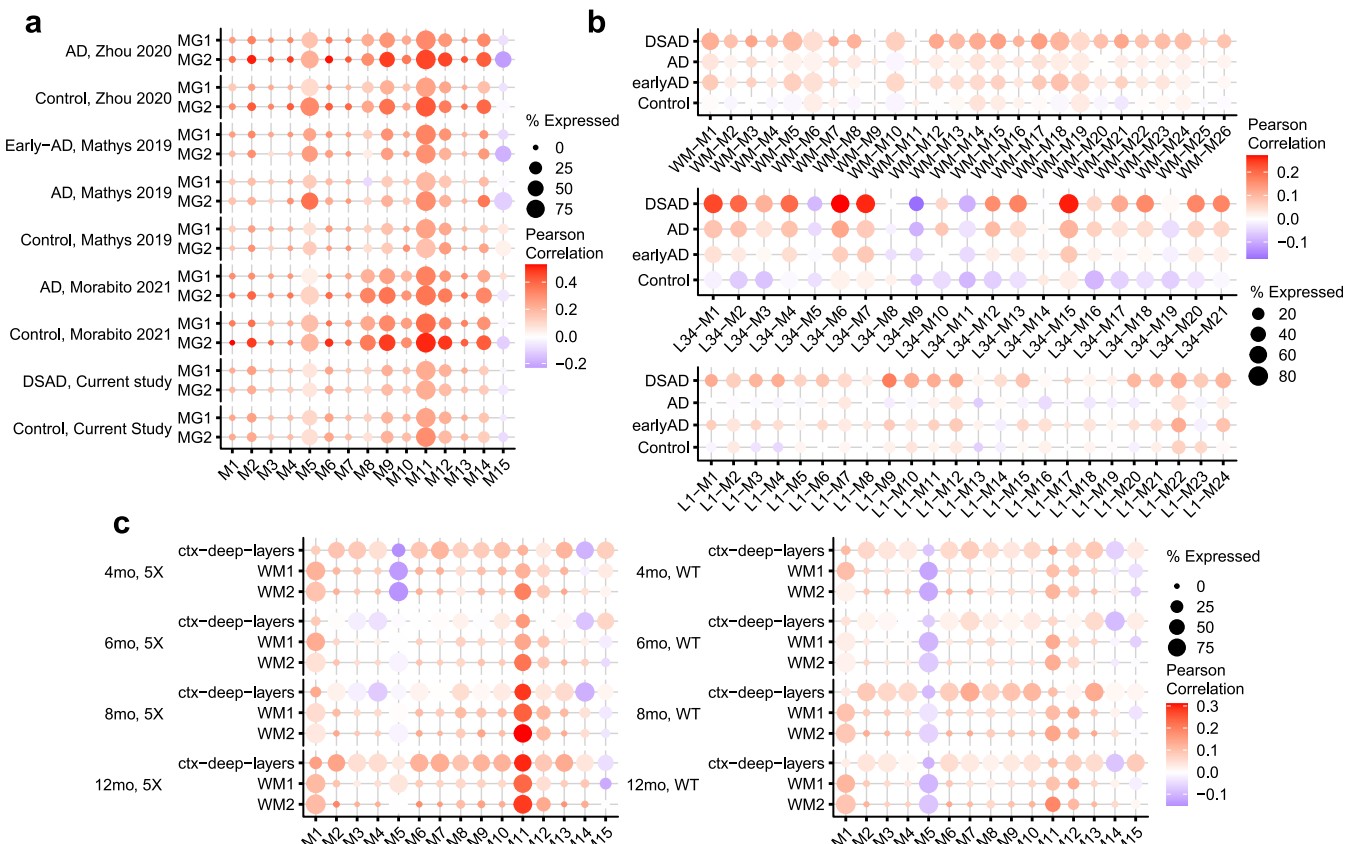

**Extended Data Fig. 5 | Correlation of co-expression network module eigengenes and scDRS genetic enrichment.** Dot plots show the percentage of snRNA-seq nuclei or ST spots in each group as the size and the correlation of the module eigengenes (MEs) as the color in the human snRNA-seq dataset (**a**), the human spatial transcriptomics (ST) dataset (**b**) and the mouse ST dataset (**c**). For this visualization, only groups with a significant group-level association (microglia clusters MG1 and MG2 for example) are included.

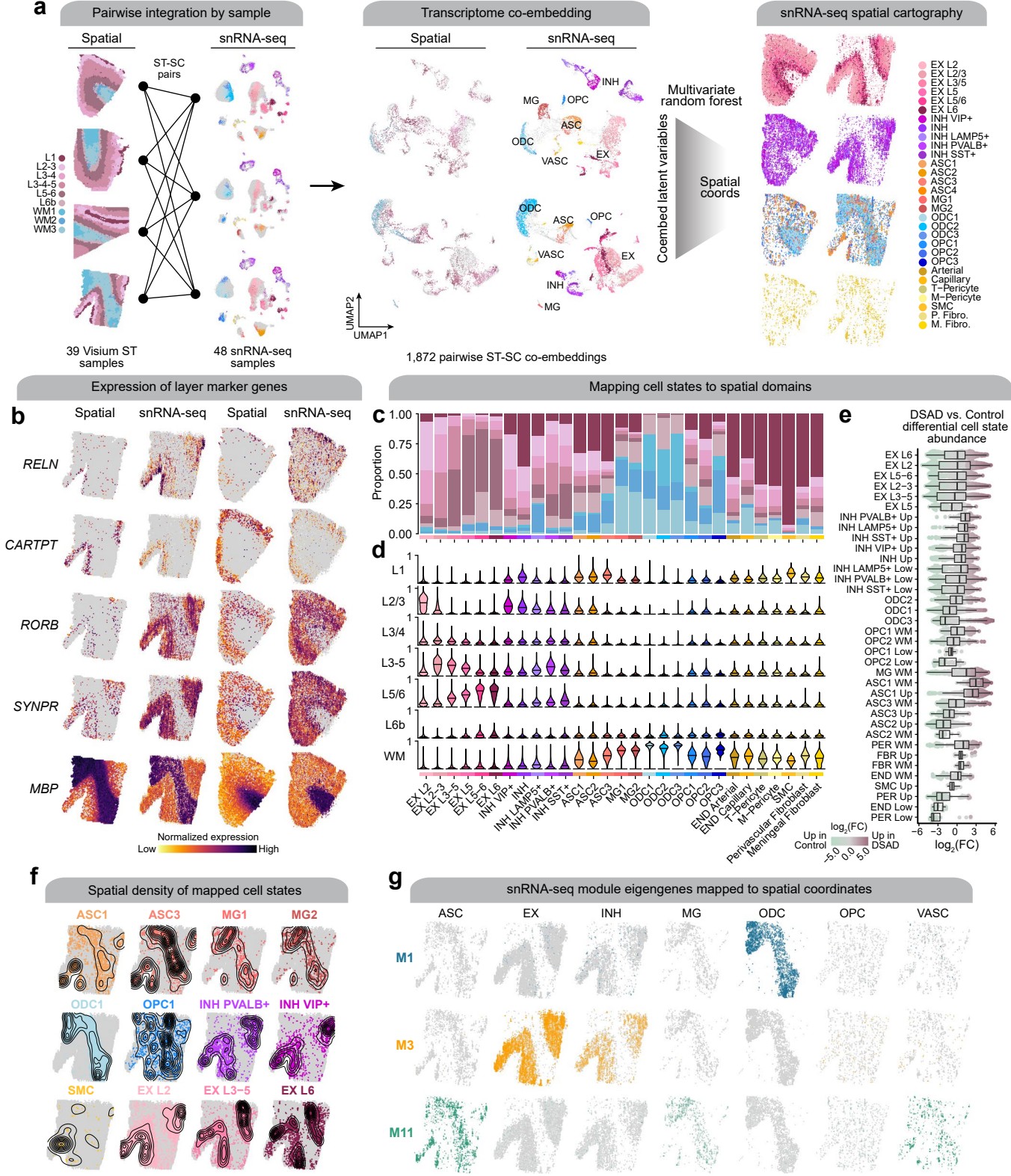

**Extended Data Fig. 6 | See next page for caption.**

**Extended Data Fig. 6 | Systematic integration of spatial and single-nucleus expression profiles. a**, Pairwise integration of samples from spatial and single-nucleus transcriptomics (left). For all possible pairs of ST + snRNA-seq samples, we constructed a transcriptomic co-embedding (middle) and used a multivariate random forest (CellTrek[45]) to predict the spatial coordinates of snRNA-seq cells in the given spatial context. The snRNA-seq dataset is shown on the right projected into two different spatial contexts (left: control sample; right: DSAD sample), split by major cell lineages and colored by cell annotations. **b**, Spatial feature plots of selected layer-specific marker genes, shown side-by-side in the ST dataset and the snRNA-seq dataset projected into the spatial context for one DSAD sample (left) and one control sample (right). **c**, Proportion of nuclei from each snRNA-seq cluster mapped to the spatial domains defined by the ST clustering.

**d**, Distribution of spatial domain mapping probabilities for nuclei from each of the snRNA-seq clusters. **e**, Box and whisker plots showing differential cell composition between disease and control. Groups are organized on the y-axis by major cell types and ordered by median fold-change values within each cell type. Box boundaries and lines correspond to the IQR and median, respectively. Whiskers extend to the lowest or highest data points that are no further than 1.5 times the IQR from the box boundaries. Each data point represents a single-cell neighborhood from Milo; the number of cell neighborhoods per cluster is shown in Supplementary Table 5. **f**, Spatial density plot showing the snRNA-seq dataset in predicted spatial coordinates, highlighting selected cell populations. **g**, Spatial feature plots showing selected module eigengenes in the snRNA-seq dataset in predicted spatial coordinates.

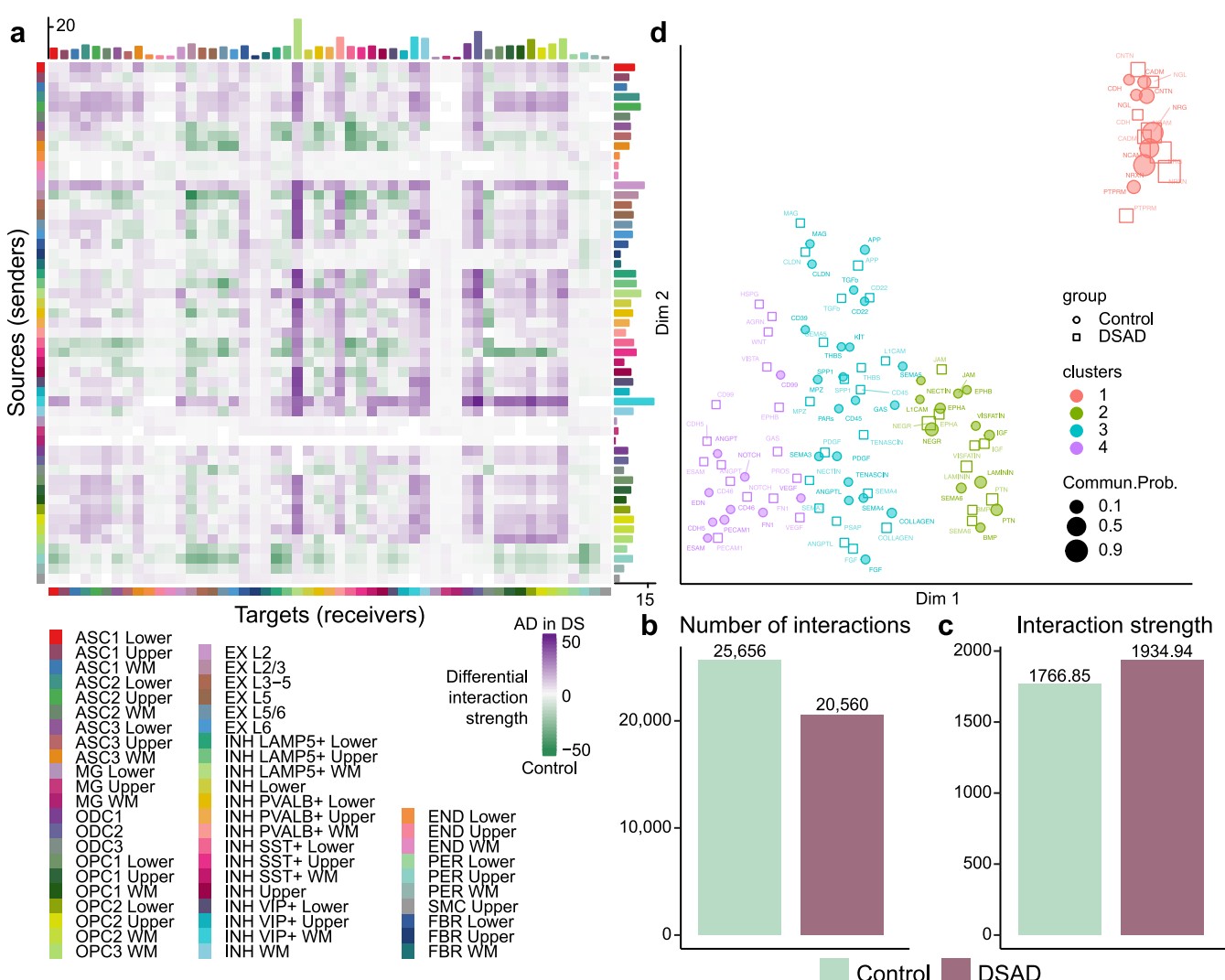

**Extended Data Fig. 7 | Overview of cell–cell communication analysis. a,** Heatmap showing the differential cell–cell communication (CCC) interaction strength between AD in DS and control. Each cell represents a snRNA-seq cell population, where rows correspond to signaling sources and columns correspond to signaling targets. Bar plots on the top right show the sum of the incoming and outgoing signaling, respectively. **b,c,** Bar plots showing the total number of CCC interactions (**b**) and interaction strength (**c**) for control and AD in DS. **d,** Joint dimensionality reduction and clustering of signaling pathways inferred from AD in DS and control data based on their functional similarity. Each point represents a signaling pathway.

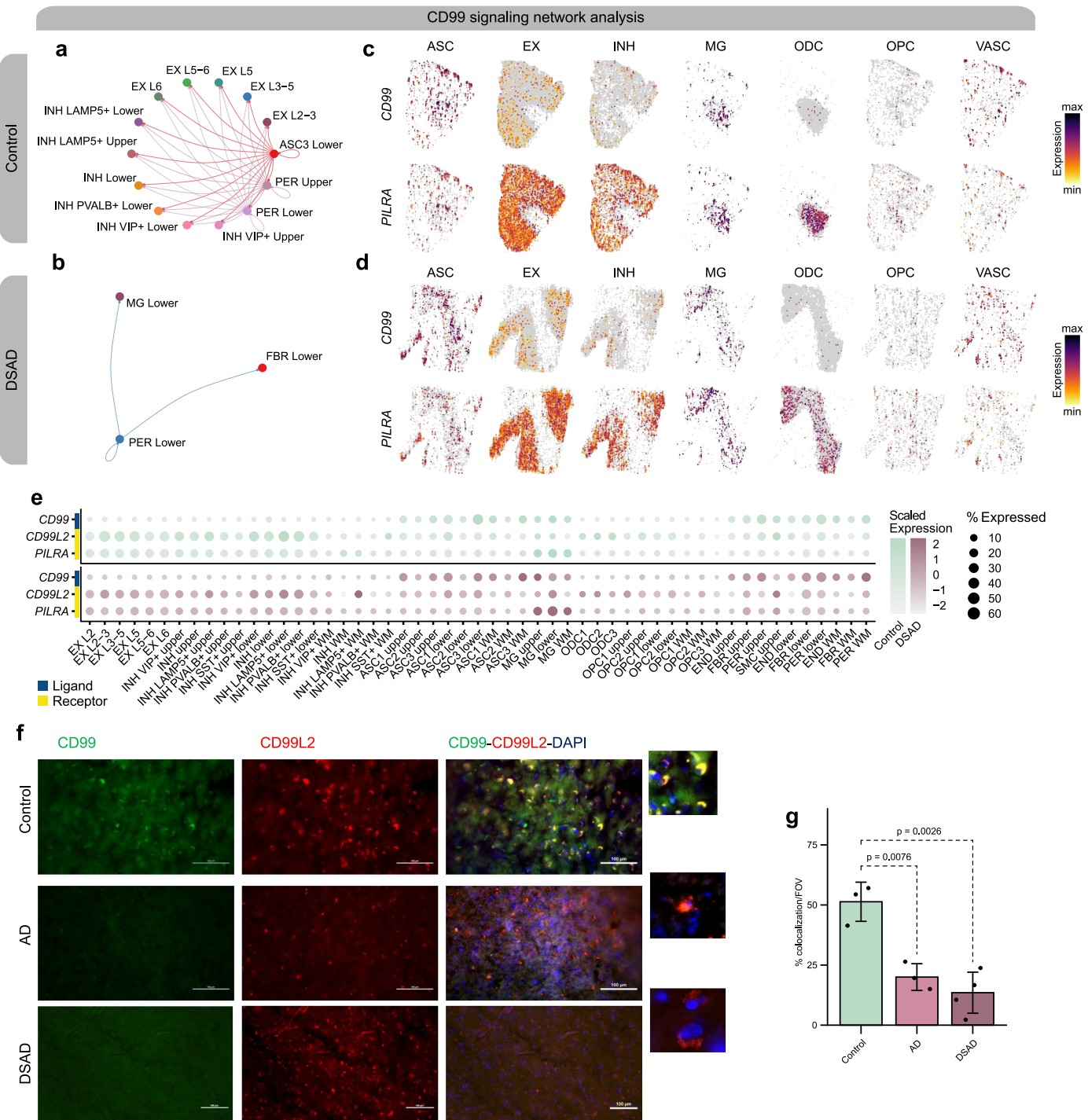

**Extended Data Fig. 8 | CD99 signaling changes between DSAD and control.**
**a,b**, Network plot showing the CCC signaling strength between different cell populations in controls (**a**) and DSAD (**b**) for the CD99 signaling pathway. **c,d**, Spatial feature plots of the snRNA-seq in predicted spatial coordinates for one control sample (**c**) and one DSAD sample (**d**) for one ligand and one receptor in the CD99 pathway. **e**, Dot plot showing gene expression in the snRNA-seq dataset of ligands and receptors in the CD99 signaling pathway with significant interactions based on CellChat. **f**, Representative double immunofluorescence

images for CD99 (green), CD99L2 (red) and DAPI (blue) from postmortem human brain tissue (prefrontal cortex, PFC) of control, AD and ADDS cases. Images were captured using a Nikon ECLIPSE Ti2 inverted microscope. **g**, Bar graph representing results of colocalization analysis from ×60 images (n = 3 cognitively healthy control, n = 3 AD and n = 4 DSAD cases) using the JACoP Plugin from ImageJ and Manders' correlation coefficient. Data are presented as the average of three different fields of view (FOVs) per sample. P-values from two-way t-tests are shown.

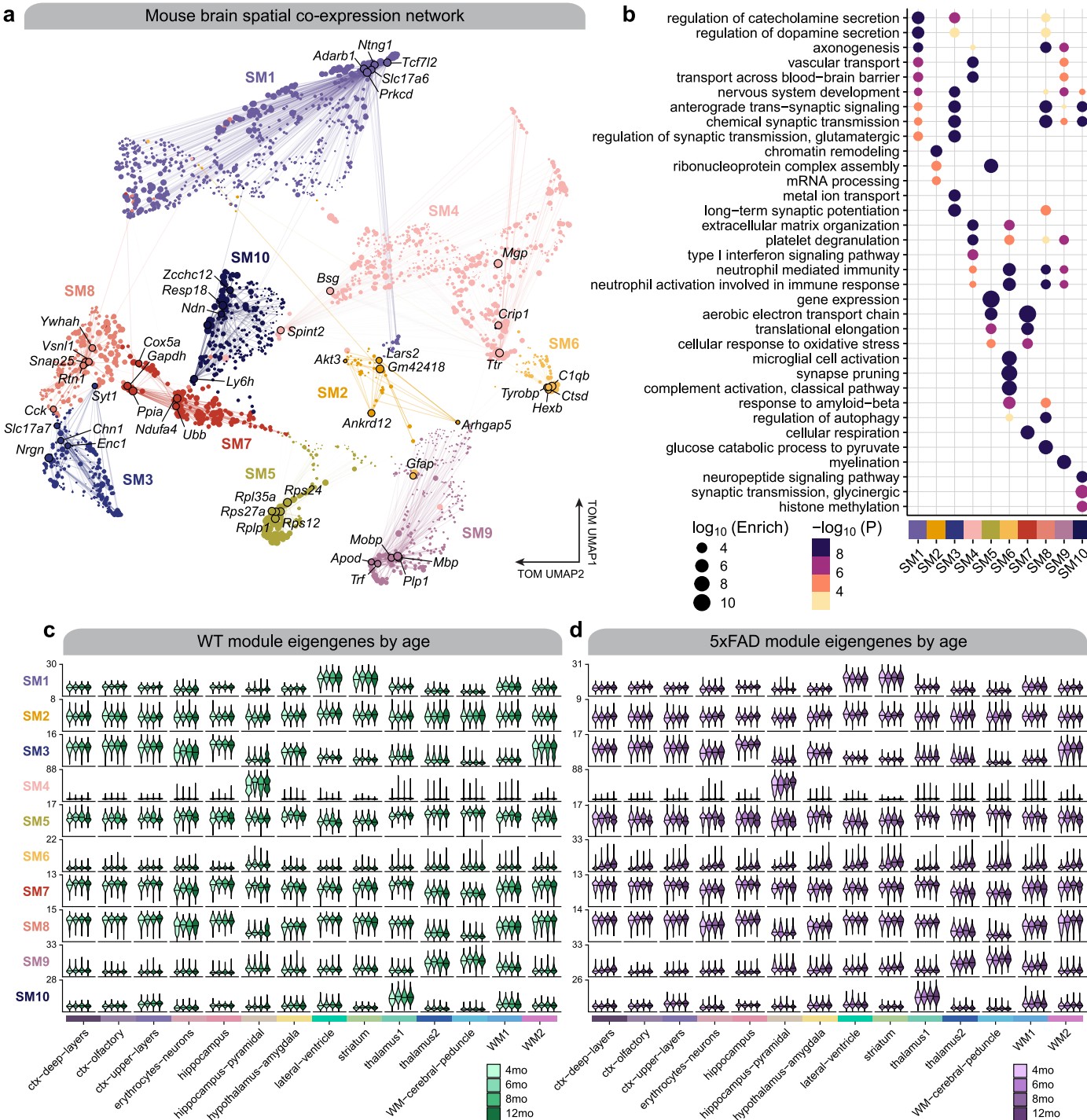

**Extended Data Fig. 9 | Co-expression network analysis in the mouse ST dataset. a**, UMAP plot of the mouse spatial co-expression network. Each node represents a single gene, and edges represent co-expression links between genes and module hub genes. Point size is scaled by eigengene-based connectivity. Nodes are colored by co-expression module assignment. The top five hub genes per module are labeled. Network edges were downsampled for visual clarity.

**b**, Dot plot showing selected GO enrichment results for each co-expression module. **c,d**, Module eigengene (ME) distributions for the ten mouse co-expression modules in each mouse age group (control, early-stage AD, late-stage AD and AD in DS) stratified by cluster for wild-type (**c**) and 5xFAD mice (**d**). One-sided Fisher's exact test was used for enrichment analysis.

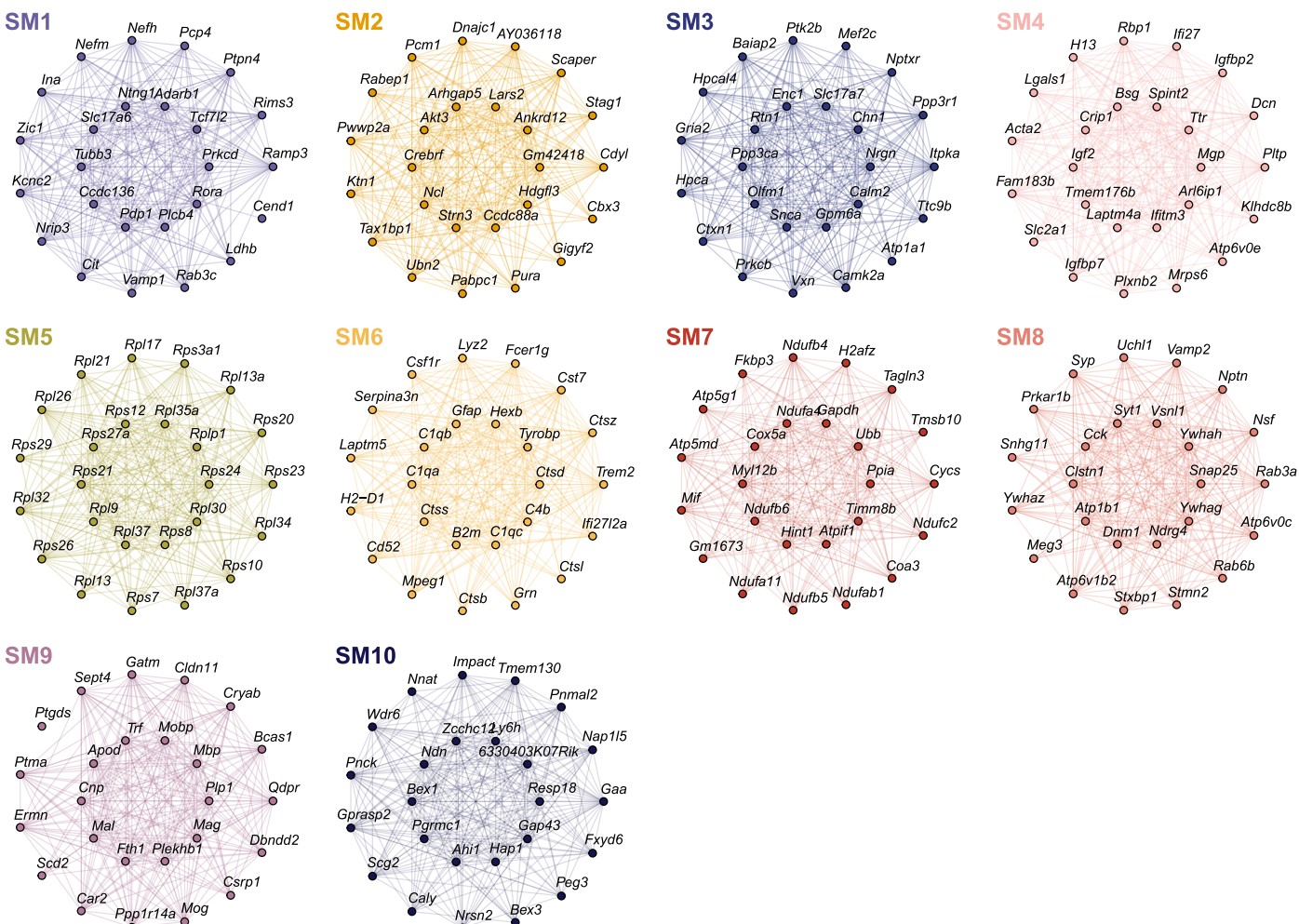

**Extended Data Fig. 10 | Module hub gene networks from the mouse co-expression network analysis.** Hub gene networks for each of the 10 mouse spatial co-expression modules. The top 25 hub genes ranked by kME are visualized. Nodes represent genes, and edges represent co-expression links.

# Reporting Summary

## Statistics

For all statistical analyses, confirm that the following items are present in the figure legend, table legend, main text, or Methods section.

| n/a | Confirmed | |
|---|---|---|
| ☐ | ☒ | The exact sample size (*n*) for each experimental group/condition, given as a discrete number and unit of measurement |
| ☐ | ☒ | A statement on whether measurements were taken from distinct samples or whether the same sample was measured repeatedly |
| ☐ | ☒ | The statistical test(s) used AND whether they are one- or two-sided<br>*Only common tests should be described solely by name; describe more complex techniques in the Methods section.* |
| ☐ | ☒ | A description of all covariates tested |
| ☐ | ☒ | A description of any assumptions or corrections, such as tests of normality and adjustment for multiple comparisons |
| ☐ | ☒ | A full description of the statistical parameters including central tendency (e.g. means) or other basic estimates (e.g. regression coefficient) AND variation (e.g. standard deviation) or associated estimates of uncertainty (e.g. confidence intervals) |
| ☐ | ☒ | For null hypothesis testing, the test statistic (e.g. *F*, *t*, *r*) with confidence intervals, effect sizes, degrees of freedom and *P* value noted<br>*Give P values as exact values whenever suitable.* |
| ☒ | ☐ | For Bayesian analysis, information on the choice of priors and Markov chain Monte Carlo settings |
| ☒ | ☐ | For hierarchical and complex designs, identification of the appropriate level for tests and full reporting of outcomes |
| ☐ | ☒ | Estimates of effect sizes (e.g. Cohen's *d*, Pearson's *r*), indicating how they were calculated |

*Our web collection on statistics for biologists contains articles on many of the points above.*

## Software and code

Policy information about availability of computer code

| Data collection | No software was used for data collection |
|---|---|
| Data analysis | All analysis was done in R (v4.0 and higher) or python (v 3.6 or higher). All software used in the analysis is described in materials and methods section and code used is available in swaruplab github. snRNA-seq and spatial transcriptomics analyses were conducted using Spaceranger (v 1.2.1), Scanpy (v1.8), BayesSpace (v 1.10.1), CellTrek (v 0.0.94), CellChat (v 1.1.3), UCell R package (v 2.2.0), monocle3 (v 1.3.1), EnrichR (v 3.1), GeneOverlap (v 1.34.0), Seurat (v 4.3.0), Voyager (v 1.0.10), UpSetR (v 1.4.0), hdWGCNA (v 0.2.19), Cellbender (v 0.2.0), Scrublet (v 0.2.3), scDRS (v 1.0.0), MiloR (v 1.8.1), and LIANA (v 0.1.13) softwares. Spatial proteomic analysis was done using Seurat (v 4.3.0). |

For manuscripts utilizing custom algorithms or software that are central to the research but not yet described in published literature, software must be made available to editors and reviewers. We strongly encourage code deposition in a community repository (e.g. GitHub). See the Nature Portfolio guidelines for submitting code & software for further information.

## Data

Policy information about <u>availability of data</u>

All manuscripts must include a <u>data availability statement</u>. This statement should provide the following information, where applicable:

- Accession codes, unique identifiers, or web links for publicly available datasets
- A description of any restrictions on data availability
- For clinical datasets or third party data, please ensure that the statement adheres to our <u>policy</u>

All data in this manuscript have been deposited in the NIH GEO database. GSE233208 can be accessed with token: aduxqegclhmjzcj.

## Research involving human participants, their data, or biological material

Policy information about studies with <u>human participants or human data</u>. See also policy information about <u>sex, gender (identity/presentation), and sexual orientation</u> and <u>race, ethnicity and racism</u>.

| | |
|---|---|
| Reporting on sex and gender | Sex was used as a covariate throughout the analysis. Case and control was balanced for sex and sex-specific differences are reported throughout the study. |
| Reporting on race, ethnicity, or other socially relevant groupings | Race and ethnicity data where available are provided. No separate analysis for race and ethnicity were performed. |
| Population characteristics | Population characteristics were individually collected and available clinical and demographic data including age and sex are available in Supplemental Table 1. |
| Recruitment | Human brain tissue from prefrontal cortex and posterior cingulate cortex was obtained from UC Irvine's Alzheimer's Disease Research Center and the NIH NeuroBioBank. Samples were assigned to groups based on both NFT and plaque staging, in addition to clinical diagnoses. Samples were also selected based upon several covariates, including age, sex, race, postmortem interval (PMI), RNA integrity number (RIN), and disease comorbidity. |
| Ethics oversight | Postmortem tissue were de-identified before acquisition and thus exempt from IRB approval. Exemption to this effect was obtained from UCI's Institutional Review Board (IRB). |

Note that full information on the approval of the study protocol must also be provided in the manuscript.

# Field-specific reporting

Please select the one below that is the best fit for your research. If you are not sure, read the appropriate sections before making your selection.

☒ Life sciences ☐ Behavioural & social sciences ☐ Ecological, evolutionary & environmental sciences

For a reference copy of the document with all sections, see nature.com/documents/nr-reporting-summary-flat.pdf

# Life sciences study design

All studies must disclose on these points even when the disclosure is negative.

| | |
|---|---|
| Sample size | Sample sizes were chosen based on sample availability and power analysis with previously available data (Morabito et al., Nature Genetics, 2021 and Chen et al., Cell, 2020) |
| Data exclusions | We used network connectivity based outlier removal to remove outliers from the analysis. However the raw data included the outlier samples. |
| Replication | Additional samples were used for replication and where applicable (like snRNA-seq and spatial transcriptomics data) data were correlated by published datasets for replication, |
| Randomization | Randomization is not relevant for this study - blinded identity (de-identified) samples were obtained from brain banks and were given sample level metadata by the brain bank. when subset-level data was analyzed the subset was randomized. |
| Blinding | Blinding is not relevant for this study, samples were obtained from the brain banks. |

# Reporting for specific materials, systems and methods

We require information from authors about some types of materials, experimental systems and methods used in many studies. Here, indicate whether each material, system or method listed is relevant to your study. If you are not sure if a list item applies to your research, read the appropriate section before selecting a response.

## Materials & experimental systems

| n/a | Involved in the study |
|-----|----------------------|
| ☐ | ☒ Antibodies |
| ☒ | ☐ Eukaryotic cell lines |
| ☒ | ☐ Palaeontology and archaeology |
| ☐ | ☒ Animals and other organisms |
| ☒ | ☐ Clinical data |
| ☒ | ☐ Dual use research of concern |
| ☒ | ☐ Plants |

## Methods

| n/a | Involved in the study |
|-----|----------------------|
| ☒ | ☐ ChIP-seq |
| ☒ | ☐ Flow cytometry |
| ☒ | ☐ MRI-based neuroimaging |

# Antibodies

| | |
|---|---|
| Antibodies used | ANGPTL4 Antibody; Cat#710186; 1:500; ThermoFisher, GFAP Polyclonal Antibody; Cat#PA3- 16727; 1:500; ThermoFisher, CD99 antibody; Cat#MA5-12287; 1:250; ThermoFisher, CD99L2 antibody; Cat#PA5-58539; 1:500; ThermoFisher, Nectin 2 Antibody; Cat#PA582470;1:250;ThermoFisher, MAP2 Antibody; Cat#PA1-10005; 1:250; ThermoFisher, C1QB Polyclonal Antibody; Cat#PA5-42554; 1:250; ThermoFisher. For Imaging mass cytometry (Hyperion) data we used these antibodies - CD44, Clone#IM7, BioLegend, 5 µg/ml; rPTPRM, Clone#MAB4446, R&D Systems, 30 µg/ml; Moesin, Clone#MSN492, Biotium, 30 µg/ml; Cystatin C, Clone#MA5-29195, ThermoFisher, 25 µg/ml; B-Amyloid, Clone#6E10, BioLegend, 5 µg/ml; CD68, Clone#KP1, BioLegend, 10 µg/ml; MAP2, Clone#EPR19691, Abcam, 2 µg/ml; ERBIN, Clone#AF7866, R&D Systems, 40 µg/ml; BIN1, Clone#EPR13463-25, Abcam, 30 µg/ml; CD163, Clone#EPR19518, Abcam, 30 µg/ml; GFAP, Clone#2E1.E9, BioLegend, 2 µg/ml; Foxp2, Clone#AF5647, R&D Systems, 20 µg/ml; NeuN, Clone#D4G4O, Cell Signaling, 20 µg/ml; APOE, Clone#WUE-4, Novus, 20 µg/ml; Midkine, Clone#EP1143Y, Abcam, 40 µg/ml; CLP-1, Clone#AF2690, R&D Systems, 20 µg/ml; COL25A1, Clone#540802, R&D Systems, 40 µg/ml; GPC5, Clone#297716, R&D Systems, 40 µg/ml; pTau, Clone#AT8, ThermoFisher, 20 µg/ml; Iba1, Polyclonal#019-19741, Wako, 15 µg/ml; Mac-2/Gal3, Clone#M3/38, Cedarlane, 20 µg/ml; YKL-40, Clone#ab180569, Abcam, 30 µg/ml; S100b, Clone#EP1576Y, Abcam, 2 µg/ml; ApoJ/Clusterin, Clone#210, ThermoFisher, 40 µg/ml. |
| Validation | Antibodies were validated by respective manufactures and available on their website. Additional validations were also run before using the antibodies for Hyperion panel by IMC core at UCI |

# Animals and other research organisms

Policy information about studies involving animals; ARRIVE guidelines recommended for reporting animal research, and Sex and Gender in Research

| | |
|---|---|
| Laboratory animals | C57BL/6J mouse, 5xFAD mouse model of AD harboring five familial AD mutations. 5xFAD hemizygous (C57BL16) and wildtype littermates were bred and housed until sacrifice at 4, 6, 8, and 12 months. |
| Wild animals | No wild animals were used in the study |
| Reporting on sex | Sex was considered throughout the study for both human and mouse studies. Sex specific differences were described in the results section |
| Field-collected samples | No field-collected samples were collected |
| Ethics oversight | All mouse work was approved by the Institutional Animal Care and Use (IACUC) committee at UCI. |

Note that full information on the approval of the study protocol must also be provided in the manuscript.

