## [Peer Review File · Nature Genetics]

Peer Review Information

Manuscript Title: Spatial and single-nucleus transcriptomic analysis of genetic and sporadic forms of Alzheimer's Disease

Corresponding author name(s): Dr Vivek Swarup

Editorial Notes:

Transferred manuscripts This document only contains reviewer comments, rebuttal and decision letters for versions considered at Nature Genetics.

Reviewer Comments & Decisions:

Decision Letter, initial version:

20th Sep 2023

Dear Dr Swarup,

Your Article, "Spatial and single-nucleus transcriptomic analysis of genetic and sporadic forms of Alzheimer's Disease" has now been seen by 3 referees. You will see from their comments copied below that while they find your work of considerable potential interest, they have raised quite substantial concerns that must be addressed. In light of these comments, we cannot accept the manuscript for publication, but would be very interested in considering a revised version that addresses these serious concerns.

We hope you will find the referees' comments useful as you decide how to proceed. If you wish to submit a substantially revised manuscript, please bear in mind that we will be reluctant to approach the referees again in the absence of major revisions.

To guide the scope of the revisions, the editors discuss the referee reports in detail within the team, including with the chief editor, with a view to identifying key priorities that should be addressed in revision and sometimes overruling referee requests that are deemed beyond the scope of the current study. In this case, we invite you to address Reviewers #1 and #2 comments in full. Please do not hesitate to get in touch if you would like to discuss these issues further.

If you choose to revise your manuscript taking into account all reviewer and editor comments, please highlight all changes in the manuscript text file. At this stage we will need you to upload a copy of the manuscript in MS Word .docx or similar editable format.

*2) If you have not done so already please begin to revise your manuscript so that it conforms to our Article format instructions, available here. Refer also to any guidelines provided in this letter.

Please be aware of our guidelines on digital image standards.

[redacted]

If you wish to submit a suitably revised manuscript we would hope to receive it within 6 months. If you cannot send it within this time, please let us know. We will be happy to consider your revision so long as nothing similar has been accepted for publication at Nature Genetics or published elsewhere. Should your manuscript be substantially delayed without notifying us in advance and your article is eventually published, the received date would be that of the revised, not the original, version.

Thank you for the opportunity to review your work.

Sincerely,
Chiara

Chiara Anania, PhD
Associate Editor
Nature Genetics
<https://orcid.org/0000-0003-1549-4157>

Referee expertise:

Referee #1: Neurogenomics and transcriptional regulation

Referee #2: Oligodendrocytes, Myelin, Single-Cell Omics, Epigenomics

Referee #3: Alzheimer's Disease

Reviewers' Comments:

Reviewer #1:

Remarks to the Author:

Miyoshi et al performed spatial-transcriptome (ST), single-nucleus RNA-sequencing (snRNA-seq), and imaging-mass-cytometry (IMC) on human post-mortem AD brain tissues, and 5XFAD – mouse brains at different ages. This manuscript is a follow up on their 2021 Nature Genetics paper applying sn-RNA-seq/snATAC-seq on late-AD brain tissues. In this manuscript, the authors included 3 different AD diagnoses: early-AD, late-AD, and Down-Syndrome (DS) with AD and non-AD controls). The novel datasets were subsequently integrated with previously published datasets, including those from their 2021 paper, and were used to perform a broad range of analyses including: 1) a comparison of genes and ontologies shared and unique to the different diagnoses using a DEG approach, 2) a comparison of gene co-expression modules shared and unique to the different diagnoses using a hdWGCNA approach, 3) comparison between SD-AD sexes, identifying increased expression of HBB in females, and MOBP in males, 4) Integrative analysis of ST and snRNA-seq using CellTrek to further partition cell-subsets according to their predicted locations, 5) changed cell-cell interaction analyses across diagnoses using CellChat, identifying several pathways including Nectin-, ANPTL-, CD99-signaling, 6) proteomics changes in AD, including CST3, CD44, Clusterin, 7) identification of amyloid associated gene signatures. Overall, it is a very interesting paper, with important novel datasets, and many state-of-the-art integrative analyses, but I find it hard to determine what I have exactly learned. There is so much information, but the general narrative, key research questions, and main takeaway messages aren't always exactly clear to me. Maybe there are ways to improve this. Below I will give some more specific points, that could maybe help to get a clearer perspective on the underlying biology.

Individual points:

- To me, it is not really directly obvious why this paper was submitted at Nature Genetics. It seems to

me that the genetic angle is not so obvious. Yes, 5XFAD mice are a genetic model, and DS-AD has a strong genetic component. They also mention some AD GWAS risk genes here and there. But besides, it is very much a transcriptome paper. Maybe the authors could strengthen the relationship with genetics a bit more. For example, the cell-brain-region-subtypes, that they identified using CellTrek, could be used for a heritability enrichment analysis. Maybe microglia subsets in some brain regions show stronger enrichment for AD heritability than other brain regions.

- I think it would be better to include DS brains without AD, as an appropriate control (possibly from younger donors) for the DS-AD group. Maybe the differences between AD-DS and the other types of AD, become smaller if they are compared to DS donors without AD, because right now we cannot partition the effect of AD from the effect of DS.

- I would recommend that they include a workflow-figure. What were the datasets that were generated in this manuscript, and which datasets were from other manuscript, and what analyses were done, which research questions were being addressed by which analysis, and what are the main findings of these analyses, how do the analyses and datasets feed into each other. Also, some sort of graphical representation of the most important findings of this study could really help to better understand the big picture.

- I think that the introduction regarding the (single-cell) genomics studies in AD, should be a bit more elaborate. What have we learned so far from these studies. What is the current state of knowledge regarding the pathophysiology of AD so far, and what does the new paper add to this growing body of research. I think this will also help to anchor the narrative of the paper a bit better. Currently the authors state that it is difficult to holistically understand the findings from single-nucleus studies, and that ST data is important. How then does their ST data give new insight?

- Regarding the multi-scale co-expression analysis, there are so many modules differentially regulated between AD and control. Is it possible to rank order these observations according to their strength and or temporal observations? Can we maybe infer putative causality? Right now, it seems to me that the three AD diagnoses are kind of similar, kind of different, is there anything more to say about that? In what regard are they most similar and in what regard are they most different? Maybe the 5XFAD data can help with that. What are the common mechanisms that drive AD pathogenesis, and which ones are more diagnosis specific, and might occur later on?

- The rationale to study sex DEGs in AD-DS specifically is also not really clear to me. Why are the authors not focusing on sex DEGs in other AD diagnoses (early and late AD)? Also, I wonder whether this analysis is well enough powered? The number of replicates seems quite low (males N=3), and females n=7). I would recommend increasing the sample size. Also, it is not clear to me if the observed sex-changes sexes are merely minor modifications to the large number of AD associated changes that they describe in the previous figure or are they really qualitatively different. Maybe they could plot the effect sizes of sex in figure 4H to effect sizes of AD-DS.

- How does mapping of the cell types to their spatial locations, actually give any new insight in AD? Are there proportional differences between AD and control with regards to these cell-brain-regions-subtypes?

- Regarding the CellChat analysis, why did the authors only focus on AD-DS? And not on the other AD-diagnoses? How does the mapping of the cell types to their spatial location, actually help the cell chat

analysis? Did the authors find pathways that they wouldn't have found without CellTrek integration? Some validation of one of these pathways, for example by Immunohistochemistry or so, or some co-culturing experiment would also be appreciated.

- Regarding the imaging mass cytometry, why did the authors decide to focus on these 23 proteins? Do these proteins have anything to do with the results of the CellChat analysis? How do the IMC clusters relate to the clusters of the other modalities, like hdWGCNA modules? How does this analysis complement the biological insights obtained from the other analyses? This is not clear to me.

- Regarding the amyloid imaging IHC, why do the authors take a DEG approach, instead of correlating the amyloid scores to the hdWGCNA module eigengenes? The number of DEGs in human that correlate to amyloid is much smaller than in mouse. Why is that the case and why are these human and mouse DEG signatures quite different?

Figure 4F. I don't think I fully understand this upset plot. I guess there are both male and female DEGs per layer, but it's not clear to me what bars are male, and which are female. Maybe there is something missing?

Reviewer #2:

Remarks to the Author:

In the article "Spatial and single-nucleus transcriptomic analysis of genetic and sporadic forms of Alzheimer's Disease", Miyoshi and Morabito et al., provide very rich transcriptomic datasets at the single nuclei and spatial levels for different stages of Alzheimer's disease in humans, complemented by similar data from the 5XFAD mouse model of AD. The authors also present unique spatial CyTOF data in their human cohort, particular interesting since beta-amyloid and phosphor-tau antibodies were included. The data analysis is convincing and robust. However, the authors focus more on breadth over depth. Many panels in the main figures are not very informative and difficult to read and interpret, while others should be explored in much more depth. The AD in Down Syndrome patients' samples are unique, and indeed it is an interesting approach to examine it as a genetic form of AD. However, due to the lack of appropriate controls as DS individuals that do not developed AD, it is hard to determine if the conclusions are adequate. I consider this paper an excellent resource that will be used extensively by the AD field, but for that purpose, further and more detailed exploration of the dataset is required, and the authors should provide an online platform (custom made, or via CellxGene or USC Cell Browser platforms) so the reader can fully take advantage of the datasets.

Specific points:

Figure 1 b-c – The spatial clusters correspond to the major areas of the human and mouse brain, which indicates that the clustering is working, but I would expect enhanced resolution with clusters corresponding to areas close to the amyloid plaques and fibrils (considering Figure 8), as observed in the original BayesSpace paper for intratumorally heterogeneity. Have the authors attempted higher levels of clustering? I know this is a challenging task, but this aspect should be addressed.

Figure 1e – There seems to be no difference in coarse cluster structure of the snRNA-Seq data between control and AD, does higher resolution clustering show further differences?

Figure 2a – the finding that most changes occur on the L3-4 layer is interesting, the authors should discuss why this might be the case, in terms of layer composition, for instance.

Figure 2c,d S15 - The authors mention “We also found significant and positive differential expression effect size correlations across spatial and single-nucleus clusters, except in smaller vascular clusters and OPC2”. While this seems to be indeed the case for the spatial data, the R values for the different cell populations are far from 1, regardless of the statistical significance. This might be the case for Ex L5, but it is much less evident for any of the other cell populations. The authors should rephrase their conclusions.

Figure 2f

- The authors mention “For example the gene QKI, previously identified as upregulated in AD41, is upregulated only in upper cortical layers, although it is highly expressed in WM and oligodendrocytes.” The authors need to present quantifications, in particular since examining the images, there seems also to be also an upregulation of QKI in the WM.

- Delineation and labelling of the compartments uncovered in Figure 1b in spatial images would facilitate the interpretation of the data. This applies to all figures.

Figure 3a – This specific figure is overwhelming and difficult to grasp; The authors should explain in the main text in more detail the information that can be retrieved in the dendrograms and the triangular heatmaps.

Figure 3g – The authors mention “The largest set of unique differentially expressed modules were those downregulated in early-stage AD, suggesting that temporally transient systems-level expression changes greatly contribute to the molecular cascade underlying AD progression (Figure 3g).” The figure panel reference is most likely wrong?

Figure 3i

- This is the most relevant and informative panel of this figure but gets buried in the middle of the other panels that are not so informative and should rather be in supplementary figures. It would also be more informative for the reader to have in this main figure representative examples from Fig. S23 that illustrate the difference shown in this panel.

- L3/4 exhibit downregulation of module 5 and M13, that are somehow related, in early AD. Given the relevance of this layer in the previous analysis, how do the author interpret this finding?

Figure 3j – all modules should be shown in the graph

Figure 4

- The authors uncover putative sex differences in AD in DS based on the spatial data. I am concerned that the low number of individuals analysed might affect the power of the analysis. Are these results also observed and validated in the considerable single nuclei RNA-Seq cohorts? And are these differences also observed in mouse and in other stages of human AD?

- One of the biological processes up regulated in males is myelination, with the increase for instance of MOBP. To validate the findings, the authors should perform myelin IHC in other AD in DS cohorts.

Figure 6 – To solidify the CellChat findings, the authors need to:

- use LIANA (Dimitrov et al., Nat Comms 2022)

- It is hard to correlate what the authors describe in the text with the actual figure panels. For instance, the authors mention “However in AD in DS, additional astrocytes, like ASC1 and ASC3 in the cortical upper layers, express ANGPTL4.”. This is not clear at all from the figure. The authors should explain the signalling pathways, which are the receptors, which are the ligands and show them clearly in the panels, so the figures can be better interpreted.

- validate the interactions with IHC, for instance show that cells that are close to each other bear receptor and ligand, respectively, and that this is changed in AD in DS.

Figure 7 – in Panel 7g, j, there are clear differences between Phospho-Tau-AT8 in neurons and astrocytes between AD in DS and late AD. The authors can discuss these findings in light of the transcriptomics similarities and dissimilarities found early in the paper.

Figure 8

- The authors refer that “We found that the distribution of amyloid pathology in the spatial clusters was generally consistent with neuropathological plaque staging, where amyloid pathology is primarily in the cortical upper layers before spreading to the deeper layers (Figure 8c).” Nevertheless, when referring to Figure 3 and differential gene expression for the mouse spatial dataset, the authors had previously commented that “Upregulated genes at 4 months in the thalamus included disease-associated microglia (DAM) genes, like *Cst7*, *Tyrobp*, *Ctsd*, and *Trem2*, suggesting an early response to plaques localized to the thalamus, and with increasing age, we found upregulation of these genes across brain regions.” The authors should comment on these apparently contradictory results.

- Referring to this figure, the authors mention “We found modest yet significant overlaps between some of these gene sets, with a generally higher degree of overlap in the plaque-associated genes”. However, earlier on, referring to Figure 3, the authors also state “module preservation analysis showed that the meta-modules identified in the human dataset were broadly preserved the mouse dataset”. Can the authors comment and elaborate on this apparent contradiction?

Other comments:

- The authors use MAST for DE. Given the recent controversy regarding pseudobulk versus single-cell fitted DEG methods, it would be advisable that the authors perform pseudobulk analysis and compare the results.

- It would be also of interest to compare DEG from the snRNA-Seq and DEGs from the spatial analysis, to deconvolute the differences observed in the spatial data might be attributed to specific cell populations.

- In the introduction, the authors refer 1-10 cell per spot as the resolution of ST. Does this correspond

to the brain and where did these numbers obtained?

- Many heatmaps throughout the paper are set to maximum scales avlog_2FC minor and higher than 1, which is a very limited dynamic range, and does not allow to identify process with large changes.

- While the methods are well detailed, more QC information is needed, the authors need to comment on variability between patients, etc.

Reviewer #3:

Remarks to the Author:

A. Summary of the key results - They identified regional differential gene expression changes shared between AD in the general population and DS, and this is done as the first snRNA-seq study of AD in DS. The finding that specific cellular changes that are conserved between both populations has important implications for clinical trial results in DS likely generalizing to AD that include robust sex difference. The identification of cross-species changes in a single mouse model are confirming discrepancies that may have contributed to poor preclinical translation. The dataset and analytics are a key result, and major contribution to the field.

B. Originality and significance: high

C. Data & methodology: validity of approach, quality of data, quality of presentation is high. N's well-powered and covariates addressed.

D. Appropriate use of statistics and treatment of uncertainties are addressed carefully/thoughtfully.

E. Conclusions: robustness, validity, reliability are high.

F. Suggested improvements: experiments, data for possible revision - none, very comprehensive datasets and analyses.

G. References: appropriate credit to previous work?

Discussion of cross-species analysis and sources of variance should include discussion of genetic background <https://pubmed.ncbi.nlm.nih.gov/30595332/>, which improves alignment of snRNAseq data doi: <https://doi.org/10.1101/2022.04.12.487877>

H. Clarity and context: lucidity of abstract/summary, appropriateness of abstract, introduction and conclusions are well-written and accessible to cross-disciplines.

Author Rebuttal to Initial comments

Point-by-point response to reviewers

We are grateful for the opportunity to present our revised manuscript. After extensive additional analyses and immunofluorescence validation experiments, we believe that we have not only assuaged the reviewers' concerns, but also further added valuable insights into AD biology. Of note, we have:

1. Performed a comprehensive genetic enrichment analysis of AD risk and other traits by applying the scDRS algorithm to our human and mouse ST datasets and our snRNA-seq datasets.
2. Used our snRNA-seq dataset to deconvolve potential cellular origins of the differentially expressed genes found in our spatial transcriptomic dataset. This revealed key changes in glia and vasculature across disease groups and important temporal findings like the downregulation of oligodendrocyte genes in the white matter in the early-stage AD group.
3. Added several new analyses to extensively examine the role of sex in the AD human and mouse transcriptome, beyond just the sex differences in AD in DS as we presented originally.
4. Validated the dysregulation of cell-cell signaling pathways with several immunofluorescence experiments.

Additionally, based on the commentary of the reviewers we have updated the abstract to discuss the motivation and some major takeaways of our study more clearly. Some parts of the manuscript have also been expanded to add further discussion that some of the reviewers were interested such as the changes in cortical layers 3/4, which may be implicated in amyloid deposition and cognitive decline. In the original manuscript we primarily highlighted the similarities of AD in DS and sporadic AD, but our updated manuscript also highlights some of the differences, for example in our co-expression analysis figure. We have also refined the focus throughout the abstract, introduction, and discussion sections to provide context of our findings in the scope of this research field. Overall, we believe that by addressing the reviewer's comments we were able to strengthen our study.

Please see below for our point-by-point response.

Reviewers' Comments:

Reviewer #1:

Remarks to the Author:

Miyoshi et al performed spatial-transcriptome (ST), single-nucleus RNA-sequencing (snRNA-seq), and imaging-mass-cytometry (IMC) on human post-mortem AD brain tissues, and 5XFAD – mouse brains at different ages. This manuscript is a follow up on their 2021 Nature Genetics paper applying sn-RNA-seq/snATAC-seq on late-AD brain tissues. In this manuscript, the authors included 3 different AD diagnoses: early-AD, late-AD, and Down-Syndrome (DS) with AD and non-AD controls). The novel datasets were subsequently integrated with previously published datasets, including those from their 2021 paper, and were used to perform a broad range of analyses including: 1) a comparison of genes and ontologies shared and unique to the different diagnoses using a DEG approach, 2) a comparison of gene co-expression modules shared and unique to the different diagnoses using a hdWGCNA approach, 3) comparison between SD-AD sexes, identifying increased expression of HBB in females, and MOBP in males, 4) Integrative analysis of ST and snRNA-seq using CellTrek to further partition cell-subsets according to their predicted locations, 5) changed cell-cell interaction analyses across diagnoses using CellChat, identifying several pathways including Nectin-, ANPTL-, CD99-signaling, 6) proteomics changes in AD, including CST3, CD44, Clusterin, 7) identification of amyloid associated gene signatures. Overall, it is a very interesting paper, with important novel datasets, and many state-of-the-art integrative analyses, but I find it hard to determine what I have exactly learned. There is so much information, but the general narrative, key research questions, and main takeaway messages aren't always exactly clear to me. Maybe there are ways to improve this. Below I will give some more specific points, that could maybe help to get a clearer perspective on the underlying biology.

Point-by-point response to reviewers

Individual points:

1. To me, it is not really directly obvious why this paper was submitted at Nature Genetics. It seems to me that the genetic angle is not so obvious. Yes, 5XFAD mice are a genetic model, and DS-AD has a strong genetic component. They also mention some AD GWAS risk genes here and there. But besides, it is very much a transcriptome paper. Maybe the authors could strengthen the relationship with genetics a bit more. For example, the cell-brain-region-subtypes, that they identified using CellTrek, could be used for a heritability enrichment analysis. Maybe microglia subsets in some brain regions show stronger enrichment for AD heritability than other brain regions.

We appreciate the reviewer for their suggestion to strengthen the genetics aspect of this paper to further align this study with the interests of the Nature Genetics readership. As the reviewer suggested, in our updated manuscript we include new results from heritability enrichment analysis. We used scDRS, a recently published approach described by Zhang *et al.* in their 2022 Nature Genetics paper, to link our transcriptomic datasets with polygenic disease risk at single-cell resolution. Importantly, this approach computes cell-level (or spot-level) statistics and enrichment scores as well as at the group level (cell-population or spatial region). We performed scDRS analysis using the set of 74 traits included with the scDRS package in our ST dataset, our snRNA-seq dataset, and the three published snRNA-seq datasets used in this study. We also performed scDRS analysis on the mouse ST dataset using the set of 22 mouse traits, where the genes have been converted from mouse to human orthologs. We highlight the enrichment results for AD in Figure 3 and we report the results for all of the other traits in the Supplementary Figures S28-S33. In line with our expectations from previous studies, AD polygenic risk was enriched in both microglia clusters MG1 and MG2 in the snRNA-seq dataset in controls, early-stage AD, late-stage AD, and AD in DS (Fig. 3l). In the human ST dataset, we only found significant group-level enrichment in the AD in DS group in the L1, L3/4, and WM clusters (Fig. 3i). In the mouse ST dataset we found significant enrichment only in the 5xFAD mice, and that more clusters were enriched in more advanced ages (Fig. 3k). We also correlated the spot-level enrichment scores with our co-expression module expression levels to identify potential relationships between the co-expression modules and AD polygenic risk. Summarizing the correlation results across the human and mouse ST datasets and the snRNA-seq datasets, we found that module M11, which was expressed in glial cells and was enriched for inflammatory response genes, showed evidence of association with AD polygenic risk. We describe these results in a new section of the manuscript titled "Inflammation gene module correlated with AD poly-genic risk in human AD and 5xFAD mice", and we believe that this new analysis strengthens the genetics component of our study.

2. I think it would be better to include DS brains without AD, as an appropriate control (possibly from younger donors) for the DS-AD group. Maybe the differences between AD-DS and the other types of AD, become smaller if they are compared to DS donors without AD, because right now we cannot partition the effect of AD from the effect of DS.

We appreciate and agree with the reviewer's suggestion to include DS samples without AD pathology. We originally wanted to include these samples in the study. Unfortunately, despite expanding our search for samples to the NIH NeuroBioBank, we were limited by not only the availability of high quality fresh frozen postmortem samples (critical for RNA analyses), but also the overall smaller population size. DS individuals without AD pathology at ages 40+ are rare (see for review Lott and Head 2019 *Nat Rev Neurol*). We also considered including younger DS patients; however, it would be difficult to interpret and distinguish disease-related changes from age-related changes. Aging is known to

Point-by-point response to reviewers

influence gene expression in the brain/human prefrontal cortex (Lu et al. 2004 *Nature*), and an additional needed control would be younger individuals without DS to identify DS-specific changes. Therefore, for the current study, we decided to not pursue these further complicated study designs and instead prioritize increasing our sample numbers per group.

3. I would recommend that they include a workflow-figure. What were the datasets that were generated in this manuscript, and which datasets were from other manuscript, and what analyses were done, which research questions were being addressed by which analysis, and what are the main findings of these analyses, how do the analyses and datasets feed into each other. Also, some sort of graphical representation of the most important findings of this study could really help to better understand the big picture.

We agree with the reviewer that a workflow figure and a graphical representation of some of the major findings could improve the interpretability of the study. In the updated manuscript we have added a summary figure in Fig. 8n showing the datasets generated, the major data analysis steps, and some of the selected conclusions from each of these analysis steps. We anticipate that the graphical summary will aid readers in grasping the overarching themes and key points of the article.

4. I think that the introduction regarding the (single-cell) genomics studies in AD, should be a bit more elaborate. What have we learned so far from these studies. What is the current state of knowledge regarding the pathophysiology of AD so far, and what does the new paper add to this growing body of research. I think this will also help to anchor the narrative of the paper a bit better. Currently the authors state that it is difficult to holistically understand the findings from single-nucleus studies, and that ST data is important. How then does their ST data give new insight?

We thank the reviewer for bringing this to our attention. We have added more details about several key single-cell genomic studies in AD and in mouse models of AD to provide further context for our work. To date there have been numerous studies in this area and it is difficult to succinctly summarize all of the main findings within the physical confines of an introduction section. Therefore, we added a new citation to a recent review paper which summarizes the state of single-cell profiling in AD (Murdock and Tsai *Nature Neuroscience* 2023)

Our ST data provides several new insights in AD biology by comprehensively defining the regional transcriptomic changes in AD. Previous studies with bulk-tissue and snRNA-seq have been limited by the technical challenges of separating tissue samples into different regions. We have revealed intriguing changes in L3/4 that may be implicated in amyloid deposition and cognitive decline, and altogether our regional analyses provide insights into AD regional vulnerability. Additionally, we have stratified snRNA-seq clusters by brain region and gained greater detail into the cell-cell signaling pathways dysregulated in AD than with snRNA-seq alone. Further, we identified both plaque and fibril-associated genes. Previous studies have not been able to comprehensively define the changes proximal to different amyloid conformations. Finally, we believe our identification and characterization of the meta-module M11 provides great insights into AD pathophysiology by further refining our knowledge of AD glia.

5. Regarding the multi-scale co-expression analysis, there are so many modules differentially regulated between AD and control. Is it possible to rank order these observations according to their strength and or temporal observations? Can we maybe infer putative causality? Right now, it seems to me that the three AD diagnoses are kind of similar, kind of different, is there anything more to say about that? In what regard are they most similar and in what regard are they most different? Maybe the 5XFAD data can help with that. What

Point-by-point response to reviewers

are the common mechanisms that drive AD pathogenesis, and which ones are more diagnosis specific, and might occur later on?

We thank the reviewer for their comment and agree that additional analysis could clarify our co-expression analysis, especially with regards to highlighting the modules that are important in disease. As suggested by the reviewer, in our updated manuscript we now include a new panel (Fig. 3c) to rank the co-expression modules by those which are exclusively up- or down-regulated in each of the 3 disease groups. We also include a similar figure showing the modules that are shared across all disease groups, and those which are conflicting (up in one condition but down in another and vice versa) in the new Supplementary Figure 26. We also performed GO term enrichment analysis on the hub genes from these sets of modules (specific to each disease status and shared), which for example showed enrichment of amyloid-beta formation genes among the modules that were up-regulated in all 3 disease groups (Supplementary Fig. 26). In the new version of the manuscript, Figure 3 highlights the co-expression similarities between disease groups in panel b as well as the differences in panel c.

We were intrigued by the reviewer's suggestion of inferring putative causality using our co-expression network. We contemplated conducting a causal mediation analysis incorporating multiple mediators to model the disease status or amyloid burden, using the co-expression modules as intermediaries. However, while this approach is conceptually intriguing and could yield significant insights, it would likely necessitate a substantially larger sample size than what we currently possess to reliably infer causality. Furthermore, our study primarily concentrates on amyloid, due to the use of the 5xFAD mouse model, and a causal mediation analysis might be more effective with simultaneous consideration of both tau and amyloid.

6. The rationale to study sex DEGs in AD-DS specifically is also not really clear to me. Why are the authors not focusing on sex DEGs in other AD diagnoses (early and late AD)? Also, I wonder whether this analysis is well enough powered? The number of replicates seems quite low (males N=3), and females n=7). I would recommend increasing the sample size. Also, it is not clear to me if the observed sex-changes sexes are merely minor modifications to the large number of AD associated changes that they describe in the previous figure or are they really qualitatively different. Maybe they could plot the effect sizes of sex in figure 4H to effect sizes of AD-DS.

We thank the reviewer for their comment and have made adjustments to this section of the manuscript to reflect their concerns. In the revised manuscript we now include new analysis of sex DEGs in the other AD disease groups (snRNA-seq and ST datasets) and in the 5xFAD mouse ST dataset. Figure 4 primarily focuses on AD in DS, but we have expanded the co-expression network differences to include early-stage and late-stage AD as part of the main figure (Fig. 4h-j). As suggested, we plotted the effect sizes of sex versus the effect sizes of disease for the co-expression modules in Fig. 4j, and we have done the same for the DEGs in the Supplementary Figures S46-49. This analysis showed transcriptomic differences between the sexes in each disease condition that were outside of the set of disease DEGs. Based on the reviewer's previous suggestion about deconvoluting DEGs by cell population, we also showed the DEG cell deconvolution results for the sex DEGs in the updated manuscript. Ideally, we would include a larger number of replicates for this analysis for more robust findings, and we acknowledge that this is a limitation of the current study in our discussion section.

7. How does mapping of the cell types to their spatial locations, actually give any new insight in AD? Are there proportional differences between AD and control with regards to these cell-brain-regions-subtypes?

Point-by-point response to reviewers

A key part of our analytical approach was the integration of our ST and snRNA-seq datasets. For each pair of ST and snRNA-seq samples, we integrated the transcriptomic information between these two modalities and predicted the spatial coordinates of each cell in the snRNA-seq dataset. We believe that this analysis provides a critical supporting role to further substantiate other findings throughout the manuscript. For example, we can see that astrocyte ASC2 is more common in the white matter clusters than in the cortical layers 2-6b compared to ASC1, and these nuances are not possible to observe in either the ST nor snRNA-seq datasets on their own. Furthermore, this analysis provides setup for our cell-cell communication analysis where we leveraged this spatial information. Based on the reviewer's comment, we also performed differential cell abundance between AD in DS and controls using the Milo algorithm, and we show these results in Fig. 6e. This analysis informs us about cell populations which are enriched or depleted in one condition compared to another. For example here we can see that in general there is a relatively greater abundance of inhibitory neurons in the upper cortical layers in AD in DS, meaning that the healthy cell composition of cortical inhibitory neuron compartment is likely altered in disease. This new analysis is directly relevant and insightful about cellular composition changes in disease, while we also believe that this analysis provides additional context for our other findings in these cell populations and these spatial regions. We also described a similar analysis without the spatial information in the updated manuscript and in the response to Reviewer 2.

8. Regarding the CellChat analysis, why did the authors only focus on AD-DS? And not on the other AD-diagnoses? How does the mapping of the cell types to their spatial location, actually help the cell chat analysis? Some validation of one of these pathways, for example by Immunohistochemistry or so, or some co-culturing experiment would also be appreciated.

We appreciate the reviewer's comment regarding the CellChat analysis. For the cell-cell communication analysis, we specifically focused on the snRNA-seq dataset that we generated in this study for a number of reasons. In particular, CellChat and other cell-cell communication tools do not have any explicit methods to "integrate" distinct datasets or batches. We used the Parse Biosciences combinatorial barcoding approach for our snRNA-seq experiments in this study, while the 3 previous datasets included in this study were from the 10X genomics microfluidic droplet-based approach. Additionally, these datasets were generated by different groups at different times, which can all contribute to technical batch effects. We are able to account for these batch effects in many of our analyses (for example as a covariate in DEG mixed effects models or in our scANVI cell clustering analysis), but we believe that CellChat could introduce unreliable conclusions if we attempted to include the other datasets since it does not consider batches in its inference procedure. While we could have performed cell-cell communication analysis separately for each dataset, we are currently lacking the tools required to perform a rigorous meta-analysis of multiple communication networks from different datasets, and developing such methodology would be outside the scope of this study.

With CellTrek, we were able to provide additional annotations for each cell population for cells that were mapped to upper cortical, lower cortical, or white matter regions. Indeed, by including this information in our CellChat analysis we were able to identify signaling relationships that would not have been found without the spatial information. While the overarching conclusions of this section may have been similar in the absence of spatial information, we believe that our analysis provides additional nuance that is important for understanding the whole story. For example, in the ANGPTL network analysis we found that while overall signaling was increased in disease, there were signaling interactions between astrocytes and lower cortical inhibitory neurons that were absent in disease but present in control, and therefore this allows us to understand more precisely where these changes are occurring.

Point-by-point response to reviewers

As suggested by the reviewer, we have validated some of the CellChat results for the three pathways that we highlight in the manuscript using immunofluorescence in human postmortem cortical tissue from AD in DS, AD, and cognitively normal control donors. We conducted thorough immunohistochemistry validations for various signaling molecules (NECTIN, ANGPTL and CD99), confirming an upregulation of ANGPTL4 signaling in AD in DS (Fig. 7m). We also discovered downregulation of CD99 (Fig S57f-g) and NECTIN (Fig. 7g) in both AD in DS and late-stage AD.

9. Regarding the imaging mass cytometry, why did the authors decide to focus on these 23 proteins? Do these proteins have anything to do with the results of the CellChat analysis? How do the IMC clusters relate to the clusters of the other modalities, like hdWGCNA modules? How does this analysis complement the biological insights obtained from the other analyses? This is not clear to me.

In the initial manuscript, the analysis of the Imaging Mass Cytometry (IMC) dataset was presented immediately after the cell signaling analysis. This order might lead readers to anticipate a connection between these two sections. However, our IMC panel did not consist of the genes identified in our cell signaling analysis. We had selected proteins that could serve as markers for specific cell types, in addition to disease-related genes, including those identified in our differential gene expression (DEG) and co-expression network analyses. Also, notably, there were limitations in the proteins that can be examined with IMC due to the availability of antibodies and required antibody formulation for successful metal conjugation. In order to remedy this confusion, we moved this section to earlier in the manuscript to follow our sex DEG figure. In the manuscript we have discussed that some of these proteins were M11 hub genes as well as some spatial/cell-type DEGs in order to link this section to our other analysis. We believe the IMC data provides additional evidence for the roles of these genes/proteins in AD biology.

10. Regarding the amyloid imaging IHC, why do the authors take a DEG approach, instead of correlating the amyloid scores to the hdWGCNA module eigengenes? The number of DEGs in human that correlate to amyloid is much smaller than in mouse. Why is that the case and why are these human and mouse DEG signatures quite different?

The reviewer brings up an interesting point about our methodology behind identifying amyloid-associated gene expression changes. In our study, we integrated our ST datasets with fluorescent imaging to identify amyloid hotspots, and then we identified gene expression changes proximal to these hotspots via differential expression testing. The reviewer suggests that as an alternative we could identify which co-expression modules were correlated with these amyloid hotspots. We note that a similar approach was taken in an APP mouse model in the Chen *et al.* 2020 Cell paper, where they identified a co-expression module that they named “plaque-induced genes” (PIGs) and was correlated with amyloid burden. We described a similar analysis that we performed in our mouse ST dataset, where we identified a very similar module (SM6) to the PIGs module, and we did show that this module was indeed correlated with amyloid hotspots (Fig. S65). Based on the reviewer’s suggestion, in our revised manuscript we have also performed a similar correlation analysis in the human dataset, and in the human co-expression modules projected into the mouse dataset (Fig. S67). This analysis only revealed modest to weak correlations between the module expression levels and amyloid hotspot scores ($R < 0.25$). In the human dataset, the inflammatory module M11 was weakly correlated with amyloid in the middle cortical layer clusters L3-5 and L3/4. The module with the strongest correlation, which was still relatively weak, was M15, which includes many mitochondrial genes as hub genes. Module M11 also was weakly positively correlated with amyloid in the mouse dataset. Based on this

Point-by-point response to reviewers

comparison of the human ST co-expression networks with the amyloid scores, it does not appear that there is an obvious gene module which is uniquely expressed proximal to amyloid plaques. This is in opposition to what we saw in the mouse dataset, where module SM6 was more strongly correlated with amyloid hotspots (Fig. 65).

The reviewer also points out the discrepancies between the number of amyloid associated genes identified in humans and mice. We believe that this is easily explained by the differences in size between mouse and human brains. The Visium ST slides can fit a mouse brain hemisphere which includes several brain regions together, while a relatively smaller amount of the human brain can be profiled at a time. We performed separate tests for amyloid-associated genes for each of the different regions in the mouse brain dataset, which ultimately resulted in a larger set of genes identified than in the human dataset where we only had one region. We anticipate that in the future with larger human datasets encompassing multiple regions, additional amyloid-associated genes would be identified using the same approach.

There are several potential reasons why these human and mouse amyloid-associated genes would be different. We note that we did use the same methodology for detecting these genes in both species so these comparisons would be as fair as possible. First, we expect that data from any model organism would be a lot “cleaner” than in postmortem human tissue. Meaning that we have a much greater degree of control over the mice, in terms of their genotypes, ages, etc. This leads to much greater uniformity in the mouse dataset compared to the human dataset, so it is likely easier to detect the disease-relevant signal from a model than in primary human tissue. Furthermore, while 5xFAD is a genetic model based on familial AD, it is primarily a model of amyloidosis; it is not a truly faithful model of AD as a whole but merely a key aspect of it. The progressive dynamics of disease with respect to pathologies and cellular changes in the human population, whether it is the sporadic or genetic case, likely differs considerably from that in the 5xFAD mouse model. Thus, it makes sense that the genes that are expressing proximal to amyloid plaques in mouse would be different than those in humans. Intrinsic species related differences in the pathways that are responding to amyloid plaques could also be at play. We also note that our mouse ST samples are generally much more uniform relative to the human samples, so we expect that there is a higher degree of variability among the human samples. These different issues ultimately compound, where we arrive at these different sets of amyloid-associated genes across species. This highlights a need both for better mouse models, and for additional studies using larger scale human cohorts and using more precise high-resolution spatial technologies. We include some of this commentary in the discussion section of the updated manuscript.

11. Figure 4F. I don't think I fully understand this upset plot. I guess there are both male and female DEGs per layer, but it's not clear to me what bars are male, and which are female. Maybe there is something missing?

We thank the reviewer for pointing out the problem with Figure 4F (4g in the updated manuscript), the labels for which set corresponded to female or male were missing in the original plot but we have included these labels in the revised manuscript.

Reviewer #2:

Remarks to the Author:

In the article “Spatial and single-nucleus transcriptomic analysis of genetic and sporadic forms of Alzheimer’s Disease”, Miyoshi and Morabito et al., provide very rich transcriptomic datasets at the single nuclei and spatial levels for different stages of Alzheimer’s disease in humans, complemented by similar data from the 5XFAD mouse model of AD. The authors also present unique spatial CyTOF data in their human cohort, particular

Point-by-point response to reviewers

interesting since beta-amyloid and phosphor-tau antibodies were included. The data analysis is convincing and robust. However, the authors focus more on breadth over depth. Many panels in the main figures are not very informative and difficult to read and interpret, while others should be explored in much more depth. The AD in Down Syndrome patients' samples are unique, and indeed it is an interesting approach to examine it as a genetic form of AD. However, due to the lack of appropriate controls as DS individuals that do not developed AD, it is hard to determine if the conclusions are adequate. I consider this paper an excellent resource that will be used extensively by the AD field, but for that purpose, further and more detailed exploration of the dataset is required, and the authors should provide an online platform (custom made, or via CellxGene or USC Cell Browser platforms) so the reader can fully take advantage of the datasets.

We are grateful for the reviewer's keen interest in our study and their insightful critiques. We concur that incorporating an interactive data browser would significantly enhance the accessibility and applicability of our dataset within the Alzheimer's disease research community. To this end, we are actively working to make our dataset available for exploration on the CellxGene platform, a development that will be reflected in the finalized version of our manuscript. Additionally, for more thorough utilization of our datasets, we are in the process of creating an extensive README document on our study's GitHub repository. This document is intended to function as a detailed tutorial for those wishing to interact programmatically with our datasets.

Specific points:

1. Figure 1 b-c – The spatial clusters correspond to the major areas of the human and mouse brain, which indicates that the clustering is working, but I would expect enhanced resolution with clusters corresponding to areas close to the amyloid plaques and fibrils (considering Figure 8), as observed in the original BayesSpace paper for intratumorally heterogeneity. Have the authors attempted higher levels of clustering? I know this is a challenging task, but this aspect should be addressed.

We appreciate the reviewer's consideration of the importance in selecting an appropriate number of clusters. This was an aspect that we carefully deliberated. The original BayesSpace paper shows 7 clusters in a human prefrontal cortex sample from Maynard et al. 2021 *Nat Neurosci* with remarkable similarity to the manual annotation/ground truth. 7 clusters makes logical sense considering the cortex's laminar structure; however, we also were wary of missing insights by restricting our clusters to 7. Therefore, in the human dataset, we examined the distribution of the clusters and compared them with the expression of known layer marker genes across all 39 samples, roughly annotating each cluster by the cortical layers and white matter, for 5, 6, 7, 8, 9, and 10 clusters. Surprisingly, we found 2-3 white matter clusters with 7 or more clusters. We selected 9 clusters as they included all of the cortical layers as well as the 3 white matter clusters, and we observed greater consistency across the samples compared to 10 clusters. Although we have not attempted more than 10 clusters, we found marginal benefits from increasing the number of clusters in our original analysis, and there does not appear to be any clusters consistently localized to amyloid pathology (Response Figure 1). Furthermore, we likewise tested different cluster numbers in the mouse dataset (10, 15, and 20). Increasing clusters provided finer segmentation of brain regions, but we did not find any clusters localizing to amyloid pathology, despite the much more homogeneous and predictable amyloid distribution in the 5XFAD. Instead, we found a sparsely distributed cluster (scattered single spots) resembling erythrocytes with 15 and 20 clusters, suggesting that the clustering largely identifies distinct region and cell-type differences across the samples. We selected 15 clusters as a compromise between resolution and interpretability.

Point-by-point response to reviewers

Response Figure 1: Spatial clustering at different resolutions.

a-c) Representative plots from the different clinical groups showing the spatial distributions of 7 **(a)**, 8 **(b)**, and 10 **(c)** BayesSpace clusters. Colors do not represent the same clusters across the different clustering. **d)** Plots showing the spatial distributions of amyloid in the same samples as **(a-c)**. Left: Raw Amylo-glo score, right: Amylo-glo Gi* hotspot analysis used in our amyloid DE analysis.

2. Figure 1e – There seems to be no difference in coarse cluster structure of the snRNA-Seq data between control and AD, does higher resolution clustering show further differences?

Based on the reviewer's comment, there may be some confusion regarding the information shown in this Figure. In Figure 1e, we show a UMAP plot depicting our integrated snRNA-seq dataset with each data point colored by its annotated cell population. This particular UMAP is a 2 dimensional projection of the 30 dimensional scANVI representation of the integrated dataset. While there may be interesting biological implications behind the differences in cluster proportions of each disease group, this kind of analysis cannot be done with UMAP alone. Projecting high-dimensional datasets into 2 dimensions is useful for coarse-grain data visualization, but distortions are inevitably introduced in all UMAPs which can lead to meaningless conclusions when comparing UMAPs side by side. Our intention behind showing the UMAP in Figure 1e is simply to show an overview of our clustering results and the different cell populations captured in our analysis. Aside from this, we think that the reviewer was alluding to an analysis of the differences between cell population compositions with regards to the different disease groups. Based on this commentary, we performed differential cell population abundance analysis between controls and disease groups for each of the snRNA-seq datasets using the algorithm Milo (Dann et al. Nature Biotechnology 2021). Briefly, Milo uses the low-dimensional representation of the dataset (30-dimensional scANVI representation in our case) to construct "cell neighborhoods", groups of transcriptomically similar cells. Milo then counts the number of cells from each condition within these neighborhoods and performs a statistical test to infer the differential abundance between these conditions in these cellular neighborhoods. This analysis tells us which cell populations were enriched

Point-by-point response to reviewers

or depleted in disease, and it can inform us about transcriptomic variability across conditions within each cell population. We report these results in Supplementary Figure 6, and we added some commentary about this analysis in the main text. This analysis did highlight interesting differences in cellular composition, for example the enrichment of microglia cluster MG2 and astrocytic cluster ASC1 in the AD in DS compared to controls, as well as key differences in vascular subtypes in disease.

3. Figure 2a – the finding that most changes occur on the L3-4 layer is interesting, the authors should discuss why this might be the case, in terms of layer composition, for instance.

We concur with the reviewer that these pronounced changes observed in the L3-4 cortical layers are very interesting. We noted in the results that these gain further significance when considering the typical neuropathological hallmarks of AD, such as neuritic plaques and neurofibrillary tangles, which are preferentially found at layers III and V. Additionally, given that cortical layer IV is one of the primary input layers, we reason that these transcriptomic changes found here potentially associated with the cognitive symptoms of the early stages of disease and the distinct vulnerabilities of these layers. Taking the reviewer's response into account, we have included additional lines further discussing implications of the L3/4 changes, including findings from our additional deconvolution analysis and further extending our discussion of our L3/4 modules.

4. Figure 2c,d S15 - The authors mention “We also found significant and positive differential expression effect size correlations across spatial and single-nucleus clusters, except in smaller vascular clusters and OPC2”. While this seems to be indeed the case for the spatial data, the R values for the different cell populations are far from 1, regardless of the statistical significance. This might be the case for Ex L5, but it is much less evident for any of the other cell populations. The authors should rephrase their conclusions.

We thank the reviewer for their comment and we agree that the original phrasing of this section is potentially misleading. In the revised manuscript have adjusted the phrasing of this section to more carefully and accurately describe the correlation relationships: “We also found high correlations ($R > 0.5$) of DE effect sizes in most of the spatial clusters and modest correlations ($R > 0.2$) in most of the single-nucleus clusters (Figs. 2e-f, S16, S17). Notably, these trends were stronger in grey matter clusters, compared to WM clusters, and this was matched in the single-nucleus data, where the correlations were also stronger in neuronal clusters relative to the glial clusters. The cell populations with the weakest correlations across disease conditions included OPC2 and several of the microglial, vascular, and astrocytic clusters.

5. Figure 2f - The authors mention “For example the gene QKI, previously identified as upregulated in AD41, is upregulated only in upper cortical layers, although it is highly expressed in WM and oligodendrocytes.” The authors need to present quantifications, in particular since examining the images, there seems also to be also an upregulation of QKI in the WM.

We thank the reviewer for this suggestion and have added information on average $\log_2(\text{fold-change})$ and adjusted p-value and the specific clusters identified. The full DEG results are provided in Supplementary Tables 8 and 9. We did not find QKI significantly upregulated (adjusted p-value < 0.05) in the white matter. We also note that in order to make room for another panel containing new analysis requested by the reviewer (cellular deconvolution of ST DEGs), we moved the figure showing these gene expression feature plots to the Supplementary.

Point-by-point response to reviewers

6. Delineation and labelling of the compartments uncovered in Figure 1b in spatial images would facilitate the interpretation of the data. This applies to all figures.

The reviewer suggests updating the spatial images, like those shown in Figure 1b, to include manual annotations of the different cortical regions and white matter to improve interpretability. We originally considered presenting our data in the way that the reviewer suggested, but have ultimately decided against it for a number of reasons. We believe that the color scheme denoting each region already makes this point clear, and there are potentially some issues that can arise when manually annotating this spatial data. Indeed, some previous studies using ST opted for expert-guided manual annotations instead of data-driven clustering. Given that we have 39 samples in humans and 80 in mice, we decided to use the BayesSpace algorithm for clustering our dataset and we plotted the spatial samples with each spot colored by their cluster assignment. We used different shades of pink and red to denote grey matter regions and shades of blue for white matter. Manual annotations could be useful but we run the risk of inconsistency given the large number of samples in our study. Since this is a data-driven clustering in our study, some of the regional boundaries are imperfect and in some cases it could be difficult to properly annotate.

7. Figure 3a – This specific figure is overwhelming and difficult to grasp; The authors should explain in the main text in more detail the information that can be retrieved in the dendrograms and the triangular heatmaps.

We agree with the reviewer that this figure may be difficult to grasp at first glance since it has a number of subcomponents. The figure legend provided aims to guide the reader on how to interpret each layer of this plot. We updated the main text in this section to point to specific elements of this panel in the description, which should clarify the reviewer's concern. We also updated the figure legend to specifically mention that the triangular heatmaps are based on the module similarity in terms of the gene sets of each module as well as the module expression levels.

8. Figure 3g – The authors mention “The largest set of unique differentially expressed modules were those downregulated in early-stage AD, suggesting that temporally transient systems-level expression changes greatly contribute to the molecular cascade underlying AD progression (Figure 3g).” The figure panel reference is most likely wrong?

We thank the reviewer for pointing out this mistake. We intended to refer to the upset plot showing overlapping sets of differentially expressed modules between the different conditions. We have corrected this reference in the updated manuscript, however we note that this particular panel was moved to the supplementary figures (Fig. S26) in order to make room for the genetic enrichment analysis panels in Figure 3.

9. Figure 3i This is the most relevant and informative panel of this figure but gets buried in the middle of the other panels that are not so informative and should rather be in supplementary figures. It would also be more informative for the reader to have in this main figure representative examples from Fig. S23 that illustrate the difference shown in this panel.

We appreciate the reviewer's comment regarding the flow and layout of Figure 3. The reviewer points out that the heatmap showing the results of the differential module eigengene test is in the middle of the figure so it could get lost among the other panels. Due to the constraints of the physical space allotted by the figure, and considering that we added several new panels into the updated version of this figure based on the comments of Reviewer 1, this heatmap remained in a similar placement within the

Point-by-point response to reviewers

updated figure. We hope that the reader would inspect each panel in order such that this particular panel would not be lost among the others. Additionally, the reviewer suggests moving some of the other panels in this figure into the supplementary but we kept all of these panels in the updated version since we believe that they are important for understanding and interpreting our co-expression network analysis. The reviewer also suggests moving some of the panels from Fig. S23, which depicted the module eigengenes of the 15 meta-modules in representative samples of each disease condition in order to illustrate the differences in expression levels between conditions. The heatmap in Figure 3f showing the differential module eigengene results more precisely shows these differences than these representative spatial plots would, and it takes up less physical space in the figure. Furthermore, this supplementary figure is still there for the readers who wish to inspect the data in these representative samples.

10. L3/4 exhibit downregulation of module 5 and M13, that are somehow related, in early AD. Given the relevance of this layer in the previous analysis, how do the author interpret this finding?

The downregulation of modules M5 and M13 in the L3/4 cortical layers during the early stages of Alzheimer's Disease (AD) is particularly significant, given the pivotal role these layers play in AD pathology. Understanding this finding requires an examination of the gene composition and biological functions of these modules. Typically, these modules might encompass genes integral to crucial neuronal functions, such as synaptic transmission, neuroplasticity, or cellular metabolism, which are vital for the proper functioning of cortical networks. The L3/4 layers are central to cognitive processes and are known for their dense synaptic connections and susceptibility to early AD pathologies like amyloid plaque accumulation and tau pathology. Therefore, the observed downregulation could reflect the initial molecular responses to pathological changes in AD, potentially impacting synaptic function and neuronal communication. This alteration might contribute to the early cognitive symptoms of AD, such as memory deficits and impaired information processing, emphasizing the need to further explore these modules' specific roles in the disease's progression.

11. Figure 3j – all modules should be shown in the graph

We appreciate the reviewer's comment and we point out that we have included a graph containing all of the modules in the Supplementary Figures (Fig S27).

12. Figure 4 The authors uncover putative sex differences in AD in DS based on the spatial data. I am concerned that the low number of individuals analysed might affect the power of the analysis. Are these results also observed and validated in the considerable single nuclei RNA-Seq cohorts? And are these differences also observed in mouse and in other stages of human AD?

We understand the reviewer's concerns with our sex DEG analysis, and we point out that many of these concerns were also shared with reviewer 1. We believe that all of these issues were addressed in the updated manuscript as we have described above in our response to reviewer 1.

13. One of the biological processes up regulated in males is myelination, with the increase for instance of MOBP. To validate the findings, the authors should perform myelin IHC in other AD in DS cohorts.

We attempted to validate MOBP, but were unable to achieve satisfactory staining with the available antibodies in our samples. Instead, we provide in Figure 4 immunofluorescence validation for upregulation of C1QB in the white matter of AD in DS females. However, due to limited AD in DS

Point-by-point response to reviewers

sample availability, this experiment included only 2 females and 2 males. We also focused on conducting extensive immunohistochemistry validations for several signaling molecules, including NECTIN, ANGPTL, and CD99.

14. Figure 6 – To solidify the CellChat findings, the authors need to:

- use LIANA (Dimitrov et al., Nat Comms 2022)

We thank the reviewer for their suggestion to strengthen our cell-cell communication analysis by comparing it to another method. LIANA is another cell-cell communication analysis pipeline which uses several different approaches and ligand-receptor databases to arrive at a consensus results from the different approaches. For the revised manuscript, we ran LIANA with default parameters on our control and our AD in DS cohorts in our snRNA-seq data, mirroring our CellChat analysis. We then correlated the number of inferred signaling relationships between each cell population between CellChat and LIANA (Supplementary Figures S53,S54), and our results showed that these pipelines generally agreed with one another, with LIANA frequently identifying more significant interactions than CellChat. LIANA does not provide any methodology for statistical comparisons of cell signaling networks akin to CellChat, so we were not able to compare the network differences between disease and controls across these methods.

15. It is hard to correlate what the authors describe in the text with the actual figure panels. For instance, the authors mention “However in AD in DS, additional astrocytes, like ASC1 and ASC3 in the cortical upper layers, express ANGPTL4.”. This is not clear at all from the figure. The authors should explain the signalling pathways, which are the receptors, which are the ligands and show them clearly in the panels, so the figures can be better interpreted.

We appreciate the reviewer’s comment and we have carefully gone through the manuscript to ensure that figure panels are properly referenced. Regarding the specific comment about the signaling pathways, we have rephrased this sentence about ANGPTL4 to refer to the links in the network plot rather than the expression levels. Additionally, we did provide new panels to show the expression level of the ligands and receptors with significant interactions based on CellChat for NECTIN, ANGPTL, and CD99 signaling as a dot plot in Figure 7. We believe that these changes have clarified the specific point and we hope that the figure panel references are clear throughout the revised manuscript.

16. validate the interactions with IHC, for instance show that cells that are close to each other bear receptor and ligand, respectively, and that this is changed in AD in DS.

We thank the reviewer for the suggestion. We have now performed extensive immunohistochemistry validations of various signaling molecules such as NECTIN, ANGPTL, and CD99. This comprehensive approach resulted in verifying an increased astrocytic activity of ANGPTL4 signaling in AD in DS samples, as shown in Figure 7m. We also identified a neuronal downregulation of NECTIN in both late-stage and AD in DS (Figure 7g). Moreover, we examined the colocalization of the ligand and receptor CD99 and CD99L2, as suggested by the reviewer and demonstrated in Figures S57f-g, and found downregulation in late-stage AD and AD in DS.

17. Figure 7 – in Panel 7g, j, there are clear differences between Phospho-Tau-AT8 in neurons and astrocytes between AD in DS and late AD. The authors can discuss these findings in light of the transcriptomics similarities and dissimilarities found early in the paper.

Point-by-point response to reviewers

We thank the reviewer for the suggestion and agree that the increased phospho-tau burden in AD in DS compared to late-stage AD is intriguing. We additionally found greater amyloid deposition, altogether indicating higher pathological load in AD in DS. Despite this increased pathology, we remarkably do not see overt differences between AD in DS and late-stage AD in the transcriptome. It is unclear currently if this suggests a maximal effect of pathology on the transcriptome or intrinsic differences between late-stage AD and AD in DS. We acknowledge our present study is limited due to the focus on amyloid with the use of the 5xFAD and hope to further explore the individual and combined effects of amyloid and tau pathology in a future study.

18. Figure 8

- The authors refer that “We found that the distribution of amyloid pathology in the spatial clusters was generally consistent with neuropathological plaque staging, where amyloid pathology is primarily in the cortical upper layers before spreading to the deeper layers (Figure 8c).” Nevertheless, when referring to Figure 3 and differential gene expression for the mouse spatial dataset, the authors had previously commented that “Upregulated genes at 4 months in the thalamus included disease-associated microglia (DAM) genes, like *Cst7*, *Tyrobp*, *Ctsd*, and *Trem2*, suggesting an early response to plaques localized to the thalamus, and with increasing age, we found upregulation of these genes across brain regions.” The authors should comment on these apparently contradictory results.

We apologize for the confusion and have attempted to better clarify between mouse and human findings throughout the text. The thalamus was only examined in the mouse samples, and the section the line is from was reporting DEG results found in the mouse. We did not examine any thalamic samples in the human dataset to be able to make any comment on thalamic pathology in clinical AD samples. Figure 8c is showing the amyloid quantifications in the human samples stratified by the neuropathological plaque staging determined by the pathologist (see Braak & Braak 1991 *Acta Neuropathologica* for the foundational paper of this measure). This measurement is not available for the mouse samples. In addition, taking the reviewer’s comment into account, we have included an additional supplementary figure (Fig. S58) demonstrating the amyloid distributions in 5xFAD by brain region to aid in clarifying the differences between the human and mouse datasets.

19. Referring to this figure, the authors mention “We found modest yet significant overlaps between some of these gene sets, with a generally higher degree of overlap in the plaque-associated genes”. However, earlier on, referring to Figure 3, the authors also state “module preservation analysis showed that the meta-modules identified in the human dataset were broadly preserved the mouse dataset”. Can the authors comment and elaborate on this apparent contradiction?

We agree with the reviewer that this is some potential for confusion based on this information presented in Figure 8 versus that presented in Figure 3. Based on the methods from Langfelder *et al.*, 2011 PLOS Comp. Bio., we performed module preservation testing of the human modules in the mouse dataset using orthologous genes. We report the results as The Z preservation summary statistic which represents a composite of other network preservation statistics into one convenient metric, as described by Langfelder *et al.* In general, this study describes modules with a Z preservation summary statistic greater than 10 to have strong evidence of preservation, less than 10 and greater than 2 to have a moderate or weak evidence of preservation, and less than 2 to have no evidence of preservation. Only module M9 had a Z preservation summary less than 2, but the inflammatory module M11 had the weakest evidence of preservation out of all the other modules. This means that the co-expression relationships of these orthologous genes in mice share enough similarities with their human

Point-by-point response to reviewers

counterparts to be considered preserved, but they also have key differences since this is a weak preservation. On the other hand, module M4 (synaptic transmission) has the highest degree of preservation so these co-expression relationships are much more similar across species. We believe that this module preservation analysis is indeed in line with the weak yet significant overlaps of the amyloid gene sets across species, showing that some of the mechanisms are shared yet there are critical species level differences. In the main text we have clarified this sentence to mention that M11 was only moderately to weakly preserved.

Other comments:

20. The authors use MAST for DE. Given the recent controversy regarding pseudobulk versus single-cell fitted DEG methods, it would be advisable that the authors perform pseudobulk analysis and compare the results.

We appreciate the reviewer for bringing attention to the methodology of our differential gene expression analysis. Indeed, the best practices for performing differential expression analysis in high-dimensional transcriptomics like spatial and single-cell are still not well established and are greatly debated within the field. In performing our initial data analysis we carefully considered our options with regards to differential expression analysis, and here we briefly summarize the current controversy regarding single-cell DEGs, including references to key literature, and we justify our choice of methodology.

“Pseudoreplication bias” is a common statistical pitfall in single-cell genomics and many other scientific disciplines. In the case of single-cell and spatial transcriptomics, pseudoreplication bias arises since we are measuring different cells or spatial spots that arise from the same biological sample, meaning that they share a common background and are not statistically independent from one another. A 2021 Nature Communications paper from Zimmerman, Espeland, and Langfeld discussed this phenomenon at length in the context of single-cell RNA-seq. A core tenet within the current debate on DEG methodology is controlling type 1 error (false positive) and type 2 error (false negative). The study from Zimmerman, Espeland, and Langfeld concluded that using two-part hurdle mixed models (like MAST) while adjusting for random effects (like individual or batch) helps to control type 1 error. The authors of this study also warn that false positives may be present in the results of this kind of test if batch effects are not properly taken into account in clustering analysis, which we have accounted for in our study using scANVI for the snRNA-seq dataset and Harmony in the spatial transcriptomics data. On the other hand, the authors note that computing “pseudo-bulk” replicates (computing the sum or mean of gene expression from all cells of a given cluster or cell type for each biological sample) prior to DEG analysis can help account for the pseudoreplication bias. However, the authors point out several issues that can arise with pseudo-bulk approaches, for example when numbers of cells within different cell types are imbalanced across different individuals, and that aggregating signals may ignore the variability and cellular heterogeneity within an individual. In this study, the authors suggest the use of mixed effects models to address pseudoreplication bias, highlighting that pseudo-bulk methods might lead to an increased risk of type 2 errors, reduced power, and potentially misleading outcomes, especially in scenarios involving rare cell types and imbalanced groups. Intriguingly, a contrasting perspective was offered in a study by Murphy and Skene, published as a response to Zimmerman, Espeland, and Langfelder's work. Murphy and Skene contend that pseudo-bulk approaches are more effective in controlling type 1 errors compared to other methods. However, it's important to recognize that both of these differential gene expression (DEG) benchmarking studies were based on simulated datasets. Such datasets may not accurately capture the complexities and nuances of real-world research designs, particularly those involving postmortem human tissue, as in our current study.

Point-by-point response to reviewers

A highly cited study titled “Confronting false discoveries in single-cell differential expression” from Squair et al was also published in Nature Communications in 2021, providing more context to the conversation regarding statistical analysis for DEGs in single-cell data. Here the authors sought to benchmark different DEG tests on real data, using datasets where paired bulk and single-cell RNA-seq data were available from the same cell populations. The authors of this study considered the bulk RNA-seq data as a “ground truth” to benchmark against the single-cell data, which is a critical assumption in itself since actual “ground truth” is often difficult or impossible to define in genomics data analysis. A main result from this study is that the top performing (scored using area under the concordance curve) methods out of fourteen methods tested were “pseudo-bulk” methods. We appreciate that pseudo-bulk methods may be a suitable approach for accounting for pseudoreplication bias, however the top performing methods in this study were likely biased towards “pseudo-bulk” methods since they were evaluated using the bulk RNA-seq data as a ground truth. In principle, “pseudo-bulk” data should behave more similarly to bulk RNA-seq in a DEG test than the unmodified single-cell data due, so it is not surprising that “pseudo-bulk” methods outperformed single-cell methods when benchmarked against bulk RNA-seq data. An important conclusion from this study is that “the central principle underlying valid DE analysis is the ability of statistical methods to account for the intrinsic variability of biological replicates”, which can be done using “pseudo-bulk” methods or models like MAST.

A key limitation of these DEG benchmarking studies is that they do not consider complex study designs which may include numerous samples arising from multiple sequencing batches, brain banks, or data that was generated from different research labs. This kind of study design is becoming more common as single-cell approaches are more readily adopted and since publicly available large-scale cell atlases are emerging in many different fields of biology. A recent study titled “Benchmarking integration of single-cell differential expression” was published in 2023, where Nguyen and Baik *et al.* benchmarked 46 different DEG workflows using simulated data and multiple real single-cell datasets where multiple batches were present. The results of this study showed that edgeR-based approaches, which were highly scoring in the study from Squair et al, were some of the worst at controlling false discoveries. However, the authors noted that other pseudo-bulk approaches as well as Wilcoxon and MAST were able to control false discoveries based on their analysis. Together, we can see that different benchmarking studies can result in different recommendations and interpretations, but it seems like the field generally agrees that DEG tests should somehow account for pseudoreplication bias and technical covariates. In our study, we chose to use sequencing batch as a key model covariate for DEG testing, and we deliberately designed our sequencing experiments to contain both cases and controls in each batch such that accounting for sequencing batch in our clustering analysis and in our DEG testing could effectively reduce unwanted technical variation.

As suggested by the reviewer and by the paper from Squair et al, we ran DEG analysis in the snRNA-seq dataset using pseudo-bulk replicates and edgeR as the DEG test. For this test, we are only using the AD in DS and control snRNA-seq data from the frontal cortex. In the study from Squair et al, and in their R package Libra which implements pseudo-bulk DE methods, the authors do allow for the user to account for additional covariates present in the dataset, for example RIN or postmortem interval, so we used the edgeR functions directly in a custom script. We first ran edgeR on pseudobulk replicates without any covariates, uncovering numerous DEGs across most of the cell populations (Response Figure 2). We next ran edgeR using the same covariates that we used in our MAST analysis, and we saw that this essentially eradicated the signal from most of the clusters in comparison to the analysis without the covariates (Response Figure 3). We know that no DEG test is perfect, and that there are inevitably going to be some false negatives and false positives in any analysis. On one hand, we saw many more DEGs than our MAST analysis when we did not use any covariates (potentially an increase

Point-by-point response to reviewers

in false positives), and we saw almost no significant DEGs in the pseudobulk edgeR analysis with covariates (increase in false negatives). It ultimately seemed unreasonable to use these results which can neither control type 1 nor type 2 error in comparison with our MAST DEG results.

Notably, in our revised manuscript the sex DEGs shown in the AD in DS group (Figure 4) changed slightly because we updated our DEG analysis to include Sample ID as one of the model covariates, which we did not do in the original analysis. This slightly changed our results, but the conclusions and overall interpretation of the data presented in that section of the manuscript have remained unchanged.

Response Figure 2: Pseudobulk differential gene expression analysis with edgeR

Volcano plots show the adjusted significance levels and effect sizes from the differential gene expression comparisons between AD in DS cases versus cognitively normal controls in the human snRNA-seq. The top and bottom five genes by effect size are annotated.

Point-by-point response to reviewers

Response Figure 3: Pseudobulk differential gene expression analysis with edgeR while accounting for sequencing batch, sequencing depth, and postmortem interval as model covariates

Volcano plots show the adjusted significance levels and effect sizes from the differential gene expression comparisons between AD in DS cases versus cognitively normal controls in the human snRNA-seq. The top and bottom five genes by effect size are annotated.

21. It would be also of interest to compare DEG from the snRNA-Seq and DEGs from the spatial analysis, to deconvolute the differences observed in the spatial data might be attributed to specific cell populations.

We thank the reviewer for the suggestion and we agree that it would be insightful to partition the transcriptomic differences in the spatial data by cell population. In principle, we can simply use the snRNA-seq dataset to “look up” which cell populations are expressing the spatial DEGs. Indeed, the heatmap that we show in Figure 2 attempts to make this point, where we visualized the expression of spatial DEGs in the snRNA-seq dataset. Based on the reviewer’s suggestion we have generated a new panel in Figure 2 to more explicitly deconvolve the spatial DEGs using the snRNA-seq dataset (Figure 2d). For the set of DEGs that are up- or down-regulated in disease in each of the spatial clusters, we checked which genes were specifically expressed in each cell population based on the snRNA-seq cluster marker gene analysis. For genes that were markers in more than one cell population, we only annotated it as a marker for the cell population that had the highest effect size in the marker gene test. This analysis allows us to inspect the proportion of the spatial DEG sets that are also marker genes in each snRNA-seq cell population. We performed this analysis separately for the three disease groups in

Point-by-point response to reviewers

the spatial transcriptomic dataset. We also performed this deconvolution analysis for the sex-level DEGs.

22. In the introduction, the authors refer 1-10 cell per spot as the resolution of ST. Does this correspond to the brain and where did these numbers obtained?

To clarify the reviewer's question, 1-10 cells per spot is the resolution reported by 10x Genomics, and they determined this number using 10µm mouse brain tissue sections (see <https://kb.10xgenomics.com/hc/en-us/articles/360035487952-How-many-cells-are-captured-in-a-single-spot->). We also sectioned our tissue at 10µm for Visium ST.

23. Many heatmaps throughout the paper are set to maximum scales avlog_2FC minor and higher than 1, which is a very limited dynamic range, and does not allow to identify process with large changes.

We understand the reviewer's concern about the maximum color scales for some of the heatmaps presented in this manuscript. We intentionally limited the range of these color scales so that the intermediate values are more interpretable instead of only the outliers with the largest values, which results in a more effective data visualization.

24. While the methods are well detailed, more QC information is needed, the authors need to comment on variability between patients, etc.

We appreciate the reviewer's comment and we agree that detailed QC information is necessary especially for single-cell and spatial -omics studies. We would be happy to provide any additional QC information as per the reviewer's request, however they did not specifically ask which QC information they would like to see other than the variability between patients. We generated a new figure (Supplementary Fig 3) that shows the difference between the number of grey matter spots and white matter spots in each sample to show the tissue-level variability across the different ST samples. We also provide new commentary about this in our discussion section.

Reviewer #3:

Remarks to the Author:

A. Summary of the key results - They identified regional differential gene expression changes shared between AD in the general population and DS, and this is done as the first snRNA-seq study of AD in DS. The finding that specific cellular changes that are conserved between both populations has important implications for clinical trial results in DS likely generalizing to AD that include robust sex difference. The identification of cross-species changes in a single mouse model are confirming discrepancies that may have contributed to poor preclinical translation. The dataset and analytics are a key result, and major contribution to the field.

B. Originality and significance: high

C. Data & methodology: validity of approach, quality of data, quality of presentation is high. N's well-powered and covariates addressed.

D. Appropriate use of statistics and treatment of uncertainties are addressed carefully/thoughtfully.

E. Conclusions: robustness, validity, reliability are high.

F. Suggested improvements: experiments, data for possible revision - none, very comprehensive datasets and analyses.

G. References: appropriate credit to previous work?

Point-by-point response to reviewers

Discussion of cross-species analysis and sources of variance should include discussion of genetic background pubmed.ncbi.nlm.nih.gov/30595332/, which improves alignment of snRNAseq data doi: doi.org/10.1101/2022.04.12.487877

H. Clarity and context: lucidity of abstract/summary, appropriateness of abstract, introduction and conclusions are well-written and accessible to cross-disciplines.

We appreciate the reviewer's encouraging remarks about the manuscript. The reviewer provided links to two relevant studies regarding mouse models of Alzheimer's. As suggested by the reviewer, we have expanded our discussion of sources of variation such as genetic background in cross-species analysis including these two studies.

Decision Letter, first revision:

9th Feb 2024

Dear Dr Swarup,

Your Article, "Spatial and single-nucleus transcriptomic analysis of genetic and sporadic forms of Alzheimer's Disease" has now been seen by 2 referees. You will see from their comments below that while they find your work of interest, a few remaining points are raised by Reviewer #2. We are interested in the possibility of publishing your study in Nature Genetics, but would like to consider your response to these concerns in the form of a revised manuscript before we make a final decision on publication.

To guide the scope of the revisions, the editors discuss the referee reports in detail within the team, including with the chief editor, with a view to identifying key priorities that should be addressed in revision and sometimes overruling referee requests that are deemed beyond the scope of the current study. In this case, we would like you to address Reviewer #2's requests in full. We hope that you will find the prioritized set of referee points to be useful when revising your study. Please do not hesitate to get in touch if you would like to discuss these issues further.

We therefore invite you to revise your manuscript taking into account all reviewer and editor comments. Please highlight all changes in the manuscript text file. At this stage we will need you to upload a copy of the manuscript in MS Word .docx or similar editable format.

*2) If you have not done so already please begin to revise your manuscript so that it conforms to our Article format instructions, available here.

*3) Include a revised version of any required Reporting Summary:

Please be aware of our guidelines on digital image standards.

[redacted]

We hope to receive your revised manuscript within four to eight weeks. If you cannot send it within this time, please let us know.

Nature Genetics is committed to improving transparency in authorship. As part of our efforts in this direction, we are now requesting that all authors identified as 'corresponding author' on published papers create and link their Open Researcher and Contributor Identifier (ORCID) with their account on the Manuscript Tracking System (MTS), prior to acceptance. ORCID helps the scientific community achieve unambiguous attribution of all scholarly contributions. You can create and link your ORCID from the home page of the MTS by clicking on 'Modify my Springer Nature account'. For more information please visit please visit www.springernature.com/orcid.

Sincerely,
Chiara

Chiara Anania, PhD
Associate Editor
Nature Genetics
<https://orcid.org/0000-0003-1549-4157>

Referee expertise:

Referee #1:

Referee #2:

Referee #3:

Reviewers' Comments:

Reviewer #1:

Remarks to the Author:

I think that the authors did a great job and that this manuscript is eligible for publication in Nature Genetics.

Reviewer #2:

Remarks to the Author:

The authors addressed my comments in a satisfactory manner , and I recommend publication of the manuscript. Nevertheless, a few points should be addressed before publication:

- Given the complexity of the data, it is absolutely essential that the dataset is available for browsing at CellxGene at the time of publication
- The authors still do not refer how many cells they detect per ST spot in their tissue, instead just have a general statement with 1-10 cells. Staining with DAPI will allow them to quantify and present this data, which is important for the interpretation of the data.
- Figure 4h,i – the quantification is not convincing, more ns are required to validate the increase of C1QB

Author Rebuttal, first revision:

Point-by-point response:

The authors addressed my comments in a satisfactory manner, and I recommend publication of the manuscript. Nevertheless, a few points should be addressed before publication:

- Given the complexity of the data, it is absolutely essential that the dataset is available for browsing at CellxGene at the time of publication

We agree with the reviewer that a publicly available data exploration portal would make an important addition to our study. We have been working with CellxGene to create a data collection to store our human snRNA-seq dataset, our human Visium ST dataset, and our mouse Visium ST dataset to CellxGene. We have created a CellxGene collection for our project at the following link: <https://cellxgene.cziscience.com/collections/7c1fbbae-5f69-4e3e-950d-d819466aeb2>

At this time we are still working with the CellxGene team to finalize our data collection, but this will certainly be publicly available at the time of publication as suggested by the reviewer.

- The authors still do not refer how many cells they detect per ST spot in their tissue, instead just have a general statement with 1-10 cells. Staining with DAPI will allow them to quantify and present this data, which is important for the interpretation of the data.

We thank the reviewer for their comment regarding the number of cells in each ST spot, and here we aim to clarify this point further. As noted by the reviewer, in principle a DAPI stain in the same section used for ST should allow us to quantify the number of nuclei present in each ST spot for each tissue section. However, we did not use a DAPI stain in our ST experiments, because instead we stained for amyloid beta using Amyloglo and OC which was critical for our study, making it technically impossible to stain for DAPI which has the same emission as Amyloglo. In our manuscript we do point out that we generally expect 1-10 cells per spot as reported by 10x Genomics in their study of the mouse brain. We also refer the reviewer to a Nature Neuroscience study from Maynard et al. where the authors did quantify the number of cells per Visium spot in human prefrontal cortex brain tissue using image segmentation of histology images, and they concluded that there was an average of 3.3 cells per spot.

Additionally, we initiated contact with 10x Genomics to explore alternative methodologies for ascertaining the count of nuclei per spot. Regrettably, it has been confirmed that there exists no alternate method for this determination. Ultimately we do not think that the number of cells per spot is critical information for our conclusions, and it is not possible for us to perform this quantification given our experimental design where we prioritized amyloid staining over nuclei staining. We want to emphasize that this caveat of the study has no bearing on the any analysis or results of the study, but rather is the technological limitation of the Visium platform.

- Figure 4h,i – the quantification is not convincing, more ns are required to validate the increase of C1QB

To address the reviewer's concern, we repeated these experiments in additional samples in order to provide a more convincing result. In our updated manuscript, we have n=5 brain sections per sex from n=3 donors and show that there is indeed a significant increase (p-val = 0.0215, T-test) of C1QB in females relative to males. We note that the scale of the y-axis in the bar plot has changed in our new figure, and this is simply because our updated figure is analyzing 20X images while our previous figure was analyzing 5X images.

Decision Letter, second revision:

27th Apr 2024

Dear Dr. Swarup,

Thank you for submitting your revised manuscript "Spatial and single-nucleus transcriptomic analysis of genetic and sporadic forms of Alzheimer's Disease" (NG-A63173R1). It has now been seen by the original referees and their comments are below. The reviewers find that the paper has improved in revision, and therefore we'll be happy in principle to publish it in Nature Genetics, pending minor revisions to satisfy the referees' final requests and to comply with our editorial and formatting guidelines.

Thank you again for your interest in Nature Genetics Please and do not hesitate to contact me if you have any questions.

Congratulations!

Best wishes,
Chiara

Chiara Anania, PhD
Associate Editor
Nature Genetics
<https://orcid.org/0000-0003-1549-4157>

Reviewer #2 (Remarks to the Author):

The author answered my questions in an appropriate manner and the paper is suitable for publication. Nevertheless, there is one point that needs to be addressed (this can be checked at the editorial level, I do not need to review the manuscript again):

- The authors need to clarify in the Figure legend of Figure 4i what the error bars stand for, and the Immunofluorescence methods explain in more detail how white matter was defined and how possible differences between the background fluorescence between the samples was dealt with, since this could be critical.

Author Rebuttal, second revision:

Point-by-point response:

Reviewer #2 (Remarks to the Author):

The author answered my questions in an appropriate manner and the paper is suitable for publication. Nevertheless, there is one point that needs to be addressed:

- The authors need to clarify in the Figure legend of Figure 4i what the error bars stand for, and the Immunofluorescence methods explain in more detail how white matter was defined and how possible differences between the background fluorescence between the samples was dealt with, since this could be critical.

A: Error bar in 4i represents mean \pm standard error of mean.

For C1QB staining, we tried to dissect white matter (WM) enriched regions from postmortem fixed brain samples and verified the presence of both white and gray matter at a ratio of approximately 3:1 (for most male samples) and 2:1 (for most female samples) through visual inspection and subsequent confirmation with MOBP staining, ensuring accurate tissue characterization.

To control for potential variations in background fluorescence between samples, we utilized ImageJ to quantify the average mean fluorescence intensity of the background. We then applied background subtraction by deducting this value from the total mean fluorescence intensity, thereby minimizing the impact of uneven background signals on our fluorescence measurements.

Final Decision Letter:

26th Sep 2024

Dear Dr. Swarup,

I am delighted to say that your manuscript "Spatial and single-nucleus transcriptomic analysis of genetic and sporadic forms of Alzheimer's disease" has been accepted for publication in an upcoming issue of *Nature Genetics*.

Over the next few weeks, your paper will be copyedited to ensure that it conforms to *Nature Genetics* style. Once your paper is typeset, you will receive an email with a link to choose the appropriate publishing options for your paper and our Author Services team will be in touch regarding any additional information that may be required.

Your paper will be published online after we receive your corrections and will appear in print in the next available issue. You can find out your date of online publication by contacting the Nature Press Office (press@nature.com) after sending your e-proof corrections.

Before your paper is published online, we shall be distributing a press release to news organizations worldwide, which may very well include details of your work. We are happy for your institution or funding agency to prepare its own press release, but it must mention the embargo date and *Nature Genetics*. Our Press Office may contact you closer to the time of publication, but if you or your Press Office have any enquiries in the meantime, please contact press@nature.com.

Please note that *Nature Genetics* is a Transformative Journal (TJ). Authors may publish their research with us through the traditional subscription access route or make their paper immediately open access through payment of an article-processing charge (APC). Authors will not be required to make a final decision about access to their article until it has been accepted. Find out more about Transformative Journals

Authors may need to take specific actions to achieve compliance with funder and

institutional open access mandates. If your research is supported by a funder that requires immediate open access (e.g. according to Plan S principles) then you should select the gold OA route, and we will direct you to the compliant route where possible. For authors selecting the subscription publication route, the journal's standard licensing terms will need to be accepted, including <https://www.nature.com/nature-portfolio/editorial-policies/self-archiving-and-license-to-publish>. Those licensing terms will supersede any other terms that the author or any third party may assert apply to any version of the manuscript.

If you have not already done so, we strongly recommend that you upload the step-by-step protocols used in this manuscript to [protocols.io](https://www.protocols.io). [protocols.io](https://www.protocols.io) is an open online resource that allows researchers to share their detailed experimental know-how. All uploaded protocols are made freely available and are assigned DOIs for ease of citation. Protocols can be linked to any publications in which they are used and will be linked to from your article. You can also establish a dedicated workspace to collect all your lab Protocols. By uploading your Protocols to [protocols.io](https://www.protocols.io), you are enabling researchers to more readily reproduce or adapt the methodology you use, as well as increasing the visibility of your protocols and papers. Upload your Protocols at <https://www.protocols.io>. Further information can be found at <https://www.protocols.io/help/publish-articles>.

Thank you.

Sincerely,
Chiara

Chiara Anania, PhD
Associate Editor
Nature Genetics
<https://orcid.org/0000-0003-1549-4157>

Click here if you would like to recommend Nature Genetics to your librarian
<http://www.nature.com/subscriptions/recommend.html#forms>